# tACS motor system effects can be caused by transcutaneous stimulation of peripheral nerves

Boateng Asamoah[1], Ahmad Khatoun[1] & Myles Mc Laughlin[1]

Transcranial alternating current stimulation (tACS) is a noninvasive neuromodulation method which has been shown to modulate hearing, motor, cognitive and memory function. However, the mechanisms underpinning these findings are controversial, as studies show that the current reaching the cortex may not be strong enough to entrain neural activity. Here, we propose a new hypothesis to reconcile these opposing results: tACS effects are caused by transcutaneous stimulation of peripheral nerves in the skin and not transcranial stimulation of cortical neurons. Rhythmic activity from peripheral nerves then entrains cortical neurons. A series of experiments in rats and humans isolated the transcranial and transcutaneous mechanisms and showed that the reported effects of tACS on the motor system can be caused by transcutaneous stimulation of peripheral nerves. Whether or not the transcutaneous mechanism will generalize to tACS effects on other systems is debatable but should be investigated.

[1] Exp ORL, Department of Neurosciences, KU Leuven, B-3000 Leuven, Belgium. These authors contributed equally: Boateng Asamoah, Ahmad Khatoun. Correspondence and requests for materials should be addressed to M.M.L. (email: myles.mclaughlin@kuleuven.be)

Neural entrainment is the synchronization of a population of neurons in the central nervous system to an external stimulus[1–3]. Applying a rhythmic stimulus to one of the sensory systems—auditory, visual or somatosensory—is known to cause neural entrainment: stimulation of the auditory nerve with acoustic[4–6] or electrical[7,8] repetitive stimuli results in an EEG response at the stimulus frequency. Similarly, looking at a flickering light rhythmically excites the optic nerve which leads to an enhanced brain oscillation at the flicker frequency[9–12]. Neural populations in the somatosensory system can be entrained using mechanically vibrating tactile stimuli and the response measured as an increase in EEG power at the modulation frequency[6,13–16]. Interestingly, stimulation of peripheral nerves in the arm using alternating current waveforms is also known to entrain brain activity[17]. These are all classical approaches for *indirectly* controlling brain oscillations—rhythmic stimulation of cranial or peripheral nerves causes entrainment of neurons in the brain.

Recently, transcranial alternating current stimulation (tACS) has been proposed as a new method to *directly* entrain neuronal populations in the cortex[18,19]. tACS electrodes are positioned on the scalp to target the cortex of interest and sinusoidally alternating electric current is passed between the electrodes. The current flows through the scalp, skull and cerebrospinal fluid before a weak current reaches the cortex. Figure 1 shows a quantitative illustration of the typical electric field strength in each tissue layer. tACS has been shown to modulate perception[20], motor function[21,22], cognition[23], memory[24,25] and hearing[26,27]. It is tacitly assumed that these effects are caused by the weak alternating current in the cortex directly modulating the membrane potential of a large population of neurons and causing neural entrainment. This transcranial mechanism of action is partially supported by evidence from brain slice and in vivo animal studies that have shown that electrical fields as low as 1 V/m can modulate the membrane potential[28] and cause neural entrainment[29,30].

However, the reported effects of tACS on behavior are controversial. Studies have shown that the skin and skull attenuate most of the current, with only a fraction reaching the brain[31,32] (see Fig. 1). For the typical amplitudes at which tACS is applied in humans, the electrical field reaching the cortex is less than 1 V/m: two recent studies made direct measurements of the electrical field strength caused by tACS in epilepsy patients with electrodes implanted on the cortex[32,33]. They found the electric field strength to be <1 V/m. Results from in vivo animal experiments and slice work show that electric fields of that strength cause only minimal or no neural entrainment[29–31].

Thus, there is a paradox: on the one hand tACS can have strong behavioral effects, but on the other hand, evidence suggests that the electric field reaching the cortex may not cause significant neural entrainment. We propose a new hypothesis that may resolve this paradox: the effects of tACS are caused by stimulation of peripheral nerves in the skin and not by direct cortical stimulation. The electric fields in the skin can be between 20 and 100 times stronger than in the cortex (Fig. 1). For 1 mA tACS, electric field strengths in the skin easily reach 20 V/m; well above the 4–6 V/m threshold for peripheral nerve stimulation[34]. The stimulation of peripheral nerves by tACS is further supported by the fact that most subjects report feeling a tingling sensation when the current is on[35,36]. Thus, we suggest that tACS motor system effects are mediated by transcutaneous and not transcranial stimulation. This proposed transcutaneous mechanism for tACS motor system effects fits with other classical methods that indirectly entrain brain oscillations through stimulation of cranial or peripheral nerves[4,9,14,17]. Whether or not our

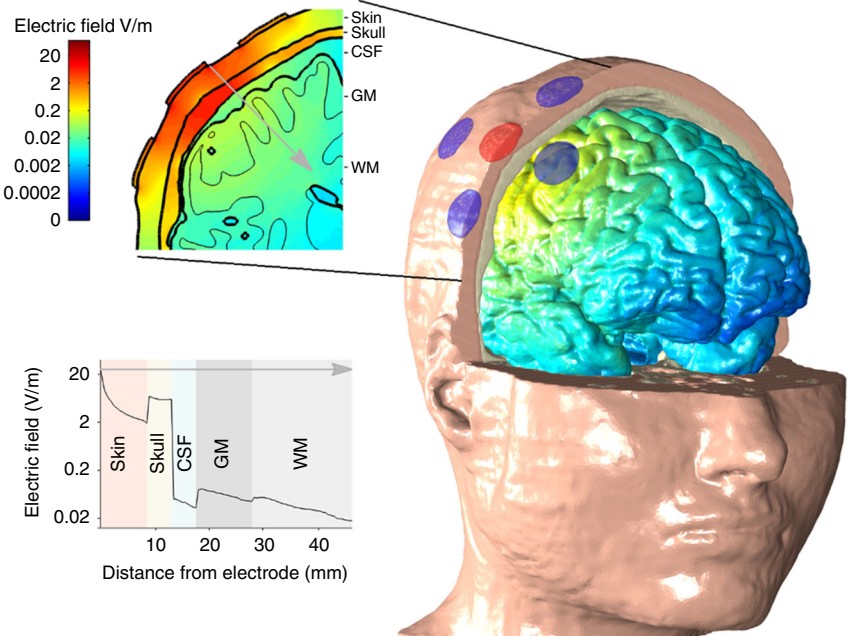

**Fig. 1** Computational model showing the electric field distribution in different tissues. Using a focused montage, 1 mA tACS is applied. The three-dimensional head model shows the electric field distribution over the cortical surface, with color encoding the field magnitude. The color scale is shown in the upper left panel—note that it is logarithmic. The skin and the skull have been partially cut away and the cerebrospinal fluid (CSF) is not shown. tACS electrodes are represented by circles with the anode in red and the cathodes in blue. A two-dimensional coronal, partial, cross-section is shown in the upper left panel. The logarithmic color scale is the same as in the 3D head model. The different tissues represented in the model are indicated—skin, skull, CSF, gray matter (GM) and white matter (WM). A one-dimensional plot of the electric field strength along the position indicated by the gray arrow is shown in the bottom left panel. This shows how the electric field magnitude decreases with distance from the electrode and how it is affected by each tissue type. Again, note the logarithmic scale on the axis showing the electric field. Electric field strengths in the skin can be 20–100 times higher than in the brain

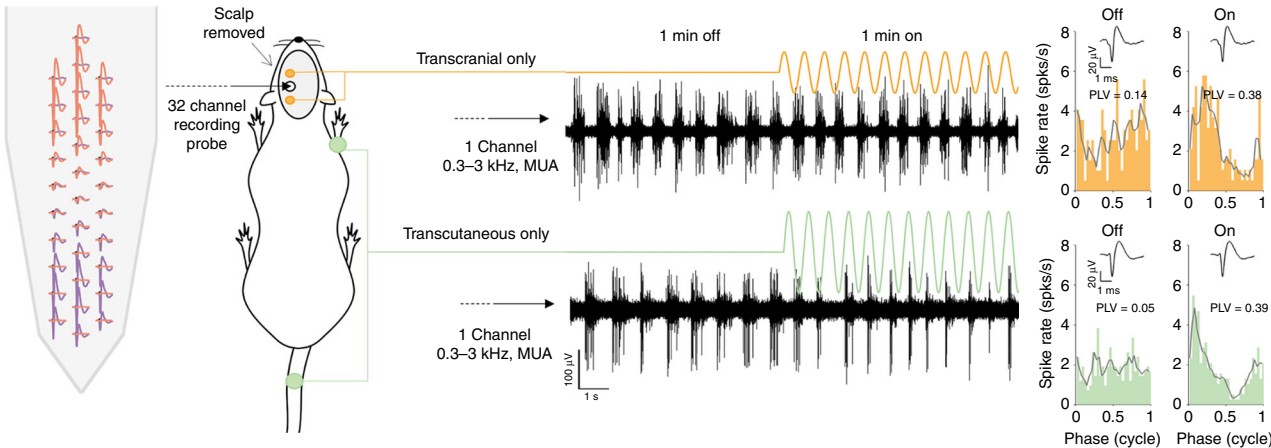

**Fig. 2** Setup to study transcranial- and transcutaneous-only stimulation in rat. A 32-channel silicon probe (far left) was inserted into the motor cortex through a burr hole and used to make extra-cellular recordings of slow "bursting" multiunit activity (MUA, two example recordings are shown in black). A large section of the scalp was retracted allowing transcranial-only sinewave stimulation to be applied through two micro-screws driven halfway through the skull. The orange line shows transcranial-only stimulation. During the experiment, 2-min recordings of MUA were made to compare 1 min with no stimulation (OFF condition), to 1 min with stimulation (ON condition). Transcutaneous-only stimulation (green line) was also a sinewave applied via crocodile-clip electrodes attached to the rectal temperature probe and the skin on one of the four limbs (contralateral fore shown; contra-hind, ipsi-fore and ipsi-hind were also tested). The same 2-min recordings of MUA with OFF and ON conditions were made for transcutaneous-only stimulation. Automatic (Klusta Suite) followed by manual spike sorting was used to isolate single-unit activity. An example of two single-units recorded from the same site are shown in pink and purple on the 32-channel probe. Single-unit spike times were used to calculate cycle histograms (far right panels) to compare the amount of neural entrainment when no stimulation was delivered (OFF, left column) to neural entrainment during stimulation (ON, right column). Neural entrainment in each condition was quantified by calculating the phase locking value (PLV, shown on each cycle histogram). The average single-unit spike waveform for each condition is shown as an inset. In this example, both transcranial-only and transcutaneous-only stimulation cause an increase in neural entrainment from the OFF to the ON conditions

hypothesis extends to non-motor system tACS effects is debatable, but could certainly be tested.

Here, we conduct four experiments to test the hypothesis that tACS motor system effects are caused by transcutaneous stimulation of peripheral nerves. In Experiment 1, we record single neuron activity in the rat motor cortex. We find that both transcranial and transcutaneous stimulation can entrain neural oscillations (~1 Hz) in the rat motor cortex. In healthy volunteers and essential tremor (ET) patients, we use tACS to entrain enhanced physiological (~10 Hz) and pathological tremor (~3 Hz) and investigate which mechanism, transcranial or transcutaneous, causes tremor entrainment. In Experiment 2, we apply a topical scalp anesthetic to block the transcutaneous mechanism. We show that anesthetizing the scalp significantly decreases the effect of tACS on tremor. In Experiment 3, we move the tACS electrodes to the contralateral arm, effectively blocking the transcranial mechanism. When the cortex was not being directly stimulated, tACS still causes significant tremor entrainment. Finally, in Experiment 4, we show that rhythmic stimulation of peripheral nerves in healthy volunteers entrains EEG beta activity (~20 Hz). Combined results from all experiments show that transcutaneous stimulation of peripheral nerves can account for tACS effects on the motor system across a wide range of frequencies. This suggests that, at least for the motor system, tACS may not work by directly stimulating neural populations. Rather, tACS may work similarly to other classical methods for controlling brain oscillations—it indirectly entrains central neural populations by stimulating peripheral sensory nerves.

## Results

**Experiment 1—electrophysiology in rats.** It has been previously demonstrated that tACS, matched to the frequency of ongoing oscillations, in anesthetized rat cortex causes neural entrainment[29]. In those experiments, the electrodes were positioned directly on the skull, thus eliminating the possibility that effects were caused by transcutaneous stimulation. We refer to this as transcranial-only stimulation. The aim of Experiment 1 was to establish whether transcutaneous-only stimulation could also cause neural entrainment in the rat motor cortex. To deliver transcranial-only stimulation, we removed the scalp and placed two stimulation electrodes directly on the skull to target the forelimb area of the motor cortex[37,38]. We ensured the skull was always dry to prevent current shunting to nerves in the skin. To deliver transcutaneous-only stimulation, we placed one stimulation electrode on one of the four limbs: contralateral (to the recording probe) fore, ipsilateral fore, contralateral hind or ipsilateral hind. The return electrode was always on the rectal temperature probe. We used a 32-channel silicon probe to record neural activity in the rat motor cortex under general anesthesia. During the experiment, we observed slow oscillatory, "bursting", multiunit activity (MUA)[29,39]. The setup and example recordings of MUA (black time-series data) are shown in Fig. 2. For both transcranial-only and transcutaneous-only, stimulation was a 1-min sinewave between 1 and 2.5 Hz (Fig. 2, orange and green sinewaves, respectively), chosen to match the frequency of the ongoing brain oscillation.

After the experiment, we applied spike sorting to isolate single unit activity[40]. Two example single units (pink and purple) are shown with different spatial positions on the 32-channel probe in Fig. 2. We recorded the response of 97 neurons to transcranial-only stimulation and 101 neurons to transcutaneous-only stimulation. We calculated spike time cycle histograms (i.e. a spike time histogram averaged over each stimulus cycle) for each single unit and then quantified neural entrainment by calculating the phase locking value (PLV). To test if there was an effect of stimulation, we compared PLV during periods of 1 min with no stimulation (OFF) to PLVs during periods of 1 min with stimulation (ON). During periods of no stimulation we assumed the same stimulus period as during stimulation. Figure 2 shows example OFF and ON cycle histograms for transcranial-only (top

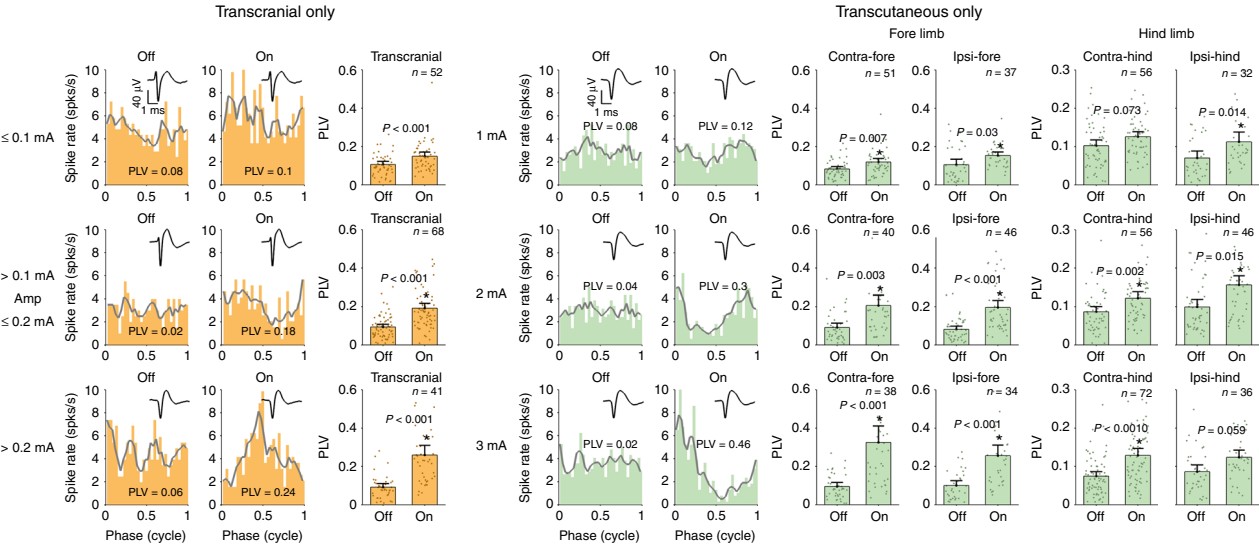

**Fig. 3** Effect of transcranial- and transcutaneous-only stimulation on neural entrainment. The rows show the effect of increasing stimulation amplitude on phase locking value (PLV). The transcranial-only data are grouped into three amplitude ranges: LOW (≤0.1 mA), MEDIUM (>0.1 mA, but ≤0.2 mA), HIGH (>0.2 mA). While the transcutaneous-only data were collected at three fixed amplitudes: LOW (1 mA), MEDIUM (2 mA), HIGH (3 mA). Entrainment increased with increasing amplitude (linear model followed by one-sided Wilcoxon sum rank). As in Fig. 2, the PLV values quantifying the amount of neural entrainment are reported on the example cycle histograms. The bar graphs compare the mean PLV for stimulation OFF and ON for each amplitude condition. Error bars show the confidence intervals. A star indicates that there was a significant increase in neural entrainment from stimulation OFF to ON for that particular condition. For transcutaneous-only stimulation, four different electrode configurations were tested: contralateral (to the 32-channel recording probe) fore, contralateral hind, ipsilateral fore and ipsilateral hind. The fore limb electrode configurations caused more neural entrainment than the hind limb configurations. In general, we found that transcutaneous-only stimulation produces similar patterns of neural entrainment to transcranial-only stimulation. (Wilcoxon signed rank test, one-sided, Bonferroni corrected for transcranial and transcutaneous)

two right panels in orange) and transcutaneous-only (bottom two right panels in green, contralateral fore limb). Comparing the OFF and ON cycle histograms for transcranial-only stimulation shows that an action potential is more likely to occur at one particular phase range in the ON histogram than in the OFF. Accordingly, the PLV increases from 0.14 in the OFF condition to 0.38 in the ON. In other words, transcranial-only stimulation entrains neural activity. Interestingly, comparing the OFF and ON cycle histograms for transcutaneous-only stimulation shows a remarkably similar effect. In this example, transcutaneous-only stimulation (contralateral fore) causes the PLV to increase from 0.05 in the OFF condition to 0.39 in the ON. Thus, transcutaneous-only stimulation also entrains neural activity.

For transcranial-only stimulation, we collected single-unit data in response to a range of sinewave stimulation amplitudes between 0.025 and 0.5 mA (peak-amplitude). A table showing the complete range of amplitudes collected for each neuron is shown in Supplementary Table 1. To compare animal tACS results with human tACS results, it is important to calculate the electric field strength generated in the brain. With our setup, we found that transcranial-only stimulation at 0.1 mA generated an electric field of approximately 0.9 V/m, averaged across all animals and recording sites. These values scaled linearly, meaning that an amplitude of 0.2 mA generated approximately 1.8 V/m, etc (see Supplementary Figure 1). The orange panels on Fig. 3 show data for all the neurons in response to transcranial-only stimulation grouped together for LOW, MEDIUM and HIGH amplitude ranges: amplitude ≤0.1 mA, 0.1 < amplitude ≤ 0.2 mA, amplitude > 0.2 mA. Example OFF and ON condition cycle histograms and PLVs are shown for each amplitude range with insets showing the average spike waveform. The bar graphs show the mean PLV for each amplitude range, dots show the individual PLVs, with the error bars showing confidence intervals. Comparison of the OFF and ON conditions showed that transcranial-only stimulation

caused significant neural entrainment in all three amplitude groups (Wilcoxon signed rank test, one-sided, LOW amplitude group: $Z = -3.7$, $p < 0.001$, MEDIUM amplitude group: $Z = -5.8$, $p < 0.001$, HIGH amplitude group: $Z = -5.4$, $p < 0.001$, Bonferroni corrected $p$-values are reported ($n = 3$)).

For transcutaneous-only stimulation, we collected single-unit data in response to sinewave stimulation with amplitudes of 1, 2 and 3 mA (peak amplitude). A table showing the complete range of amplitudes and stimulation configurations collected for each neuron is shown in Supplementary Table 2. These amplitudes are similar to those used in human tACS, although it should be cautioned that factors such as electrode size and tissue dimensions will play an important role in determining the actual electric field strength in the tissue[41]. Analysis of the signals recorded on the 32-channel probe during transcutaneous-only stimulation showed that very little, if any, current from the limb reached the brain via volume conduction (see Supplementary Figure 2). The green panels in Fig. 3 show the data for each of the four transcutaneous-only stimulation configurations. The example cycle histograms are for the contralateral fore configuration. Additional examples are shown in Supplementary Figure 3. Comparison of the OFF and ON conditions showed that transcutaneous-only stimulation caused significant neural entrainment in all three amplitude groups and for all four stimulation configurations except for Contralateral hind at 1 mA and ipsilateral hind stimulation at 3 mA (Wilcoxon signed rank test, one-sided, Bonferroni corrected ($n = 12$) $p$-values and $Z$ values are as follows: contra-fore: 1 mA $Z = -3.2$, $p = 0.007$; 2 mA $Z = -3.5$, $p = 0.003$; 3 mA $Z = -4.6$, $p < 0.001$; ipsi-fore: 1 mA $Z = -2.8$, $p = 0.03$; 2 mA $Z = -5.1$, $p < 0.001$; 3 mA $Z = -4.3$, $p < 0.001$; contra-hind: 1 mA $Z = -2.5$, $p = 0.07$; 2 mA $Z = -3.6$, $p = 0.002$; 3 mA $Z = -4.8$, $p < 0.001$; ipsi-hind: 1 mA $Z = -3.0$, $p = 0.014$; 2 mA $Z = -3.0$, $p = 0.015$; 3 mA $Z = -2.6$, $p = 0.058$). Supplementary Figure 4 presents a similar analysis of the

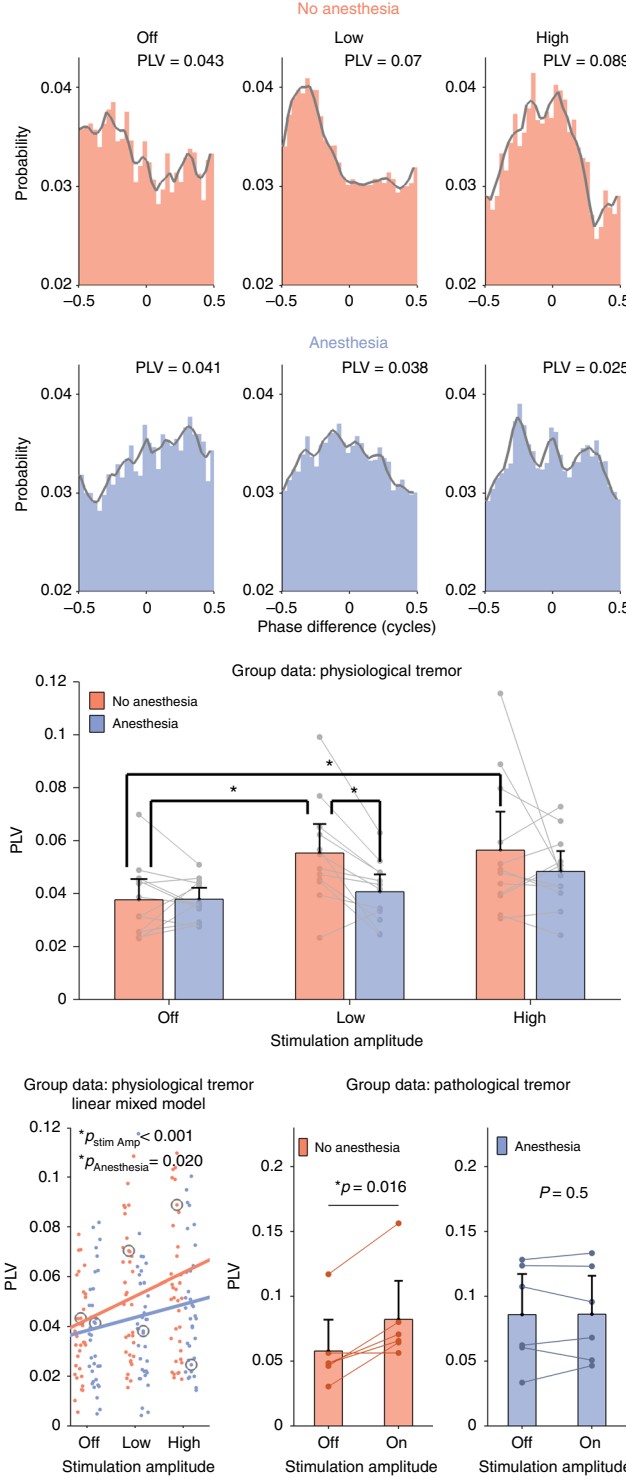

**Fig. 4** tACS entrainment of physiological and pathological tremor. Effects of focused tACS applied over the motor cortex physiological tremor were tested for three amplitude conditions (OFF, LOW and HIGH) and two topical scalp anesthesia conditions—no anesthesia (blue) and with anesthesia (red). The top two rows show phase-difference histograms and corresponding phase locking values (PLVs) that, in this experiment, quantify the amount of tremor entrainment. The top row shows that increasing tACS amplitude caused an increase in tremor entrainment in one example subject during the no anesthesia condition. The next row shows data from the same subject when the contribution from transcutaneous stimulation of peripheral nerves in the skin was reduced using topical anesthesia. In this condition, increasing stimulation amplitude does not increase tremor entrainment. The gray lines show the smoothed version of the histogram bin values and serve only to highlight the contour. The panel in the third row shows the mean PLV for each condition for all subjects (n = 12). Individual data points (averaged over three repetitions) from the same subject are shown as gray dots and connected with a line. Error bars show the confidence intervals. Significantly different conditions (exploratory post-hoc testing) are indicated with a star. The bottom left panel shows each individual data point (repetitions not averaged), with the gray circles indicating the examples shown in the top two rows. The trend lines were calculated by fitting a linear mixed model to the PLV data and using amplitude and anesthesia as predictors. As expected, increasing stimulation amplitude causes a significant increase in tremor entrainment ($p < 0.001$). Interestingly, anesthetizing the scalp causes a significant decrease in tremor entrainment ($p = 0.020$). The center and right panels on the bottom row show data from a similar experiment on 12 essential tremor patients with two amplitude conditions (OFF and ON). In the group that received no topical anesthesia ($n = 6$, center panel), motor cortex tACS entrained pathological tremor. In the group that did receive topical anesthesia ($n = 6$, right panel), motor cortex tACS did not entrain pathological tremor (one-sided Wilcoxon signed rank). Individual patient data points (averaged over five repetitions) are shown as dots and lines connect the same subject. Error bars show confidence intervals

effect of transcranial and transcutaneous-only stimulation on spike-rate. None of the stimulation conditions caused a significant increase in spike rate.

So far, our analysis of the single-unit data has shown that both transcranial-only and transcutaneous-only stimulation can cause neural entrainment—i.e. PLVs increased significantly between the OFF and the ON conditions for a range of stimulation amplitudes. Next, we wanted to test if increasing the amplitude for transcranial-only stimulation caused an increase in neural entrainment. Therefore, for each neuron, with each stimulation amplitude, we calculated the difference in PLV from OFF to ON condition, giving the $PLV_{dif}$. We then used a linear model to test the effect of stimulation amplitude on $PLV_{dif}$. We found that increasing stimulation amplitude caused an increase in $PLV_{dif}$ (linear model $p < 0.001$). Post-hoc testing showed that the HIGH amplitude group had significantly more neural entrainment than the LOW amplitude group (one-sided Wilcoxon sum rank test, $Z = -4.5$, $p < 0.001$); as did the MEDIUM group in relation to the LOW group (one-sided Wilcoxon sum rank test, $Z = -2.9$, $p = 0.006$). Compared to the MEDIUM group, the HIGH group also showed significantly more neural entrainment (one-sided Wilcoxon sum rank test, $Z = -2.3$, $p = 0.033$). The reported $p$-values are Bonferroni corrected for multiple comparisons ($n = 3$).

Finally, we tested if the transcutaneous-only data showed a similar increase in neural entrainment with increasing stimulation amplitude. We also tested if neural entrainment was affected by the stimulation configuration. We used a linear model to test the effect of stimulation amplitude and configuration on $PLV_{dif}$. Just as with transcranial-only stimulation, we found that increasing the transcutaneous-only stimulation amplitude caused an increase in neural entrainment (linear model, $p < 0.001$). We also found that neural entrainment was affected by stimulation configuration (same linear model, $p < 0.001$). There was a significant interaction between stimulation amplitude and configuration (same linear model, $p < 0.001$).

To test which transcutaneous amplitudes gave the most neural entrainment, we did post-hoc comparisons for each of the three amplitudes within one electrode configuration, yielding 12 comparisons for all four configurations. To test which

transcutaneous electrode configurations gave the most neural entrainment, we did post-hoc comparisons for configurations using only the data from the 3 mA conditions (six comparisons). This yielded 18 comparisons in total and reported $p$-values are Bonferroni corrected accordingly.

The recording electrode targeted the contralateral forelimb area. Accordingly, the contralateral fore configuration caused significantly more neural entrainment than the two hind limb configurations (two-sided Wilcoxon sum rank test: contralateral hind $Z = -4.2$, $p < 0.001$ and ipsilateral hind $Z = -4.0$, $p = 0.001$) but not the ipsilateral fore configuration ($Z = -1.1$, $p = 1$). The ipsilateral fore configuration gave significantly more entrainment that the two hind limb configurations (two-sided Wilcoxon sum rank test, contralateral hind $Z = -3.1$, $p = 0.032$ and ipsilateral hind $Z = -3.2$, $p = 0.025$). The two hind limb configurations did not give significantly different amounts of neural entrainment (two-sided Wilcoxon sum rank test, $Z = -0.8$, $p = 1$). For all the amplitude comparisons, only the contralateral fore at 3 mA caused significantly more neural entrainment in comparison with 1 mA for the same configuration (one-sided Wilcoxon sum rank test, $Z = -4.6$, $p < 0.001$). All other comparisons showed no significant difference in neural entrainment, probably due to the strict Bonferroni correction. Further analysis detailing the specific percentage of neurons showing an increase in entrainment for each condition is presented in Supplementary Table 3. Combined with the first statistical analysis, looking specifically at the increase in neural entrainment from the OFF to the ON conditions, our results indicate that any transcutaneous stimulation configuration will give some amount of neural entrainment at the population level.

**Experiment 2—block the transcutaneous mechanism in humans.** tACS can entrain both pathological[42] and enhanced physiological tremor[43,44], and has been shown to reduce pathological tremor amplitude in Parkinson's disease patients[21]. The tacit assumption is that these effects are caused by transcranial stimulation of cortical neurons. We wanted to test this assumption by using a topical skin anesthetic (lidocaine/prilocaine topical cream) to reduce any potential contribution from transcutaneous stimulation of peripheral nerves. Topical anesthetics block sodium channels in peripheral nerves in the skin, thereby stabilizing the membrane potential and increasing the threshold for firing an action potential[45,46].

In Experiment 2A, we tested 12 healthy volunteers (three female, mean $27 \pm 4$ years old, ± indicates standard deviation). We caused enhanced physiological tremor by asking the subjects to rest their wrist on a table with fingers spread. We applied tACS, matched to the individual's tremor frequency ($8.70 \pm 0.62$ Hz), at three different amplitudes (OFF, LOW and HIGH) over the motor cortex using a focused montage. Mean stimulation peak-amplitudes (not peak-to-peak), across all subjects, in LOW and HIGH condition were $0.588 \pm 0.267$ mA and $2.192 \pm 0.499$ mA, respectively (see Supplementary Table 4 for more details). A focused montage was chosen because it limits the spread of current to other cranial nerves such as the optic nerve, and it is known that both visual and electrical stimulation of the optic nerve also causes neural entrainment[9,43,47]. Tremor was measured via an accelerometer to the volunteer's middle finger. To measure how much tACS entrains, tremor histograms showing the phase difference between the tACS and tremor signal were calculated for each amplitude condition. During the OFF condition, a simulated tACS signal was used. PLVs were then used to quantify tremor entrainment. Finally, to examine the contribution of the transcutaneous stimulation a local anesthetic was applied to the scalp area under the electrodes. The effect of

tACS at each of the three stimulation amplitudes (OFF, LOW and HIGH) on tremor entrainment were measured in each volunteer during two sessions (on two different days)—one with skin anesthesia and one without—using a double blinded, randomized, crossover design. During each session, three repetitions for each amplitude condition were collected.

Figure 4 shows the results. The top two rows show example phase difference histograms for one individual subject and the bottom rows show group data. Individual data for each subject are shown as dots and as separate plots in Supplementary Figure 5. The top row shows data from one subject on the no anesthesia day, i.e. standard tACS. Tremor entrainment increases as tACS peak amplitude is increased from OFF to LOW and then HIGH. The PLV for each condition is indicated in the top right. The gray lines highlight the histogram contour. The second row shows data from the same subject during the session when the scalp was anesthetized, i.e. any potential contribution from the transcutaneous mechanism has been reduced. When the scalp was anesthetized, increasing tACS amplitude did not cause a numerical increase in tremor entrainment.

The bar graph in the third row shows the mean PLVs for all 12 subjects in each condition (no anesthesia session in red, anesthesia session in blue) and the error bars show confidence intervals. Individual data points (averaged over three repetitions) are shown as gray dots with points from the same subject connected with a line. The bottom left panel shows the same data with each individual subject's three repetitions per amplitude condition and anesthesia/no anesthesia session shown as dots. The trend lines for the anesthesia and no anesthesia sessions are indicated (linear mixed model regression). The gray circles highlight the individual subject data shown in the top rows.

At the group level, when the scalp was anesthetized, the effects of tACS on tremor entrainment were significantly reduced (linear mixed model, $p = 0.020$). These results indicate that stimulation of peripheral nerves in the scalp plays a significant role in entraining tremor and thus are consistent with our hypothesis that tACS motor system effects are mediated by a transcutaneous mechanism. We also found that increasing tACS amplitude caused a significant increase in tremor entrainment (same linear mixed model, $p < 0.001$).

The interaction between tACS amplitude and scalp anesthesia was not significant (same linear mixed model, $p = 0.294$). This means that a post-hoc analysis would not allow us to draw firm conclusions as to which specific conditions caused more or less tremor entrainment. As an exploratory analysis, we did conduct a post-hoc testing and present the results without drawing firm conclusions. To ensure each observation was independent, we first averaged across each of the three repetitions to get one PLV per condition for each subject. We make nine comparisons between conditions—three between each amplitude for the no anesthesia conditions, three between each amplitude for the anesthesia conditions and three between the anesthesia and no anesthesia conditions for each amplitude. The $p$-values reported below have been Bonferroni corrected for multiple comparisons ($n = 9$).

The exploratory post-hoc analysis showed that the no anesthesia LOW and HIGH amplitude conditions caused more entrainment than the OFF condition (one-sided Wilcoxon signed rank test, $p = 0.031$ and $p = 0.031$, respectively). Cohen's effect size values ($d = 0.99$ and $d = 0.85$) suggested a high practical significance for LOW and HIGH, respectively. However, in the anesthesia condition, LOW and HIGH amplitude conditions did not cause an increase in entrainment compared to the OFF condition (one-sided Wilcoxon signed rank test, $p = 1$ and $p = 0.58$, respectively). In the LOW amplitude conditions, applying anesthesia caused a significant decrease in tremor entrainment

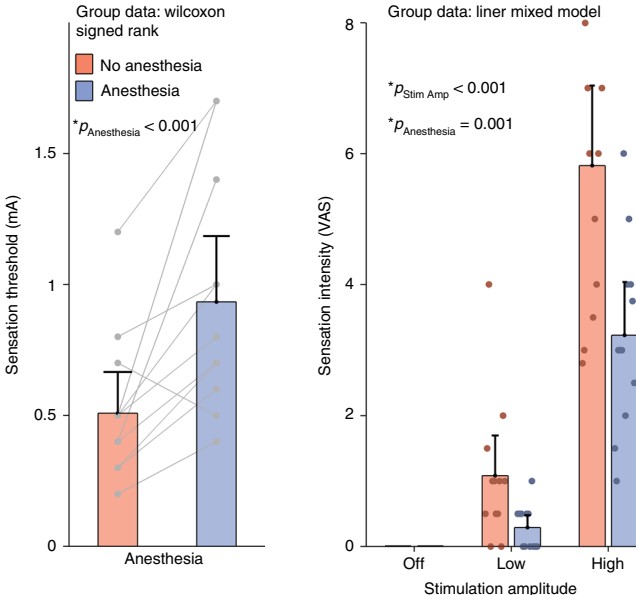

**Fig. 5** Sensation threshold and intensity ratings. The left panel shows the mean sensation threshold amplitude (mA) for the anesthesia (blue) and no anesthesia conditions (red). Error bars on both panels show the confidence intervals. Individual data points are shown as gray dots and lines connect points from the same subject. Topical scalp anesthesia caused a significant increase in sensation threshold ($p < 0.001$, one-sided Wilcoxon signed rank test). The right panel shows the mean sensation intensity rating for all subjects, using a visual analog scale (VAS) during OFF, LOW and HIGH amplitude stimulation conditions and during the anesthesia and no anesthesia conditions. Individual data points are shown as dots. The sensation intensity significantly increases with increasing stimulation amplitude ($p < 0.001$, linear mixed model). Additionally, the application of topical scalp anesthesia significantly reduces sensation intensity ratings ($p = 0.001$, linear mixed model). Both panels confirm that topical anesthesia does indeed block peripheral nerves in the scalp

(one-sided Wilcoxon signed rank test, $p = 0.015$). Cohen's effect size value ($d = 1.14$) suggests a large effect with a high practical significance. However, the difference between the two HIGH amplitude conditions was not significant (one-sided Wilcoxon signed rank test, $p = 1$). The two OFF conditions did not show different amounts of tremor entrainment (one-sided Wilcoxon signed rank test, $p = 1$).

In Experiment 2B, we used a similar procedure to test the effect of tACS in 12 ET patients (seven females, $71 \pm 9$ years old). Patients details are provided in Supplementary Table 5. All patients had advanced ET and had deep brain stimulation (DBS) electrodes implanted in the ventral intermediate nucleus of the thalamus. However, DBS was not completely effective and all patients had significant hand or arm tremor when DBS was on (mean tremor frequency $3.54 \pm 0.54$ Hz). With DBS off, all patients had very large tremor. Therefore, all tACS testing was conducted with DBS on.

The 12 patients were randomly divided into two groups of six. The first group received focused motor cortex tACS, with no scalp anesthesia, delivered at maximum of 2 mA (average $2 \pm 0$ mA) and matched to the individual tremor frequency ($3.33 \pm 0.52$ Hz). In the no anesthesia group, we found that tACS entrained pathological tremor (Fig. 4, bottom center panel) ($p = 0.016$, Wilcoxon signed rank test, one sided). Cohen's effect size value ($d = 1.69$) suggested a large effect with very high practical significance. In the second group, we applied topical anesthesia to the scalp and again delivered motor cortex tACS matched to the

individual tremor frequency ($3.75 \pm 0.52$ Hz). To increase our chances of observing a true transcranial effect of tACS on tremor, we increased the stimulation amplitude to a maximum of 5 mA (average $4.67 \pm 0.61$ mA). Our computational model estimates that this will increase the electric field in the motor cortex to 0.610 V/m, compared to 0.26 V/m in the first group. In the second group with anesthesia, in spite of the stronger electric field in the motor cortex, we still did not observe any effect of tACS on pathological tremor entrainment (Fig. 4 bottom right panel) ($p = 0.5$, Wilcoxon signed rank test, one sided). The OFF conditions between the two groups were not significantly different ($p = 0.132$, Wilcoxon rank sum, two sided). Individual patient data are shown in Supplementary Figure 7.

The lidocaine/prilocaine anesthetic cream blocks sodium channels in peripheral nerves[45]. Thus, we expected to observe a threshold increase for electrical stimulation. To verify this, we measured the perceptual threshold for tACS in both sessions (Fig. 5, left panel). Topical anesthesia did significantly increase the perceptual threshold for tACS (one-sided Wilcoxon signed rank test, $p = 0.001$). Subjects reported that the threshold for feeling tACS was $0.508 \pm 0.278$ mA in the no anesthesia session, increasing to $0.933 \pm 0.444$ mA in the anesthesia session. We also had subjects rate the perception of tACS on a visual analog scale (VAS) (Fig. 5 right panel). Increasing tACS amplitude caused a stronger perception and applying topical anesthesia caused a reduction in perception (linear mixed model, $p < 0.001$ and $p < 0.001$, respectively). There was a significant interaction between tACS amplitude and anesthesia ($p = 0.001$). Supplementary Figure 6 presents an analysis of the correlation between sensation intensity rating and tremor entrainment. Subjects who rated the sensation of the stimulation as being more intense showed higher levels of tremor entrainment (linear mixed model, $p = 0.001201$)

**Experiment 3—block the transcranial mechanism in humans.** In Experiment 3, we tested if physiological tremor could be entrained when the tACS electrodes were positioned on the upper arm contralateral to the tremor hand. This montage isolated the transcutaneous mechanism by completely blocking any contribution from the transcranial mechanism. We tested this on 12 subjects (five female, mean $27 \pm 5$ years old) using procedures similar to Experiment 2. The mean stimulation peak-amplitudes, in the OFF, LOW and HIGH condition were 0, $0.495 \pm 0.173$ mA and $2.475 \pm 0.086$ mA, respectively (see Supplementary Table 6) and the mean tremor frequency was $8.00 \pm 0.64$ Hz. The top three panels in Fig. 6 show phase difference histograms from one subject. As tACS amplitude was increased, tremor entrainment also increased. Individual data for each subject are shown in Supplementary Figure 8. The bottom left panel on Fig. 6 shows the mean PLV for all 12 subjects. The middle bottom panel shows the same data with each individual subject's three repetitions per amplitude condition shown as dots and the trend line indicated (linear mixed model regression). At the group level we found that even when the tACS electrodes were not positioned on the head, increasing tACS amplitude still caused a significant increase in tremor entrainment (linear mixed model, $p = 0.013$). The LOW and HIGH amplitude conditions caused more tremor entrainment than the OFF condition (post-hoc, Bonferroni corrected, $n = 3$, one-sided Wilcoxon signed rank test, $p = 0.024$ and $p = 0.0103$, Cohen's $d = 0.80$ and 0.89, respectively). However, increasing tACS amplitude from LOW to HIGH did not increase in tremor entrainment (Bonferroni corrected $n = 3$, one-sided Wilcoxon signed rank test, $p = 1$). Finally, the right bottom panel shows the mean VAS perceptual ratings (error bars indicate confidence interval) An increase in tACS amplitude caused an increase in the sensation intensity (linear mixed model, $p <$

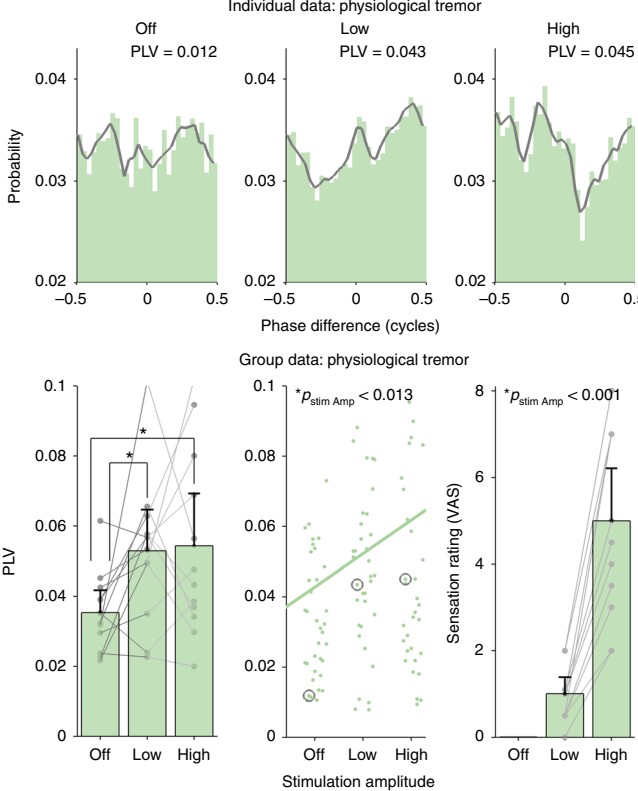

**Fig. 6** Effect of transcutaneous-only stimulation on physiological tremor. In healthy volunteers, tACS electrodes were placed on the upper arm contralateral to the hand on which tremor was measured, effectively blocking any contribution from a potential transcranial mechanism. The top row shows phase-difference histograms for the three different amplitude conditions (OFF, LOW and HIGH) in one example subject (similar to Fig. 4). Even when the tACS electrodes were not positioned on the head, stimulation still caused physiological tremor entrainment in this subject. The bottom left panel shows the mean PLV for each subject (n = 12) for each amplitude condition. Error bars on all panels show the confidence intervals. Individual data points (averaged over three repetitions) are shown as gray dots. The middle panel shows each individual data point (repetitions not averaged), with the gray circles indicating the examples shown in the top row. The trend line was calculated by fitting a linear mixed model to the PLV data and using stimulation amplitude as a predictor. Even when tACS electrodes are not located on the head, increasing stimulation amplitude still caused a significant increase in tremor entrainment (p = 0.013, linear mixed model). The right panel shows the mean sensation intensity rating for all subjects, using a visual analog scale (VAS) during each amplitude condition. Individual data points are shown as gray dots. The sensation intensity significantly increases with increasing stimulation amplitude (p < 0.001, linear mixed model). This set of data indicate that transcutaneous stimulation of peripheral nerves in the skin can entrain tremor

0.001). Four subjects completed Experiments 2A and 3. Their results are discussed in Supplementary Figure 9.

## Experiment 4—transcutaneous stimulation entrains EEG.
Experiment 4 tested if transcutaneous stimulation effects extend beyond tremor entrainment and beyond the frequency ranges tested in Experiments 2 (~10 Hz) and 3 (~3 Hz). We tested if transcutaneous stimulation (i.e. tACS electrodes positioned on the upper arm as in Experiment 3) could entrain neural activity in 12 healthy volunteers (3 female, average 27 ± 4 years old). We measured beta activity at rest (16–30 Hz) using a single-channel EEG system with one electrode positioned over the

somatosensory cortex and the reference over the motor cortex (FP3, CP3)[48]. It was not possible to completely separate the electrical artifact caused by sinewave stimulation on the arm from the EEG activity. Therefore, we used single, short duration, biphasic pulses (0.44 ms per phase) repeated at the desired frequency, giving a small, time limited artifact that was completely removed with a low-pass filter (Supplementary Figure 10, for additional controls see Supplementary Figure 11, and for sensation ratings see Supplementary Figure 12). Individual beta peak frequencies were first measured by recording 3 min of EEG and then averaging the power spectral density calculated over 1 s epochs. Most subjects showed a clear peak in the beta range (22.08 ± 1.87 Hz, Supplementary Figure 13). Then using tACS electrodes positioned on the upper arm, we delivered single-pulse stimulation matched to the individual's beta frequency. We tested three conditions: stimulation off (OFF), low amplitude (LOW) and high amplitude (HIGH). We determined the amplitude for the HIGH condition by first delivering a 2.5 mA sinewave at the beta frequency, asking the subject to rate this sensation intensity. We then increased the pulsed stimulation amplitude until it produced an equivalent sensation. The LOW amplitude was set to half of the HIGH amplitude. Average amplitude in the HIGH condition was 6.33 ± 1.23 mA and 3.17 ± 0.62 mA in the LOW condition. Note, pulsed stimulation injects <10% of the charge used for sinewave stimulation: pulsed 0.0028 mC per stimulus phase vs. sinewave 0.0398 mC per stimulus phase.

We quantified the effect of transcutaneous stimulation on the EEG signal using the same methods from Experiments 2 and 3. However, we now replaced the tremor signal with the EEG signal. The top row in Fig. 7 shows the phase difference histograms (now between the pulsed transcutaneous stimulation and the EEG signal) for one subject. The histogram in the OFF condition is flat and becomes progressively more peaked at one particular phase in the LOW and HIGH conditions. Group level PLVs (bottom left panel) showed a significant increase from OFF to LOW to HIGH, indicating that rhythmic transcutaneous stimulation entrains beta activity (linear mixed model, p < 0.001). Post-hoc analysis using one-sided Wilcoxon test (corrected for n = 3) showed a significant effect of LOW vs. OFF (p = 0.0015, d = 1.01), HIGH vs. OFF (p < 0.001, d = 1.21), HIGH vs. LOW (p < 0.001, d = 1.02).

As a second analysis, we calculated the amplitude of the EEG signal at the stimulation frequency. The middle row shows the amplitude spectrum calculated by taking the Fourier transform of the complete 3 min EEG recording. The red dots indicates the peak in the neural response at the stimulus frequency. The inset shows the averaged EEG time-series signal with an epoch duration selected to span three stimulus cycles. The light-green borders indicate confidence intervals. Transcutaneous stimulation caused an increase in the amplitude component at the stimulus frequency, associated with an increased brain oscillation. The group results for the Fourier amplitude analysis are shown in the bottom right panel. At the group level (bottom right panel) rhythmic transcutaneous stimulation caused an increase in the EEG Fourier-component at the stimulus frequency (linear mixed model, p < 0.001). Post-hoc using one-sided Wilcoxon test (corrected for n = 3) shows significant effects of LOW vs. OFF (p < 0.001, d = 0.99), HIGH vs. OFF (p < 0.001, d = 1.18) and HIGH vs. LOW (p < 0.001, d = 1.20).

## Discussion
The tACS research field is currently faced with a paradox: there are now hundreds of studies that show significant effects of tACS on brain oscillations[49,50], perception[20], motor function[21,22], cognition[23], memory[24] and hearing[26,27]. However, there is also

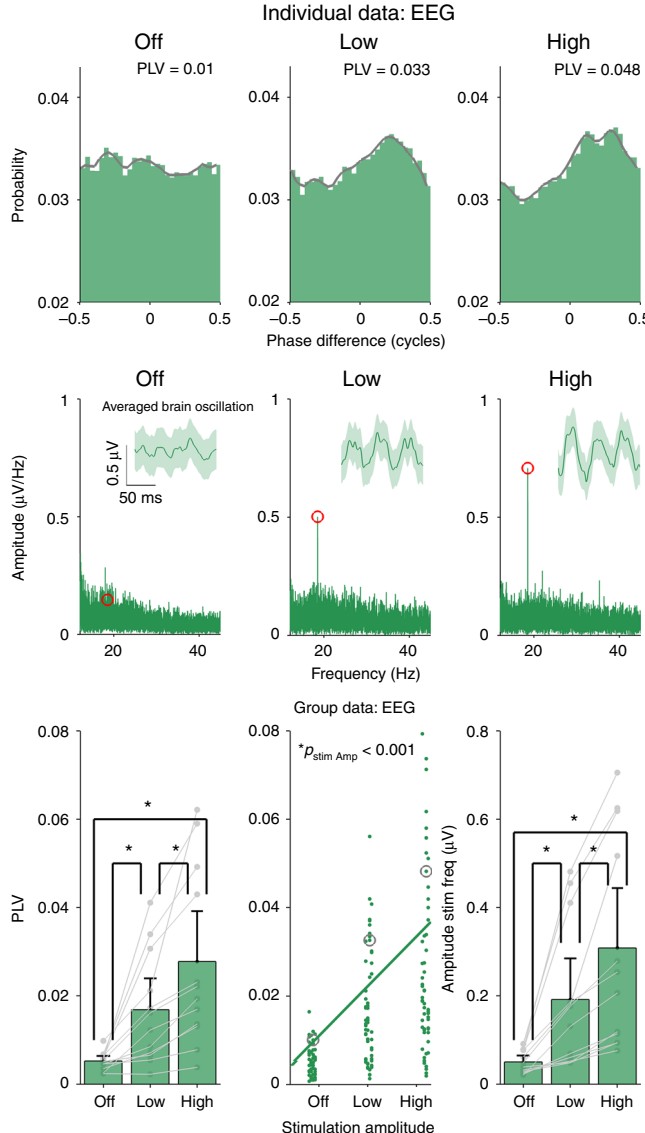

**Fig. 7** Effect of transcutaneous-only stimulation EEG beta activity. We measured EEG beta activity and applied rhythmic single-pulse transcutaneous stimulation to the contralateral upper arm at three intensities (OFF, LOW and HIGH). The top three panels show how EEG activity measured from the somatosensory-motor cortex entrains to rhythmic transcutaneous stimulation. These phase difference histograms and PLVs were calculated in the same way as those in Figs. 4 and 6, but the tremor signal was now substituted with the EEG signal. The bottom left panel shows the mean PLV for each subject ($n = 12$) for each amplitude condition. Error bars on all panels show the confidence intervals. Individual data points (averaged over five repetitions) are shown as gray dots. The middle panel shows each individual data point as green dots (repetitions not averaged), with the gray circles indicating the examples shown in the top row. The trend line was calculated by fitting a linear mixed model to the PLV data and using stimulation amplitude as a predictor. Increasing the amplitude of rhythmic transcutaneous stimulation caused a significant increase in EEG entrainment ($p < 0.001$). Post-hoc testing showed that all conditions were significant different from each other. The middle row shows the amplitude Fourier spectrum of the EEG signal from the same subject in the top row. Note how the amplitude of the component at the stimulation frequency (red dot) increases as stimulation goes from OFF to LOW to HIGH. The inset shows the EEG time-series signal averaged over an epoch spanning three stimulus cycles. Note how rhythmic transcutaneous stimulation is associated with an increased neural oscillation in the somatosensory-motor cortex at the stimulation frequency. The light-green areas show confidence intervals. The bottom right panel shows the mean amplitude of the Fourier component at the stimulus frequency for each subject for each stimulation condition. Individual data points (averaged over five repetitions) are shown as gray dots with lines connecting subjects. Increasing the amplitude of rhythmic transcutaneous stimulation caused a significant increase in the amplitude of the neural oscillation ($p < 0.001$, linear mixed model). Post-hoc testing using a one-sided Wilcoxon test showed that all conditions were significantly different from each other

evidence that the electrical field reaching the cortex may not be strong enough to cause neural entrainment[31,33]. How can these opposing sets of results be reconciled? For motor system effects, we suggest that the tACS paradox can be resolved by recognizing that effects are mostly caused by transcutaneous stimulation of peripheral nerves.

Results from our study provide the first direct evidence that many tACS motor system effects can be accounted for by transcutaneous stimulation of peripheral nerves. In Experiment 1, we showed that in rats, transcutaneous-only (or peripheral nerve) stimulation causes entrainment of motor cortex. Entrainment patterns were similar for both transcutaneous-only and transcranial-only stimulation. This showed that both the transcranial and the transcutaneous mechanisms can cause neural entrainment. The next step was to test which mechanism was driving tACS motor system effects. In Experiment 2, we blocked the transcutaneous mechanism by applying topical scalp anesthesia under tACS electrodes positioned over the motor cortex. We found that anesthetizing the scalp significantly reduced tremor entrainment. In Experiment 3, we isolated the effects of the transcutaneous mechanism by placing the tACS electrodes on the upper arm contralateral to the tremor hand. We found that transcutaneous-only stimulation in healthy volunteers caused

tremor entrainment, similar to that observed when tACS electrodes were positioned on the head without scalp anesthesia. In Experiment 4, we showed that rhythmic transcutaneous stimulation of peripheral nerves can entrain EEG beta activity recorded from the motor-somatosensory cortices. This finding is in line with classical methods for causing steady-state neural responses by stimulating the auditory[4–6], visual[9–12] or somatosensory systems[6,13–16]. Our combined results show that while tACS does generate a weak electric field in the brain (and this may or may not cause small amounts of neural entrainment in the human motor cortex), this field does not make a significant contribution to the observed motor system effects. Rather, tACS motor system effects appear to be dominated by the transcutaneous stimulation of peripheral nerves.

The effects of tACS on both pathological[21,42] and enhanced physiological tremor[43,51] have been well studied and replicated[44]. This makes tACS effects on tremor a good model to study tACS mechanisms in humans. Our results show that the effects of tACS on tremor appear unlikely to be mediated by transcranial stimulation of neurons in the motor cortex but instead are dominated by transcutaneous stimulation of peripheral nerves in the skin. tACS in the HIGH amplitude conditions is certainly strong enough to stimulate peripheral nerves as evidenced by the sensation ratings (see Fig. 5, right panel and 6, bottom right). tACS in the LOW amplitude conditions is just under the perceptual threshold for some subjects but above for others. Indeed this is supported by the average sensation threshold of 0.5 mA in the no anesthesia condition (Fig. 5, left panel). Even in these LOW amplitude conditions, just under the perceptual threshold, it is

likely that some peripheral nerves are still excited. Rhythmic excitation of these nerves will give rhythmic input to the contralateral somatosensory cortex via the thalamus[52]; because of sensorimotor connections[53] motor cortex activity would then entrain. There are interhemispheric projects between the left and right sensorimotor cortices that could facilitate entrainment in the ipsilateral hemisphere during peripheral nerve stimulation. These functional ipsilateral connections have been shown in various studies[54,55].

tACS effects have been shown to be present across a wide range of frequencies with differing mechanisms suggested to mediate effects within different frequency ranges[56]. Results presented here show that transcutaneous stimulation effects the motor system across a wide range of frequencies: 1–2 Hz in the rat brain, 3 Hz for pathological tremor, 10 Hz for physiological tremor and 20 Hz for EEG beta activity. Thus, the role of the transcutaneous mechanism must now be carefully considered in all studies that report a tACS effect on motor cortex function. Published studies should be reinterpreted with the transcutaneous mechanism in mind and new studies being designed must carefully consider the transcutaneous mechanism.

Many tACS studies do not focus on the motor system, but instead use tACS to modulate hearing, cognitive or memory function. These studies tacitly assume a transcranial mechanism of action. In light of our findings, a transcutaneous mechanism merits consideration. In our rat and healthy volunteer EEG experiments, we showed that transcutaneous stimulation entrains neural oscillations in the motor cortex. Evidence shows that oscillations induced in one functionally discreet brain area can influence oscillations in another functionally discreet part of the brain. For example, its common knowledge that rhythmic stimulation of the auditory nerve in some human subjects (i.e. listening to music) causes spontaneous neural entrainment in the motor system (i.e. dancing)[57–59]. Thus, it is possible that an oscillation induced in the somatosensory or motor cortex, via transcutaneous stimulation of peripheral nerves in the skin, influences brain oscillations in other functionally discreet brain networks.

On the other hand, data from our rat experiment also show that weak electric fields of around 1 V/m can cause small but significant amounts of entrainment in cortical neurons (see Fig. 2, transcranial-only data). Thus, it may be that tACS effects on non-motor systems are, at least partly, mediated by the transcranial mechanism. Future studies examining tACS effects on hearing, cognitive and memory function should now include controls, similar to those used in Experiments 2 and 3, to test which mechanism, transcranial or transcutaneous, is causing the effect. We would also caution that it is not immediately clear that small amounts of neural entrainment will translate into a measurable behavioral effect. We have previously shown that transcranial fields of around 8 V/m were needed to cause a measureable effect on limb movements in an anesthetized rat[38]. Lastly, we point out that the 0.9 V/m needed to cause neural entrainment in the rat experiments was measured at a depth of around 1 to 2 mm from the cortical surface. The electric field strength on the cortical surface (i.e. the value usually reported in human in vivo measurement or modeling studies) of the rat brain will be significantly higher than 1 V/m.

tACS is often applied using large saline-soaked sponge electrodes in an unfocused montage, giving a spatially broad current flow pattern. These unfocused montages, particularly those with non-cephalic return electrodes[43,44], do have the advantage that they draw more current into the brain. However, there are confounding drawbacks: a larger effect may be explained by a larger area of skinned being stimulated; furthermore with an unfocused tACS montage subjects often experience visual phosphenes. We

have previously shown that a focused montage eliminates tACS-induced phosphenes, most likely by limiting the current reaching the optic nerve[60]. Rhythmic stimulation of the optic nerve with light is known to cause both tremor entrainment[43] and to increase EEG power at the stimulation frequency[9]. Therefore, tACS-induced phosphenes (which are likely caused by electrical stimulation of the optic nerve[47,61], although this was debated[62]) are also very likely to cause both tremor entrainment and increase EEG power at the stimulation frequency. Therefore, for the current study, it was important to use a focused montage to isolate the putative transcutaneous and transcranial mechanisms, from any other potential tACS mechanisms such as electrical stimulation of the optic nerve.

Our results indicate that the transcutaneous mechanism is the dominant mechanism driving tACS motor system effects. However, we did not completely rule out the possibility that both the transcutaneous and the transcranial mechanisms contribute simultaneously. In fact, results from Experiment 1 indicated that electric fields of around 1 V/m do cause weak neural entrainment in some neurons. Also, in Experiment 2A, in the anesthesia session, there was a non-significant trend for more tremor entrainment with increasing amplitude. Since the peripheral nerves were anesthetized during these conditions, it is possible that this trend was caused by transcranial stimulation of cortical neurons. However, we believe this to be unlikely for three reasons: firstly, invasive measurements in epilepsy patients have already shown that the electric fields generated in the cortex with this type of tACS are not strong enough to cause neural entrainment in humans[33]. Secondly, in Experiment 2A, most subjects could still feel the stimulation in the anesthesia session; particularly in the HIGH amplitude condition (see Fig. 5). Although, importantly, they did rate the sensation as significantly lower compared to the no anesthesia session. In other words, the topical anesthesia reduced the contribution from the transcutaneous mechanisms but did not completely block it. Thus, the trend toward increased tremor entrainment in the anesthesia condition appears more likely to have been caused by transcutaneous stimulation of peripheral nerves which were not fully blocked by the topical anesthesia. Finally, in Experiment 2B, we used higher amplitude stimulation to increase the electric field strength in the cortex while simultaneously blocking the contribution from the peripheral nerves. With a stronger electric field in the cortex, we still did not observer tremor entrainment. Details discussing further limitations, the implications of our results for tDCS and for improving tACS are presented in a Supplementary Discussion.

Our results indicate that tACS motor system effects are similar to classical methods that use rhythmic stimulation of cranial[4,7–9] or peripheral nerves[14] to indirectly entrain neural oscillations. Importantly, results from our rat experiments did show that electric fields in the brain of around 1 V/m can cause some neural entrainment, but it is unclear if this would translate into a measureable behavioral effect in humans. It is also currently unknown what role, if any, the transcutaneous mechanism plays in mediating tACS effects on non-motor functions such as perception, cognition, memory, hearing and EEG. However, the methodology developed here (i.e. topical anesthetics and non-cephalic electrode montages) can now be used to separate the transcranial and transcutaneous mechanisms and independently test their contributions.

## Methods

**Human head computational model.** To obtain a quantitative estimate of electric field strength in the different tissue layers (skin, skull, cerebrospinal fluid (CSF), gray matter and white matter) during tACS, we implemented an electro-anatomical computational model. The anatomical model is based on modified data from the MIDA study: a publicly available homogenous head model, which was built by

combining different tissue classes of a multimodal imaging–based detailed anatomical (MIDA) model of human head and neck (FDA, Center for Devices and Radiological Health, MD,USA, and IT'IS Foundation, Zurich, Switzerland)[63]. The MIDA model was imported into ScanIP 7 (Simpleware Ltd., Exeter, UK) as a series of 116 surface meshes—each mesh representing a different tissue type. We first simplified the model by reducing it to tissue types relevant for this study and with known conductivity values. To do this, we converted the meshes to volumes (masks). Then we merged tissue volumes to obtain just four tissue types and assigned them the following standard electrical conductivity values ($\sigma$): skin 0.465 S/m; skull 0.01 S/m; CSF 1.65 S/m; gray matter 0.27 S/m; and white matter 0.126 S/m[60,64,65]. The gel-filled cup electrodes for the focused tACS montage used in Experiment 2 were modeled as a 1-mm thick layer with 0.3 S/m conductivity. To match the setup used in Experiment 2, each modeled electrode had a diameter of 2 cm, with the central stimulating electrode (anodic) located at C3 (over the motor cortex) and the return electrodes (cathodic) at C1, C5, CP3 and FC3 (20–10 EEG system nomenclature). In ScanIP 7, a volumetric tetrahedral model was then generated and imported into COMSOL multiphysics 5.3 (COMSOL, Inc., Burlington, MA) where electric field ($E$) and current density ($J$) were calculated by solving Laplace's equation,

$$\nabla \cdot \sigma \nabla \varphi = 0 \quad E = |\nabla \varphi| \quad J = \sigma |\mathbf{E}|, \tag{1}$$

with $\varphi$ representing the electrical potential. This assumes a quasi-static approximation of Maxwell's equations, valid for alternating electric fields in the brain with frequencies <1 MHz[66]. Boundary conditions were set to have a positive current at the anodic central electrode with peak-amplitude equal to 1 mA and the negative current divided equally between the four cathodic return electrodes.

This model was used to calculate the data shown in Fig. 1 and to make the estimates of the electric field strength in the brain and skin for the different stimulation amplitudes used in Experiment 2 and reported in the Results and Supplementary Tables. The electric field strength estimations presented in the supplementary tables are based on the average value of the electric field strength in a 10 mm³ volume containing the highest electric field strengths in one particular tissue. The 10 mm³ volume was found by first ranking all voxels in one particular tissue from high to low and then selecting the number of voxels, starting with the highest ranking and progressing to lower, which were needed to make up a 10 mm³ volume. This procedure is better than simply taking the highest value in one voxel, which can be skewed to very high values. When comparing between tissues with different volumes, this fixed-volume metric gives a fairer comparison than calculating the average electric field in a percentage of the tissue volume.

**Experiment 1—animals.** Seven male Wistar rats (305–594 g, Janvier labs, France) were used. They were housed in a rat colony at-19 °C and maintained on a 14/10 h light/dark cycle (lights on at 7:00 A.M.). Rats had unrestricted access to food and water. All procedures were approved by the KU Leuven ethics committee for laboratory experimentation (project P096/2015).

**Experiment 1—surgery and preparation.** On experiment days, the rats were anesthetized with an i.p. injection of a combination of ketamine (45 mg/kg, Anestekin, Eurovet, Belgium) and medetomidine HCl (0.3 mg/kg, Narcostart, Kela Veterinaria, Belgium), placed in a stereotaxic frame (Narishige type SR-6, No. 7905) on a heating pad and the core temperature monitored via a metal rectal probe. Anesthesia level was routinely monitored using the toe-pinch reflex. The anesthesia level was held constant by giving an additional i.p. injection of around 100 μL of the ketamine–medetomidine mixture approximately every hour.

The skull was exposed by making an incision in the scalp and then retracting the scalp tissue to expose an area measuring approximately 2 cm (medial–lateral) by 3 cm (anterior–posterior) and centered around Bregma. Upon exposure of the skull, the motor cortex in relation to Bregma was located using the Paxinos and Watson rat brain atlas[67]. A burr hole craniotomy was drilled using the US#4 HP 014 drill bit (Meissinger, Germany) in the skull at a location chosen to target the fore-limb area (right or left hemisphere for every experiment) of the motor cortex[37]. Average coordinates of the burr hole relative to Bregma were: ML ± 2 mm, AP +2 mm. The dura was left intact. Two bone anchor screws (Cat. No. 51457, Stoelting, USA) were driven approximately half-way through the skull and served as electrodes for the transcranial-only stimulation. One screw was positioned 2 mm anterior of the burr hole and connected to the positive terminal of the stimulating current source (details below) while the other screw was positioned 2 mm posterior of the burr hole and was connected to the negative terminal. These screws were used to deliver the transcranial-only stimulation. During transcranial-only the skull was always dry. To deliver the transcutaneous-only stimulation, one crocodile clip was coated in electrode gel (Signa Gel, Parker Labs, New Jersey) and clipped onto a fold of skin on one of the four limbs: contralateral (to the recording electrode, see below) hind, contralateral fore, ipsilateral hind or ipsilateral fore. No hair was removed. A second crocodile clip was always attached to the metal rectal temperature probe. The limb crocodile clip was connected to the negative terminal of the current source and the rectal probe clip was connected to the positive terminal.

At the end of the experiment, the rats received a 200-μL injection of the local anesthetic Xylocaine 2% (lidocaine HCl 20 mg/mL, Astrazeneca, UK) in relevant areas. Furthermore rats received an injection of the general analgetic Metacam (5 mg/mL, Boehringer Ingelheim). The surgery wound was treated with the antibiotic cream Fucidin (sodium fusidate 20 mg/g, Leo Pharma A/S, Denmark). The wound was sutured and rats were allowed to wake from anesthesia and recover for at least 5 days. Rats underwent between one and three experiments.

**Experiment 1—electrical stimulation setup.** Electrical stimulation, always in the form of a sinewave, was delivered with an AM 2200 analog current source (AM Systems, Sequim, WA) connected to either two stimulation screws in the skull for transcranial-only stimulation or to the two crocodile clips for transcutaneous-only stimulation. The current source was controlled by an analog voltage waveform generated using an output channel on a data acquisition card (NI USB-6216, National Instruments, Austin, TX) at a sample rate of 30 kHz. The acquisition card was controlled using a custom written MATLAB (Mathworks, Natwick, MA) software.

**Experiment 1—electrophysiological recording setup.** A 32-channel silicon probe (50 μm thick, E32Tri+R-25-S01-L10 NT, Atlas Neuro, Leuven, Belgium) with a pointy tip was used to record neurophysiological activity in the motor cortex. Electrode contacts were made of iridium oxide and had a diameter of 15 μm with an average impedance of 0.47 MΩ (SD 0.046 MΩ) as reported by the manufacturer. Electrode contacts were separated from neighboring contacts by 25 μm. The electrode contacts were arranged in three columns and spanned 275 μm in the vertical dimension (Fig. 2, far left panel). Probe output was amplified (×200), bandpass filtered (0.1 Hz to 5 kHz) and digitized (16 bit, 30 kHz) using an Open Ephys headstage and acquisition board (www.open-ephys.org/). During the experiment, data from the probe were visualized and recorded on a PC hard drive (30 kHz sampling rate) using the Open Ephys GUI[68].

The probe was lowered into the burr hole craniotomy and pierced the dura mater to access the cortex.

Once in the cortex, the probe was advanced until clear rhythmic "bursting" spiking activity was observed. Recording depths, measured from the dura surface, ranged from 850 to 2300 μm. We estimate that this range covers layers IV, V and VI of the rat motor cortex[69]. During the experiment, neural response data to transcranial-only and/or transcutaneous-only stimulation were collected at one depth before advancing the probe to another location, at least 300 μm deeper, to collect more data.

**Experiment 1—procedure.** During the experiments, the same current source was used for both transcranial-only and transcutaneous-only stimulation by switching the connections to either the screws in the skull or the crocodile clips on the body. There was never simultaneous delivery of transcranial and transcutaneous stimulation. The stimulus waveform was always a sinewave. Transcranial-only stimulation was applied at levels of between 0.025 and 0.5 mA. During the transcranial-only condition a stimulation artifact could be observed, caused by volume conduction of the electric field at the stimulation screws to cortex in the vicinity of the recording electrodes. This artifact did not saturate the headstage amplifiers, which have a maximum input range of ± 5 mV. Transcutaneous-only sinewave stimulation was applied at four different sites (see above) at 1, 2 or 3 mA. No stimulation artifact was observed during transcutaneous-only stimulation. During the experiment, MUA was displayed by band pass filtering the recorded data between 300 and 3000 Hz. The bursting frequency was then calculated as the dominant period (time from central peak to second largest peak) of the autocorrelation function of the MUA spike times. For both transcranial-only and transcutaneous-only, the stimulation frequency was between 1 and 2.5 Hz, chosen to match the frequency of the ongoing bursting activity. For both transcranial-only and transcutaneous-only, one recording always consisted of 1 min of no stimulation (OFF condition) followed immediately by 1 min of stimulation (ON condition) at a given amplitude and frequency. These 2-min recordings (1 min stimulation OFF, 1 min stimulation ON) were used for all further post-experiment analyses.

**Experiment 1—post-experiment data analysis.** After the experiment, the first data analysis step was spike sorting which was performed using the Klusta Suite[40]. The Klusta Suite took in the raw 32-channel recordings and via a series of automatic steps clustered MUA into putative single-unit clusters. An important step in the Klusta Suite is the band pass filtering (300 to 3000 Hz) of all data which removed any stimulus related artifact from the transcranial-only recordings. After the automatic spike sorting, we then manually inspected and corrected clustering where necessary. Only well isolated single-unit clusters were used for further analysis.

The following analysis steps were then completed for each identified single-unit. First, based on stimulus trigger, spike times were separated into 1 min OFF and 1 min ON conditions. For each condition, we created a cycle time histogram with 30 bins per cycle and one cycle being equal to one stimulus (transcranial or transcutaneous) period. Similar to Ozen et al.[29], cycle histograms were created by binning spikes based on the stimulus phase at which they occurred and then normalizing across the entire condition to give spike rate per phase bin. During the OFF condition, there was no stimulus present. Therefore, we constructed cycle histograms for the OFF condition using the same stimulus period from the ON

condition. This is equivalent to assuming that a sine wave at the stimulus frequency is present during the OFF condition but that it does not stimulate the neurons. This is similar to the simulated sine wave used for the OFF conditions in tremor Experiments 2 and 3. These cycle histograms are similar to a post stimulus time histograms but take into account the ongoing nature of the sinewave stimulus. Figures 2 and 3 show ON and OFF condition cycle histograms, which visualize neuronal entrainment as an increase in spike rate at one particular stimulus phase. To quantify each cycle histogram, we calculated the PLV using the following equation:

$$\mathrm{PLV} = \left| \sum R_{\mathrm{b}} e^{i\theta_{\mathrm{b}}} \right|, \qquad (2)$$

where $\theta_{\mathrm{b}}$ is the center of bin "b" and derived from the phase of the sinewave cycle; and $R_{\mathrm{b}}$ is the magnitude of bin "b". For the PLV calculation, $R_{\mathrm{b}}$ was the probability of a spike occurring in that particular bin. This definition means that if all spikes occur within one phase bin PLV will be 1 and when spikes occur equally in all phase bins PLV will be 0. Defined as such, the PLV is a good mathematical description of the term neural entrainment, which we define as timing or phase of a neural signal being adjusted to become synchronized to an external stimulus.

Some neurons are known to entrain at the double of the tACS frequency[29]. To check for the preferred entrainment frequency, we calculated cycle histograms and PLVs for all single-units at both the stimulation and double the stimulation frequency. We then selected the frequency (either equal to or double the stimulation frequency) that showed the highest PLV during the ON condition for all further analysis.

**Experiment 1—electric field strength calculation.** To calculate the electric field strength during transcranial-only stimulation, we Fourier transformed the unfiltered signal recorded on each of the 32 electrodes during the 1 min stimulation ON condition (see Supplementary Figure 1). We extracted the amplitude of the component at the stimulation frequency for each electrode. Using this measurement, we calculated spatial gradient of the voltage along each of the three electrode columns to give an estimate of the electric field strength in the dorso-ventral direction (i.e. the direction of the electrode penetration). We then calculated the mean of the voltage gradient across all three electrode columns to give one electric field strength value for each recording site in each rat tested. Measurements and calculations were done across the full range of transcranial-only current stimulation amplitudes (0.05 to 0.5 mA).

During the transcutaneous-only stimulation conditions, we needed to be sure that no current from the limb was reaching the brain due to volume conduction. Therefore, we calculated the electric field strength in the brain during all the transcutaneous-only stimulation conditions using exactly the same analysis of the unfiltered signals recorded on the 32 electrode during the 1 min of transcutaneous-only stimulation (see Supplementary Figure 2).

**Experiment 1—statistical analysis.** The transcranial-only data were collected at a range of amplitudes between 0.025 and 0.5 mA, while the transcutaneous-only data was collected at three amplitudes (1, 2 and 3 mA). To facilitate the application of the same statistical analysis on the transcranial-only and the transcutaneous-only data, we grouped the transcranial-only data into three intensity ranges, namely: ≤0.1 mA; >0.1 mA but ≤0.2 mA; and >0.2 mA. We tested for normality using the one sample Kolmogorov–Smirnov test and found that the PLV data were not normally distributed. Therefore, we used non-parametric statistics. For transcranial-only stimulation, we used a one-sided Wilcoxon signed rank test to compare PLVs during the OFF and ON conditions and determine if transcranial-only stimulation caused neural entrainment. Since there were three amplitude groups, we corrected for multiple testing using the Bonferroni correction with $n = 3$. For the transcutaneous-only stimulation, we had three amplitude groups and four different stimulation configurations (contra-hind, contra-fore, ipsi-hind and ipsi-fore). Again, we used a one-sided Wilcoxon signed rank test to compare PLVs during the OFF and ON conditions and determine if transcutaneous-only stimulation caused neural entrainment. Since there were 12 groups, we corrected for multiple testing using the Bonferroni correction with $n = 12$. Corrected $p$-values were always reported. All corrected $p$-values larger than 1 were reported as $p = 1$. The same statistical analyses were performed to test if transcranial-only and transcutaneous-only stimulation had an effect on spike rate.

Next, we wanted to test if changing the amplitude of the transcranial-only stimulation had an effect on neural entrainment and also to test if changing the amplitude or the electrode configuration of the transcutaneous-only stimulation had an effect on neural entrainment. To do this, we first calculated the change in neural entrainment between the OFF and the ON conditions for each single-unit by taking the difference in the PLV values to give $\mathrm{PLV_{dif}}$. For the transcranial-only data, we then fit a linear model where we used stimulation amplitude to predict the $\mathrm{PLV_{dif}}$ (MATLAB, fitlme.m, model notation: $\mathrm{PLV_{dif}} \sim 1 + \mathrm{Amplitude}$, only fixed effects). For the transcutaneous-only data, we fit a linear model where we used stimulation amplitude and electrode configuration to predict $\mathrm{PLV_{dif}}$ (MATLAB, fitlme.m, model notation: $\mathrm{PLV_{dif}} \sim 1 + \mathrm{Amplitude} + \mathrm{Electrode\text{-}Configuration} + \mathrm{Amplitude}*\mathrm{Electrode\text{-}Configuration}$, only fixed effects). The same statistical analyses were also performed to test if stimulation amplitude or electrode

configuration had an effect on spike rate. Post-hoc comparisons were performed using the Wilcoxon sum rank tests (one-sided for the amplitude comparisons and two-sided for the configuration comparisons) and were Bonferroni corrected for multiple comparisons. Only relevant comparisons were performed, which are described in full in the Results.

All statistical tests were performed in MATLAB with a significance level of $\alpha = 0.05$.

**Experiment 2A—subjects.** Twelve healthy volunteers participated in Experiment 2 (three females; age $27 \pm 4$). A power analysis (power = 0.80, $\alpha = 0.5$, one-tailed), based on effect sizes estimated from data in our previous study[44] showed that a sample size of 11 would be sufficient to detect significant effects. Ten were right-handed and two were left-handed (self-reported). The experiments were approved by the Medical Ethics Committee at UZ/KU Leuven (S57869) and were performed in accordance with the relevant guidelines and the Declaration of Helsinki. Written informed consent to participate in the experiment was obtained from all healthy volunteers.

**Experiment 2A—anesthesia.** The experiment (described below in the sections: Electrode placement, Tremor measurements, Stimulation amplitude, threshold and sensation measurement and Experimental protocol) was repeated on each subject on two different days. On 1 day, topical scalp anesthesia was applied to the tACS electrode site (anesthesia condition); while on the other day, saline solution was applied to the same site (no anesthesia condition). The stimulated site (both for the anesthesia and no anesthesia condition) was always the motor cortex contralateral to the self-reported dominant hand. On both days, tACS electrodes were also attached to the ipsilateral motor cortex, always without anesthesia, to serve as a reference for the stimulation sensation intensity ratings (see below).

The subjects were informed that anesthesia would be applied on 1 day and that a placebo would be used on the other day but were blinded as to which condition was tested on which day. For blinding purposes, researcher BA was not present when researcher AK applied the anesthesia or saline to the subject. For the anesthesia condition, 10 g of topical anesthetic EMLA cream (5%, AstraZeneca, Belgium) was applied to the scalp over the contralateral motor cortex, where the stimulating tACS electrodes would be placed. For the no anesthesia condition, saline solution was applied to the same site. On both days, we waited 30 min after application before removing any excess cream and drying the hair with a towel and hair dryer. On both days, the next experimental stages (described below) were identical.

**Experiment 2A—electrode placement.** A set of custom made $4 \times 1$ gel-filled cup-electrodes were used to create a focused electric field over the motor cortex contralateral to the self-reported dominant hand. The focused tACS montage, experimental setup and analysis have previously been reported in detail[44]. Cup-electrodes had a 5 mL volume and were constructed from 2 cm diameter plastic cylinders mounted in an EEG cap (EASYCAP GmbH, Germany). The cup-electrodes were filled with 5 mL of electrode gel (Signa Gel, Parker Labs, New Jersey) before an EEG AgCl ring electrode (EASYCAP GmbH, Germany) was fastened into the cup. For right-handed subjects, the electrode montage was positioned to target the left motor cortex: a central stimulating electrode was located at C3 with the four return electrodes at C1, C5, CP3 and FC3 (20-10 EEG system nomenclature). For the left-handed subjects, the right motor cortex was targeted: the stimulating electrode was placed at C4 and the return electrodes were placed at C2, C6, CP4 and FC4. In all subjects, a similar focused tACS montage was placed over the motor cortex ipsilateral to the dominant hand. This montage was only used in the stimulation sensation intensity ratings (see below) and was not stimulated during the tremor measurement stage.

A stimulation OFF condition was used to serve as a control. During the OFF condition, stimulation was still delivered but the electrodes were not connected to the subject. Instead, the stimulator was connected to two ring electrodes which were placed several centimeters apart in a beaker filled with electrode gel (Signa Gel, Parker Labs, New Jersey). Designing the OFF condition in this way is better than simply not delivering any stimulation as it allows us to use a real stimulation waveform, as opposed to a simulated one, to calculate tremor entrainment during the OFF condition (see below).

Electrodes were connected to a DS5 current source stimulator (Digitimer, Hertfordshire, UK) driven by a voltage waveform generated on a data acquisition card (4096 Hz, NI USB-6216, National Instruments, Austin, TX) and controlled via custom written MATLAB R2014a software (Mathworks, Natwick, MA).

**Experiment 2A—tremor measurements.** Subjects were seated on a chair and instructed to rest the wrist of their dominant hand on a table with fingers extended. A 15 g weight was attached to the middle finger. This induced a measurable postural enhanced physiological tremor in all subjects. A triaxial accelerometer (ADXL335, Analog Devices, Norwood, MA) was attached to the middle finger. Accelerometer data were digitized (4096 Hz) on the aforementioned data acquisition card, and stored for offline analysis. Using custom written MATLAB software, each accelerometer axis was bandpass filtered (3–30 Hz) using a second-order Butterworth filter. Principal component analysis was applied, and the first

component extracted. Our tremor measurement and analysis procedures have been described in detail in a previous publication[44].

**Experiment 2A—stimulation amplitude, threshold and sensation.** To measure tACS sensation threshold, stimulation amplitude was increased from 0 mA in 0.1 mA steps on the tACS montage over the contralateral motor cortex. After each increase, the subject was asked if they perceived the stimulation. Once they reported that they perceived stimulation, amplitude was slowly decreased until the sensation disappeared, and then increased again until sensation were just perceivable. This was noted as the sensation threshold.

In the Experimental Protocol (see below), three amplitude conditions were tested: OFF (electrodes not connected to subject), LOW amplitude and HIGH amplitude. To determine the HIGH amplitude, tACS was slowly increased in 0.2 mA steps toward a maximum peak-amplitude of 2.5 mA (i.e. 5 mA peak-to-peak). Subjects were instructed that they would feel the stimulation but it should not be uncomfortable. If subjects stated that tACS was uncomfortable, the amplitude was reduced in steps of 0.1 mA to a comfortable level. This was then used as the HIGH amplitude value. If they did not find stimulation uncomfortable, amplitude was increased to a maximum of 2.5 mA and this was used as the HIGH amplitude value. The LOW amplitude was defined as one-fifth of the HIGH amplitude value (i.e. LOW amplitude = 0.2 × HIGH amplitude) in all subjects except the first three subjects where the LOW amplitude was defined as half of the HIGH amplitude value (i.e. LOW amplitude = 0.5 × HIGH amplitude). To make the sensation intensity measurements, the subject was asked to rate the stimulation sensation intensity using a visual analog scale (VAS) from 0 to 10 for each condition: OFF, LOW and HIGH. This was repeated for both the contralateral and ipsilateral motor cortex tACS montages. Stimulation to control tremor, described in the Experimental Protocol, was always done on the contralateral motor cortex tACS montage. We obtained sensation ratings on the ipsilateral tACS montage to help serve as a reference point for the subject during the 2 days—1 day with anesthesia on the contralateral motor cortex and 1 day without.

To determine the amplitude, threshold and sensation levels the researcher who applied the anesthesia or saline also operated the tACS equipment (AK). Another researcher (BA), who was blinded to the anesthesia condition and was not informed of the specific tACS amplitude being delivered, questioned the subject to determine the amplitude, threshold and sensation levels.

**Experiment 2A—protocol.** Before stimulation, a 5-min period of tremor was recorded to extract individual tremor frequency by calculating the maximum power spectral density averaged over time. For all subjects, tACS was a sinewave with a frequency equal to the individual's tremor frequency. Electrode impedance was below 5 kΩ. Each subject completed three 12-min sessions with a 10-min break between each session. During each session, the subject was instructed to keep their hand in the tremor inducing posture and accelerometer data were continuously recorded. In the first session, the conditions were ordered as such: 60 s LOW – 15 s no stimulation – 30 s OFF – 15 s no stimulation – 60 s HIGH – 15 s no stimulation – 30 s OFF – 15 s no stimulation. This 4 minute sequence was repeated three times to give one 12-min session containing 3 min of each condition (OFF, LOW and HIGH). For half of the subjects (randomly assigned) the order of LOW and HIGH conditions were switched in the first session. The second session was the same as the first session but order of the LOW and HIGH conditions were switched. The third session had the same order as the first session.

**Experiment 2A—tremor data analysis.** The procedures used to calculate tremor entrainment are the same as in our previous publication[44]. Briefly, we performed a Hilbert transform on the tACS and tremor waveforms to extract instantaneous phase. The phase signals were subtracted and a 30 bin phase-difference histogram constructed. This was normalized by the total number of samples, yielding a phase-difference probability histogram (Fig. 4, top panels). If there is no entrainment, phase-differences are uniformly distributed. If there is perfect entrainment (i.e. constant phase-difference) all the phase-differences appear in one bin. To quantify this. we used Eq. (2), the same equation used in Experiment 1, to calculate the PLV. The difference in Experiment 2 is that a histogram now represents the likelihood that a particular tACS-tremor phase-difference occurs in relation to stimulus phase, as opposed to a particular spike-rate as in Experiment 1. As such, PLVs in Experiment 2 (and in Experiment 3) quantify the amount of tremor entrainment and not neural entrainment as is the case in Experiment 1. It is helpful to use the same equations and metrics in all experiments to facilitate comparisons about the entrainment levels. However, it is important to be aware of the slightly different meaning of PLV in the rat and human experiments. Thus, analogous to the rat experiments, PLV = 0 indicates no tremor entrainment (a uniform histogram) and PLV = 1 indicates complete tremor entrainment (all phase-differences in one bin). If tACS entrains tremor, PLV during LOW amplitude or HIGH amplitude stimulation will be greater than PLV during the OFF condition.

**Experiment 2A—statistics.** We tested for normality using the one sample Kolmogorov–Smirnov test and found that the PLV data were not normally distributed. Therefore, we used non-parametric statistics. We used a linear mixed model to test the effect of stimulation amplitude and anesthesia on tremor

entrainment. Specifically, we fit a linear mixed model to the PLV data to test if it could be predicted by the amplitude and anesthesia conditions. We set the amplitude and anesthesia conditions and their interaction as fixed effects and set the subject as a random effect (MATLAB, fitlme.m, model notation: PLV ~ Amplitude + Anesthesia + Amplitude × Anesthesia + (1|Subject)). Post-hoc testing was done using one-sided Wilcoxon signed rank test with the Bonferroni correction for multiple comparisons. Nine relevant comparisons which are described in full in the Results, were performed. Corrected p-values are reported. A Grubbs test was used to check for outliers in the data. No outliers were detected.

We used a Kolmogorov–Smirnov test to check if the sensation intensity rating data were normally distributed. The same linear mixed model was used to test effect of stimulation amplitude and anesthesia on the sensation intensity rating (MATLAB, fitlme.m, Sensation Intensity ~ Amplitude + Anesthesia + Amplitude × Anesthesia + (1|Subject_nb)). Post-hoc testing was the same as above. To test for a difference between the sensation thresholds in the anesthesia and no anesthesia conditions, a one-sided Wilcoxon signed rank test was used.

We calculated effect sizes using the Cohen's d formula which is defined as the difference in the means of two groups divided by the standard deviation[70]. For paired data, the standard deviation was calculated from the distribution of the difference in scores between the two groups. An effect size of >0.8 is considered large.

All statistical tests were performed in MATLAB with a significance level of $\alpha = 0.05$.

**Experiment 2B—subjects.** Twelve ET patients participated in this experiment (seven females; age 71 ± 9). All were right handed (self- reported). All participants had advanced ET and had DBS electrodes implanted in the ventral intermediate nucleus of the thalamus to reduce their tremor. However, all patients still exhibited significant tremor when their DBS device was on and had large uncomfortable tremor when their DBS device was off. Therefore, for all patients, the DBS device was kept switched on in normal settings during the whole experiment. There is a theoretical possibility that some current from the tACS device could either interfere with the DBS device or be conducted by the DBS electrodes to a deep brain location. Therefore, we first performed extensive bench top testing with a DBS device and electrode array and our tACS system to investigate when, if ever, this could occur. We could not find any realistic conditions under which interference between the two systems could occur. Even if one of the DBS electrodes was broken or shorted (which in theory may increase the likelihood of interference), we did not measure interference. This report was submitted to our ethics committee who approved the study. Nevertheless, as a safety measure, DBS electrode impedance and system integrity were checked before and after the experiment. No issues or cases of interference were found in any of the 12 ET patients.

The experiments were approved by the Medical Ethics Committee at UZ/KU Leuven (S57869) and were performed in accordance with the relevant guidelines and the Declaration of Helsinki. Written informed consent to participate in the experiment was obtained from all ET patients.

**Experiment 2B—anesthesia.** Six patients were selected randomly and anesthetic cream was applied under the tACS electrodes in a similar way to Experiment 2A. No anesthesia was applied to the other six patients.

**Experiment 2B—electrode placement.** Similar to Experiment 2A, five tACS electrodes in a focused montage were placed over the motor cortex contralateral to the arm on which tremor was measured.

**Experiment 2B—tremor measurements.** Patients were seated on a chair and asked to position their arm in a tremor inducing posture such holding their arm stretched out in front of them. They were asked to maintain their arm in that position during the 1 or 2 min recording sessions (described below). The tremor was measured using an accelerometer in a similar way to Experiment 2A.

**Experiment 2B—stimulation amplitude, threshold and sensation.** For the group with no anesthesia, the stimulation amplitude was set to maximum of 2 mA. To increase the electric field strength in the brain, and thus increase the possibility of observing a true transcranial effect, the stimulation amplitude was set to a maximum of 5 mA in the group with anesthesia. The same procedure, as used in Experiment 2A, for increasing the stimulation amplitude to a comfortable level or stopping at the maximum 5 mA was used here. This meant that some subjects were stimulated at amplitudes below the maximum. Full details are provided in Supplementary Table 5.

**Experiment 2B—protocol.** Before stimulation, a 2-min period of tremor was recorded to extract individual tremor frequency by calculating the maximum power spectral density averaged over time. For all subjects, tACS was a sinewave with a frequency equal to the individual's tremor frequency. Electrode impedance was below 5 kΩ. Each patient completed fifteen 1-min recordings with a 2-min break between each of them. In total, this formed five sessions where each session consisted of one OFF and one ON stimulation delivered in random order.

**Experiment 2B—tremor data analysis**. Data analysis was similar to Experiment 2A. However, the PLV was calculated for stimulation OFF and stimulation ON as we only used one stimulation amplitude per patient (not two as in Experiment 2A).

**Experiment 2B—statistics**. For each of the two groups (anesthesia and no anesthesia), a one-sided Wilcoxon signed rank test was used to test for significant increase from the OFF to the ON condition. All statistical tests were performed in MATLAB with a significance level of $\alpha = 0.05$.

**Experiment 3—subjects**. Twelve healthy volunteers participated in the Experiment 3 (five females; average $27 \pm 5$ years old). Eleven were right handed and one was left handed (self-reported). A power analysis (power = 0.80, $\alpha = 0.5$, one-tailed), based on effect sizes estimated from data in our previous study[44] showed that a sample size of 9 would be sufficient to detect significant effects. The experiments were approved by the Medical Ethics Committee at UZ/KU Leuven Research (S57869) and were performed in accordance with the relevant guidelines and the Declaration of Helsinki. Written informed consent to participate in the experiment was obtained from all healthy volunteers.

**Experiment 3—electrode placement**. The same gel-filled cup electrodes used in Experiment 2, were also used in Experiment 3. However, since the electrodes were located on the arm, it was not necessary to have a focused montage to avoid retinal stimulation. Therefore, two gel-filled cup electrodes were positioned 8 cm apart on the skin of the upper arm, over the upper portion of the brachialis muscle. The electrodes were positioned on the arm contralateral to the hand on which tremor was measured. As in Experiment 2, two electrodes were placed in a gel-filled beaker for stimulation during the OFF condition.

**Experiment 3—tremor measurements**. Same as Experiment 2A.

**Experiment 3—stimulation amplitude, threshold and sensation**. Same as Experiment 2A.

**Experiment 3—protocol**. Same as Experiment 2A.

**Experiment 3—tremor data analysis**. Same as Experiment 2A.

**Experiment 3—statistics**. The statistics were similar to Experiment 2A. However, the linear mixed model now tested the effect of just the stimulation amplitude (OFF, LOW and HIGH) on tremor entrainment (MATLAB, fitlme.m, model notation: PLV ~ Amplitude + (1|Subject)). Post-hoc testing was done using Wilcoxon signed rank test with the Bonferroni correction for multiple comparisons ($n = 3$). To test for an effect of stimulation amplitude on sensation intensity rating, a linear mixed model was used (MATLAB, fitlme.m, model notation: Sensation Intensity ~ Amplitude + (1|Subject)). Post-hoc testing was the same as above. All statistical tests were performed in MATLAB with a significance level of $\alpha = 0.05$.

**Experiment 4—subjects**. Twelve healthy volunteers participated in Experiment 4 (3 female, average $27 \pm 4$). The experiments were approved by the Medical Ethics Committee at UZ/KU Leuven (S57869) and were performed in accordance with the relevant guidelines and the Declaration of Helsinki. Written informed consent to participate in the experiment was obtained from all healthy volunteers.

**Experiment 4—electrode placement**. Stimulation electrodes were placed over the right arm of all subjects in the same position as in Experiment 3.

**Experiment 4—EEG measurements**. Subjects were seated on a chair and instructed to rest and not to move during the recording sessions.

Single channel EEG from the motor-somatosensory cortex was recorded by placing one electrode at FC3 and the other (reference) electrode at CP3. The ground was connected to the left mastoid. Electrode impedance was below 5 kΩ. The electrodes were connected to a differential amplifier (Stanford Research Systems Model SR560) and the data were digitized (4096 Hz) on the aforementioned data acquisition card, and stored for offline analysis. The amplifier settings were set as following: gain—10,000; band pass filter—0.3 to 100 Hz; low noise setting.

**Experiment 4—stimulation amplitude and sensation**. In contrast to the previous experiments, a single biphasic pulse (0.44 ms per phase), repeated at stimulation frequency (i.e. the individual's beta frequency), was used for the stimulation. Pilot experiments using sinewave stimulation showed that, even with the stimulating electrodes placed on the arm, a small sinewave artifact was still present in the EGG signal. It was impossible to completely separate this artifact from the true EEG signal. A single pulse has the advantage that the artifact is limited in time, and due to the square waveform of the pulse, is located at a much higher frequency band than the EEG signal of interest.

We wanted to deliver the pulsed stimulation at a level that gave a similar sensation intensity to sinewave stimulation. Therefore, each subject was first asked to rate the sensation intensity when a 2.5 mA sinewave stimulation (same as in experiments 2 A and 3) amplitude was applied. The waveform was then switched to single pulses, repeated at the same frequency, and the stimulation amplitude was increased in 0.5 mA steps until the subject gave a similar sensation intensity rating as in the case of the 2.5 mA sinewave. This amplitude was defined as HIGH and half of it was defined as LOW (i.e. LOW amplitude = 0.5 × HIGH amplitude). Each subject was also asked to give a sensation rating to the LOW amplitude stimulation. Due to the low amount of electric charge delivered during the pulse stimulation in comparison to the sinewave stimulation, a higher stimulation amplitude was required to reach similar sensation rates.

**Experiment 4—protocol**. Before stimulation, a 3-min period of EEG without stimulation was recorded. This was analyzed to extract the individual peak in the EEG beta activity by calculating the power spectral density for 1 s epochs and averaging this over the entire 3 min recording (Supplementary Figure 11). Pulsed stimulation was then delivered at the individual's peak beta frequency. Each subject completed fifteen 3 min recordings with a 1 min break between each of them. During the 15 sessions stimulation, amplitude was randomly cycled between either OFF, LOW or HIGH. This yielded a total of five repetitions for each condition: OFF, LOW and HIGH.

**Experiment 4—EEG data analysis**. The EEG signal was band pass filter between 0.3 and 100 Hz. This filtering step removed any small artifacts caused by the pulsed stimulation (Supplementary Figure 10). The procedure used to calculate EEG entrainment was then identical to that used to calculate tremor entrainment in Experiments 2 and 3. However, we used the EEG signal instead of the tremor signal. An additional difference was that the phase time-series for the pulsed stimulation was now calculated as a phase vector that went linearly from 0 to $2\pi$ between each stimulation pulse, creating a sawtooth function identical to the phase time-series of a sinewave. For the sinewave stimulation in Experiment 2 and 3, this stimulation phase time-series was calculated from the Hilbert transform.

To calculate the amplitude of the EEG signal at stimulation frequency during each of the three conditions (OFF, LOW and HIGH), the Fast Fourier Transform of the entire 3-min recording was performed (FFT.m, MATLAB) and the amplitude of the component at the stimulus frequency was then extracted.

**Experiment 4—statistics**. Same as Experiment 3.

## Data availability
All data and code are available upon reasonable request.

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

## Acknowledgements

This work was supported by KU Leuven Research Funding STG/14/024 and EGM-D2929-C24/17/091 and by an EIT Health Innovation by Ideas, NEURO-WEAR Project. Boateng Asamoah is SB PhD fellow at FWO.

## Author contributions

A.K. performed the computational modeling, the healthy volunteer and patient experiments and analysis. B.A. performed the rat experiments and analysis and helped with the healthy volunteer and patient experiments. M.M. formulated the hypotheses and designed the experiments. All authors contributed to manuscript writing and reviewing.

## Additional information

**Competing interests:** The authors declare no competing interests.

