## [Peer Review File · Nature Communications]

Reviewers' comments:

Reviewer #1 (Remarks to the Author):

tACS – transcranial or transcutaneous alternating current stimulation?
Boateng Asamoah, Ahmad Khatoun, Myles Mc Laughlin

This article comes with the aim of proposing a new mechanistic explanation for the results induced by transcranial alternating current stimulation (tACS) in brain research literature for sensory, cognitive and motor processes. The article is driven by a hypothesis built on a (largely opinion driven) literature survey, trying to show that external electrical stimulation of the peripheral nervous system can entrain central neuronal populations in a rhythmic fashion. The authors question the direct entrainment of central nervous system neurons by tACS, hypothesizing that transcutaneous stimulation can be accounted for the effects of tACS, given the magnitude of the generated electric field over the skin and skull compartments. This was tested in a series of experiments involving rodents and humans, with alternating currents applied to skull, scalp and arm of rats and healthy humans.

The aim of the study has high significance, nevertheless, with regard to the recent version I have the following concerns.

Overall I am sceptical concerning the use of physiological tremor measurement as a biomarker for the claim raised here. It would be much more straight forward to use methods to directly evaluate neuronal function, such as EEG or MEG, combined with the peripheral-only stimulation in experiment 3, to prove neural entrainment at the scalp level. Additionally, the terminology is problematic. I prefer a classification of physiological tremor confined to inevitable mechanical noise arising from remaining unsmoothed muscles twitches even if Hennemann's principle is accomplished optimally. What is used here is in my preferred terminology "enhanced physiological tremor" with spinal involvement.

In general, for such a strong substantial claim for tentatively criticising the whole field, better experimental evidence is needed. The animal and human part suffer from several pitfalls, which are not compensated by putting them together in a fused paper. Also the literature survey has to be put in a greater context. Data obtained by previous work on peripheral nerve stimulation in general and its impact on plasticity have to be quoted. Generally, I miss a thorough literature survey in which voltage gradients surpass the threshold excitability of peripheral neurons. This needs to be summarized in terms of V/m, (e.g. 3.8 to 5.8 V/m for the fields exceeded in 0.5% of tissue volume (skin and fat of the torso)) (So et. al, 2004). It remains unclear if the currents used in this manuscript are below or above peripheral stimulation thresholds.

Detailed comments:

The aim of experiment 1 was to prove that transcutaneous electrical stimulation can entrain neural activity as well as stimulation applied directly to the skull. My main concern here is that the transcutaneous stimulation intensities are much too high. The forelimb electrode is quite close to the brain. I would expect that a 1 mA stimulation at the forelimb generates far field potentials of 0.1 mA current flow in the brain of the rat.

Here the authors used 2 stimulation conditions called transcranial- and transcutaneous-only. They compared the recordings from 32-channel silicone probe placed over the corresponding motor area during 1 minute of ON stimulation versus previous 1 minute of OFF stimulation. They expected to find the effect of entrainment by increasing the phase locking value (PLV) in the ON condition in comparison with the OFF condition.

Throughout the article, PLV is a measure of synchrony between the applied sinusoidal wave and recorded neurophysiological activity. It is directly established by the stimulus onset, which could not be presented in the OFF condition. The authors wrote that the stimulus period from the ON condition was used for defining the stimulus period in the OFF condition. The procedure is not described in details, therefore it is difficult to evaluate how they compared the two conditions. This might be crucial because the presented PLV (Fig.2) for the transcranial-only OFF condition (0.14) is twice as high for the transcutaneous-only OFF condition (0.05). It would be reasonable to implement any baseline correction or to provide the absence of statistical differences for all OFF

conditions.

As the main conclusion of the Experiment 1 is that all stimulation conditions and all electrode configurations except of Contralateral-Hind and Ipsilateral-Hind resulted in significantly increased PLV, this may suggest the drawback of the proper control experiment or additionally that the PLV does not reflect neural entrainment.

Related to this point, the term 'neural entrainment' is not well defined. On the one hand, Vossen et. al, 2015 distinguished on-line effect which is factually "entrainment" and off-line effect which reflects rather plastic changes than entrainment. On the other hand, Hanslmayr et. al, 2014 reported that off-line effect was evident only after behaviorally relevant stimulation, resulting in memory impairment (in that case). They suggested that the evidence of entrainment ongoing oscillations is in the presence of after effect of stimulation. Therefore, I think it is necessary here to provide the evidence whether there were any differences between during and immediately after transcranial- and transcutaneous-only stimulation.

Furthermore, the conclusion of entrainment by one or another type of stimulation needs to be supported by a control condition, including two closely spaced electrodes that sufficiently stimulates skin receptors but inducing shunting in order to prevent brain stimulation. This control condition also applies for the human studies.

From the second and the third experiments, the authors reported that PLV was increased for most of the subjects in the no anesthesia condition while it decreased under anesthesia condition in the second experiment. Nevertheless, a closer look at the individual data (Supplementary fig. 2) shows that PLV changes between conditions are almost flat (S1, S4, S6, S8, S9, S10, S11, S12) or are very small (the remaining subjects). Indeed the authors described these as 'very little effect' and 'a large increase' respectively. On the group level, it resulted in significant PLV changes at maximum 2% (from 0.04 up to 0.06), and it is important to note that the PLV ranges from 0 to 1. Based on these small changes the authors made a strong conclusion, implying that the electrical stimulation of peripheral nerves in the scalp plays a significant role in entraining physiological tremor. Please provide the effect sizes for all the PLVs in the results and discuss them.

With regard to the intensities in experiment 2, the differences were splitted between 'Off', 'Low' and 'High'. Interestingly, in the 'Low' group, they varied between 0.16 and 1, while in the 'High' group between 0.8 and 2.5 mA. As there is an overlap of the intensity values in these two groups and a little inconsistency, it would be necessary to include the computational models of the intracranially induced E-fields for all intensity values for the 2nd and 3rd experiments.

Other minor points:

Experiment 2: testing the hypothesis of transcutaneous effects using anaesthesia. What I understand here is that the skin areas are not completely numb, simply the perception threshold is elevated. A sufficient proof would be needed here using some kind of complete nerve anaesthesia. Ref. 25 cannot be used as supportive in the present context since it is unclear if the DC component or the AC component were responsible for the results.

"Topical anesthetics do not penetrate deeply into the body and so will not affect the threshold potential of cortical neurons under the tACS electrodes." This sentence should be removed.

"The mean stimulation peak-amplitudes (not peak-to-peak)," still needs clarification; is it peak to baseline?

"Invasive measurements in epilepsy patients have already shown that the electric fields generated in the cortex with this type of tACS are not strong enough to cause neural entrainment³³." It depends on the experimental conditions. Other studies including nonhuman primates and epilepsy patients found that classically used intensities (current densities) can entrain neuronal oscillations (Opitz et. al, 2016).

References:

So, P. P. M., Stuchly, M. A., & Nyenhuis, J. A. (2004). Peripheral nerve stimulation by gradient switching fields in magnetic resonance imaging. *IEEE Transactions on Biomedical Engineering*, 51(11), 1907–1914. <https://doi.org/10.1109/TBME.2004.834251>

Vossen, A., Gross, J., & Thut, G. (2015). Alpha power increase after transcranial alternating current stimulation at alpha frequency (a-tACS) reflects plastic changes rather than entrainment. *Brain Stimulation*, 8(3), 499–508. <https://doi.org/10.1016/j.brs.2014.12.004>

Hanslmayr, S., Matuschek, J., & Fellner, M. C. (2014). Entrainment of prefrontal beta oscillations

induces an endogenous echo and impairs memory formation. *Current Biology*, 24(8), 904–909.
<https://doi.org/10.1016/j.cub.2014.03.007>
Opitz, A., Falchier, A., Yan, C.-G., Yeagle, E. M., Linn, G. S., Megevand, P., Schroeder, C. E. (2016). Spatiotemporal structure of intracranial electric fields induced by transcranial electric stimulation in humans and nonhuman primates. *Scientific Reports*, 6(1), 31236.
<https://doi.org/10.1038/srep31236>

Reviewer #2 (Remarks to the Author):

This work aims to distinguish the direct effects of tACS on the brain versus indirect effects due to peripheral nerve stimulation. This is a very timely and worthwhile effort. In rodent the study shows that weak 1Hz sinusoidal currents applied to the skull entrain neuronal firing (in absence of any scalp nerves that could have been stimulated). Similar effects are achieved with larger intensities applied to the limbs ruling out direct brain stimulation. In healthy human subjects it is shown that finger tremor can be entrained by sinusoidal electrical stimulation at the physiological tremor frequency (~10Hz), when the electrodes are placed on the scalp or on the contralateral arm, and that this effect is reduced when a topical anesthetic is applied to the scalp. The authors conclude that motor effects of tACS are dominated by peripheral nerve stimulation and are not the result of direct brain stimulation. The basic hypothesis that tACS effects are mediated by peripheral nerve stimulation is quite compelling and the experiments to test this are well conceived.

However, the authors overreach in the interpretation of the data when they conclude, in some places, that tACS has no direct effect on the brain. Experiment #1 on rodents shows neural entrainment in the absence of peripheral nerve stimulation, which seems to directly support the notion that transcranial stimulation with 1V/m does in fact have a direct effect on the brain. The experiment #2 with anesthesia on the scalp did not abolish the entrainment effects suggesting that peripheral stimulation may not be necessary to entrain these tremors. Experiment #3 shows that tremor can be entrained without stimulating the brain, but this is not surprising given that this kind of tremor apparently can be entrained in a number of ways, including visual stimulation. In total, it is fair to conclude from the data that peripheral nerve stimulation can entrain physiological tremor. However, one can not conclude from this data that the brain is not affected directly, or that this results generalizes beyond this tremor effect to other motor function, or that it generalizes to other frequencies. And certainly nothing can be said about other functions such as auditory perception or memory, as it is indicated at one point in the discussion.

I suggest the authors carefully edit the current paper to better calibrate their claims, namely that the data is consistent with both direct and indirect effects. Additionally, a larger sample may provide more reliable evidence, as currently the sample size of 12 subjects with weak effects seems inadequate for the importance of their central claim to this research area.

Detailed comments (more or less in the order they appear in the manuscript):

In experiment #1 the transcutaneous case uses 1, 2 and 3 mA. For a small animal these are huge intensities. In some sense the surprising result of this experiment is that in the brain 0.1mA was enough and 10 times more current was needed on the skin to achieve comparable effects on spike entrainment – approximate 0.2mA vs 2mA, right? This experiment clearly supports direct effects on the brain, and directly contradicts the main conclusion that effects are purely peripheral.

“A focused montage was chosen because it limits the spread of current to other cranial nerves such as the optic nerve, and it is known that stimulation of the optic nerve also causes neural entrainment 10,41” Not sure reference 41 is relevant here, but you should cite: Kar K, Krekelberg

B. Transcranial electrical stimulation over visual cortex evokes phosphenes with a retinal origin. *Journal of neurophysiology*. 2012 Aug 1;108(8):2173-8.

"When the scalp was anesthetized [in the sample subject], increasing tACS amplitude did not cause tremor entrainment to increase." Is this a statistical statement? If so, please add the test statistics. Otherwise say there was no "numerical" increase in PLV.

Figure 4 bottom right has circles (one for each condition), which are not explained in the figure caption. They are eventually explained in the text. By looking at the scatter this subject really seems quite cherry-picked to exaggerate the effect of the topical anesthetic.

"We applied tACS, matched to the individual volunteers physiological tremor frequency". Please report the frequencies in the main text/figures. Effects of stimulation are thought to depend critically on frequency (e.g. physiological mechanisms causing 1Hz oscillations are quite different from 10Hz). In general, the paper should qualify tACS by frequency. It is quite possible that effects on one frequency do not replicated at another, and vice-versa: lack of an effect at one frequency does not mean that tACS is ineffective at all frequencies. For a review on the different types of mechanisms/effects observed at at different frequencies see Reato D, Rahman A, Bikson M, Parra LC. Effects of weak transcranial alternating current stimulation on brain activity—a review of known mechanisms from animal studies. *Frontiers in human neuroscience*. 2013 Oct 23;7:687.

Figure 5 suggests that entrainment did not entirely disappear with local anesthetics. There seems to be no significant difference with anesthetic in the high stimulation condition, and no difference to no anesthesia. There is also no interaction between anesthetic and intensity of stimulation as would be predicted by the hypothesis. So overall this results do not rule out a residual transcranial effect. Given the strong entrainment of neural activity in rodent in Experiment 1 with transcranial-only stimulation, you can not rule out transcranial effects based on these data.

"These results indicate that stimulation of peripheral nerves in the scalp plays a significant role in entraining tremor and thus support our hypothesis that tACS effects are mediated by a transcutaneous mechanism." There is an important ambiguity in the language here. The results show that peripheral nerve stimulation plays /some/ role in entrainment. The result does not show that peripheral stimulation is the /only/ cause for entrainment. To show that, one would have to show the complete abolishment of entrainment (which is obviously difficult as a lack significant effect does not mean there is no effect at all). Apparently the local anesthetic is only partially effective (figure 5) and this may have caused the residual entrainment and its apparent dependence on stimulation amplitude (was this significant?). However, one can not rule out that the remaining entrainment is not the result of direct brain stimulation as an additional factor. In total, rather than "mediated" the better word may be "contribute". Now, of course the investigators can hypothesize anything they like. But to say that the result "supports" this hypothesis is a bit strong. At most they could say here that the result "is consistent with" the hypothesis. I would not discuss these small word choices at such length if it was not that this is at the core of the problem I see with the current writeup, which is that the results are over-interpreted as evidence for a pure peripheral effect in experiments 1 and 2. When really both are equally consistent with a concurrent direct brain stimulation effect.

The investigators are to be commended for showing individual subject data in some of the figures (3 & 4). However, some other figures still only show bar/box plots which hide the data (5 and 6 bottom right). Please show individual data in all cases. Additionally, since these are repeated measures on the same subjects it would be appropriate to connect points that belong to the same subjects with a line. This often makes it easier to see small within-subject effects despite large between-subject variances.

Incidentally, where individual points are shown, I count around 36 points, but there are only 12 subjects. Please clarify where that factor of 3 comes from. In particular, please clarify how many

points were used for the statistical analysis. Evidently repeated measures on the same subject/condition need to be averaged before using them in the analysis, as they are likely correlated and would inflate the statistical significance.

Please explain early on in the text what you mean by tremor. In most papers I have read, tremor refers to abnormal tremor such as in Parkinson. Here you test normal tremor which is more or less pronounced across subjects. This was not clear to me until I reached the methods section. An even there, the Experimental protocol don't add much clarity. It only mentions "tremor inducing posture". I finally found it when reading the methods on tremor measurement, which is really quite late given the importance of the procedure. Please explain the procedure early in the results section on experiment 2.

In caption of Figure 6 you write "Even when the tACS electrodes were not positioned on the head, increasing stimulation amplitude still caused an increase in tremor entrainment in this subject." It appears that there is no difference in entrainment between the two stimulation intensities. Suggest rephrasing.

Supplementary figure 6 is quite compelling. It would be great if this data was available for all subjects. The claim that is made in this paper is quite strong, by questioning the conclusion of docents of tACS studies. It would be important to base this claim on equally strong data. Given that some of the effects are weak, with PLV raising only a bit above noise levels and only 9 responders out of 12, this study seems underpowered. The human experiments do not seem overly complicated and could be readily reproduced on a larger cohort. A good way to do this would be to use the current data to establish required sample size with a corresponding power analysis, and then repeat the experiments exactly on a larger sample with identical outcome measures and statistical tests. This would be essentially a replication of the current experiment and, if successful, lend quite a bit of support for the strong claims made here. Collecting data as shown in figure 6 may go a long way to show that differences across subjects are larger than differences across stimulation sites, pointing at a common mechanism. But even just a repetition of Experiment #2, showing a significant interaction between anesthesia and amplitude would be helpful.

"Results from our study provide the first direct evidence that the effects of tACS on motor function are mostly caused by transcutaneous stimulation and not by transcranial stimulation as is widely assumed. The observed patterns of neural entrainment were similar for both transcutaneous-only and transcranial-only stimulation. This supports the hypothesis that a transcutaneous mechanism could be responsible for many of the tACS effects observed in humans."

All this strikes me as over-interpretation of the results. The rodent experiment showed the exact opposite of what is stated here. Namely, that transcranial-only was effective at entraining neuronal activity. No one would ever doubt that rhythmic activation of peripheral nerves causes rhythmic spiking in the brain. To the critics of tACS, the surprising finding is that the same happened when peripheral nerves were not present at all (transcranial-only). If anything, the effects seem to be stronger in that case (requiring only 1/10 of the intensity when stimulating transcranially).

"The results from Experiments 2 and 3 show that the effects of tACS on physiological tremor are mostly caused by the transcutaneous, and not the transcranial mechanism." Again, there is no support in the data for the word "mostly" in this sentence. Rather, the data suggests that peripheral stimulation alone can entrain tremor. But it does in no way rule out direct brain stimulation effects.

"Our results show that the effects of tACS on physiological tremor are not mediated by transcranial stimulation of neurons in the motor cortex but are caused by transcutaneous stimulation of peripheral nerves in the skin." And here again "not mediated by transcranial" is over-reaching. There is no evidence for that.

"tACS effects on non-motor systems". In this paragraph, again, there is an overreach. What is shown is that peripheral stimulation can entrain tremor, and this should motivate investigators to be careful in their interpretation of other tACS studies. However, the jump to say that all motor effects (beyond tremor entrainment) and all other hearing, cognitive and memory effects may all be the result of peripheral stimulation is exaggerated.

"Here, we have shown that the tACS mechanism, at least for the motor system, is mostly caused by transcutaneous stimulation and not transcranial stimulation. ... " Again, too broad. You have shown tremor entrainment with peripheral stimulation. Nothing more. The "motor system" does a lot more than tremble.

"In our study, we showed that the transcutaneous mechanism is the dominant mechanism driving tremor entrainment via tACS....Our results clearly show that tACS effects on the motor system are mostly caused by transcutaneous stimulation of peripheral nerves in the skin and not by transcranial stimulation of cortical neurons as had been previously assumed." Again, "dominant", "mostly" is too strong.

"Therefore, tACS induced phosphenes (which we assume are caused by electrical stimulation of the optic nerve 52,53 , although there is some debate on this 54)" There really is not much debate about this anymore since the publication of Kar & Krekelberg 2012.

The methods explain that the HIGH intensity condition was adjusted in steps of 0.1mA to be comfortable. Yet the table in the supplement suggests otherwise as these are mostly multiples of 0.5. Please clarify. Also, why did you not keep these numbers all constant at a less aggressive level that works for most subjects to reduce variability?

Its a bit unusual to report so many digits for z-statistics.

Reviewer #3 (Remarks to the Author):

Given the rousing debate on whether or not tACS can effectively stimulate the human cortex, this innovative study appears timely and is potentially relevant for a broad neuroscientific readership, provided that the reported findings do actually generalize from the motor cortex to other brain regions and from the phenomenon of tACS-induced tremor entrainment to other cognitive functions. However, for this generalization currently no evidence is provided. While the authors do formally restrict their interpretation to the contribution of transcutaneous contributions to tremor entrainment, the gist and general tone of the paper goes far beyond that limited claim. While I strongly encourage critical scrutiny when it comes to the mechanisms underlying non-invasive brain stimulation techniques, and I very much appreciate the presented study, particular care must be taken not to erroneously put all entrainment effects of tACS under general suspicion based on this particular finding. I have the following specific concerns:

1. It has to be made very clear in the title, abstract, and throughout the paper that conclusions can only be drawn for the contribution of transcutaneous tACS effects on tACS-induced tremor entrainment, and not to any other stimulation site or cognitive function.

2. While the study shows a contribution of transcutaneous tACS effects on tACS-induced tremor entrainment, it does not disprove that there is generally a transcranial contribution to this phenomenon. In particular, other studies have used different electrode montages (M1 to ipsilateral arm) and may have more effectively stimulated the relevant brain regions than the local centre-surround montage used in the current study. Also, since local anaesthesia did neither completely abolish tremor entrainment, nor did it abolish sensory perception of the stimulation, the two

factors remain entangled.

3. While it is very difficult to match stimulation intensity for rat and human transcranial brain stimulation, the authors did their best and actually report entrainment of spiking in rat motor cortex by "transcranial-only" stimulation at intensities comparable to what can be achieved in humans. This result is actually supporting rather than disproving the possibility for a truly transcranial contribution.

4. The neuronal mechanism behind the suggested peripheral transcutaneous entrainment effect on M1 remains obscure and is not discussed by the authors. If it is mediated via afferent input to S1 and thus indirectly to M1, it should be mainly contralateral to the stimulated limb (experiment 1), but it works equally well for both left and right limbs. Even more unclear are the mechanisms by which somatosensory stimulation anywhere on the body surface would entrain neuronal activity everywhere in the brain. In experiment 1, spiking data was only acquired from left M1, while a generalization to other brain regions would have required recordings of other (non-motor, e.g. visual and associative) cortical areas.

5. In analogy to experiment 2, the ultimate proof of a peripheral origin of the observed entrainment of spiking in rat motor cortex would be a deafferentation, lesioning the nerves projecting from a locally stimulated limb (with both electrodes on the same limb (as in experiment 3) to the motor cortex.

6. It is well possible that both transcranial and transcutaneous stimulation effects can cause entrainment of spiking and physiological tremor, and the fact that both stimulation forms cause similar entrainment in experiment 1 supports this notion. It is even conceivable that both mechanisms work in parallel without the effects accumulating, but with each one per se being sufficient to entrain spiking and tremor, respectively. In both rats and humans, concurrent transcutaneous and transcranial stimulation (with an arbitrarily phase-delay between the stimulation signals) would reveal which one provides the predominant entrainment of M1

Reviewer #1 (Remarks to the Author):

This article comes with the aim of proposing a new mechanistic explanation for the results induced by transcranial alternating current stimulation (tACS) in brain research literature for sensory, cognitive and motor processes. The article is driven by a hypothesis built on a (largely opinion driven) literature survey, trying to show that external electrical stimulation of the peripheral nervous system can entrain central neuronal populations in a rhythmic fashion. The authors question the direct entrainment of central nervous system neurons by tACS, hypothesizing that transcutaneous stimulation can be accounted for the effects of tACS, given the magnitude of the generated electric field over the skin and skull compartments. This was tested in a series of experiments involving rodents and humans, with alternating currents applied to skull, scalp and arm of rats and healthy humans.

Reply:

We thank the reviewer for their detailed and insightful comments. We have taken them on board and substantially revised the manuscript accordingly and included new data and experiments. We believe that the reviewers input has improved the revised manuscript.

Reviewer #1:

The aim of the study has high significance, nevertheless, with regard to the recent version I have the following concerns.

Overall I am skeptical concerning the use of physiological tremor measurement as a biomarker for the claim raised here. It would be much more straight forward to use methods to directly evaluate neuronal function, such as EEG or MEG, combined with the peripheral-only stimulation in experiment 3, to prove neural entrainment at the scalp level. Additionally, the terminology is problematic. I prefer a classification of physiological tremor confined to inevitable mechanical noise arising from remaining unsmoothed muscles twitches even if Hennemann's principle is accomplished optimally. What is used here is in my preferred terminology "enhanced physiological tremor" with spinal involvement.

In general, for such a strong substantial claim for tentatively criticising the whole field, better experimental evidence is needed.

Reply:

To address this specific concern we have now included two completely new experiments in the manuscript. In the first experiment (Experiment 2B in the revised manuscript), we tested the effect of tACS on pathological tremor in 12 essential tremor (ET) patients, split into two groups: 6 ET patient received tACS without scalp anesthesia and a different group of 6 ET patients received tACS with scalp anesthesia. In line with our previous results on physiological tremor, we found that tACS could entrain pathological tremor, but that when the scalp was anesthetized tACS could not entrain pathological tremor. The second experiment performed (Experiment 4 in the revised manuscript) was that suggested by the reviewer: namely, we investigated the effect of peripheral nerve stimulation on EEG activity. Briefly, in 12 healthy volunteers we measured EEG beta activity and determined each subject's individual beta peak frequency. We then applied pulsed electrical stimulation using two closely spaced electrodes positioned on the upper arm (i.e. peripheral-only stimulation) at a frequency matched to the individual's beta peak frequency. We found that rhythmic stimulation of a peripheral nerve causes entrainment of EEG activity in healthy volunteers. This new data expand our original findings beyond effects on physiological tremor and thus greatly strengthen our conclusions.

We initially tried the EEG experiment using sine wave stimulation but found that there was a small sine wave artifact present in the EEG recording. It was impossible to completely separate this artifact from the EEG beta activity. Therefore, we used a short stimulation pulse (0.44 ms) repeated at the same frequency as the sine wave. The pulses also caused a very small artifact in the recorded EEG signal of most subjects. However, since the artifact was extremely short and its frequency spectrum (that of a rectangular pulse) was much higher than the EEG frequency band of interest (~20 Hz), it could be easily removed using a low-pass filter (see Supplementary Results, Fig. 10).

The methods and results for both new experiments are explained in detail in the revised manuscript.

Where appropriate, we now use the reviewer's terminology of 'enhanced physiological tremor' instead of 'physiological tremor'. Since the new experiment now broadens our results to include pathological tremor, at some points in the manuscript we have changed 'physiological tremor' to simply 'tremor'.

Reviewer #1:

The animal and human part suffer from several pitfalls, which are not compensated by putting them together in a fused paper.

Reply:

We believe the new experiments now address any potential pitfalls in the human experiments. Other specific issues with the human experiments identified by the reviewer are addressed below.

The reviewer also described specific pitfalls in the animal experiments below in more detail and we provide our detailed response to those comments below.

With the new experiments, and the explanations given below, we believe any potential issues are now clearly resolved. We believe that grouping the animal and human experiments together in one paper makes a much more convincing case for a role of transcutaneous stimulation. It elucidates the neural mechanism, in addition to providing complimentary behavioral evidence in humans. This combined approach strengthens the manuscripts conclusions.

Reviewer #1:

Also the literature survey has to be put in a greater context. Data obtained by previous work on peripheral nerve stimulation in general and its impact on plasticity have to be quoted. Generally, I miss a thorough literature survey in which voltage gradients surpass the threshold excitability of peripheral neurons. This needs to be summarized in terms of V/m, (e.g. 3.8 to 5.8 V/m for the fields exceeded in 0.5% of tissue volume (skin and fat of the torso)) (So et. al, 2004). It remains unclear if the currents used in this manuscript are below or above peripheral stimulation thresholds.

Reply:

The currents used in the HIGH amplitude conditions (around 2.5mA) are strong enough to stimulate peripheral nerves. We know this because the subjects report feeling the stimulation (see the Sensation Rating reported in Fig. 5 and 6 in the manuscript). It is less clear if the LOW (0.5 mA) amplitude conditions are strong enough to stimulate peripheral nerves as some subjects report a light sensation and others report none. Thus, the LOW conditions are just on the edge of being perceptible, but we would still expect them to stimulate some nerves before the perceptual threshold is reached. In line with this, the average sensation threshold (Fig. 5, left panel) is 0.5 mA in the No Anesthesia condition and is 1 mA in the Anesthesia condition. Fig. 1 shows the electric field strength reached in the skin (and other tissues) when a 1 mA current is applied. The electric field in the skin easily reaches 20 V/m for 1 mA tACS and thus is certainly above the typical electrical field strengths associated with stimulation of peripheral nerves. Values from these models scale linearly.

Thus, for 2 mA tACS we would expect 40 V/m in the skin, etc. We now make this point more clearly in the Introduction and Discussion. Additionally, and in line with a comment below, we now provide an estimate of the electric field strength in the skin and brain for each subject in Experiment 2 (Supplementary Results, Table 4) and in the skin for each subject in Experiment 3 (Supplementary Results, Table 4). As explained in more detail in response to a comment below, we prefer not to expand the discussion to the effects of electrical stimulation on plasticity, as this is not a central (nor critical) point in our manuscript.

Reviewer #1:

Detailed comments:

The aim of experiment 1 was to prove that transcutaneous electrical stimulation can entrain neural activity as well as stimulation applied directly to the skull. My main concern here is that the transcutaneous stimulation intensities are much too high. The forelimb electrode is quite close to the brain. I would expect that a 1 mA stimulation at the forelimb generates far field potentials of 0.1 mA current flow in the brain of the rat.

Reply:

During the experiments, we were aware of the possibility that current from the transcutaneous only stimulation could potentially spread out into the brain (through volume conduction) and thus cause direct stimulation of neurons in the cortex (similar to the transcranial-only condition). Our experimental setup allowed us to monitor the electric field in the brain caused by both the transcranial-only stimulation condition and the transcutaneous-only stimulation condition. Fig. 1 in the supplementary methods outlines our procedure for doing this. Briefly, the stimulation (either transcranial or transcutaneous) is a low frequency sine wave and the voltage in the brain at this frequency was directly measured using the 32-channel recording probe. It was in fact this data that we use to establish that our transcranial-only stimulation caused an electric field strength in the brain of around 1 V/m for every 0.1 mA of transcranially applied current. We now used exactly the same measurement procedures to quantify the electric field strength in the brain during transcutaneous-only stimulation and found that for 3 mA transcutaneous-only we get an electric field of around 0.07 V/m in the brain. This is in the same range as the local field potential strength. Indeed, using the same procedures we quantified the average electric field strength of the local field potential during stimulation off condition for both the transcranial-only and the transcutaneous-only conditions and found it to be 0.041 V/m and 0.037 V/m respectively. Thus, the 0.07 V/m electric field measured during 3 mA of transcutaneous-only stimulation is likely to be due to the local field potential and not volume conduction from the stimulated limb. In any case, even if it was due to volume conduction it is still well below the 1 V/m needed to cause direct neural entrainment of cortical neurons. We have now added a new figure to Supplementary Results (Figure 2) fully explaining and describing these results and we refer to this in the main text.

Reviewer #1:

Here the authors used 2 stimulation conditions called transcranial- and transcutaneous-only. They compared the recordings from 32-channel silicone probe placed over the corresponding motor area during 1 minute of ON stimulation versus previous 1 minute of OFF stimulation. They expected to find the effect of entrainment by increasing the phase locking value (PLV) in the ON condition in comparison with the OFF condition.

Throughout the article, PLV is a measure of synchrony between the applied sinusoidal wave and recorded neurophysiological activity. It is directly established by the stimulus onset, which could not be presented in the OFF condition. The authors wrote that the stimulus period from the ON condition was used for defining the stimulus period in the OFF condition. The procedure is not described in detail, therefore it is difficult to evaluate how they compared the two conditions. This

might be crucial because the presented PLV (Fig.2) for the transcranial-only OFF condition (0.14) is twice as high for the transcutaneous-only OFF condition (0.05). It would be reasonable to implement any baseline correction or to provide the absence of statistical differences for all OFF conditions.

Reply:

Indeed, the calculation of the PLV during the OFF condition was rather briefly described, especially considering it is such a critical metric. We have now updated the Methods Section (Experiment 1, Post-experiment data analysis) to describe the procedure more fully.

The PLVs during the OFF conditions in the example in Fig. 2 are quite different. In general we found that PLVs from one neuron to the next could be quite different (both for OFF and ON conditions). This is one of the reasons that we always compare the PLV in the OFF condition to the PLV in the ON condition for each neuron. This is in fact the 'baseline correction' that the reviewer mentions. In line with this, all the statistics as shown on Fig. 3 are paired statistics which compare the PLV in the OFF to the PLV in the ON. Again, this is the 'baseline' correction the reviewer is referring to. Secondly, when we did higher level statistics to test the effect of increasing stimulation amplitude we always compared the PLV_{dif} which is the difference between the OFF and ON condition for each neuron. Thus, the baseline correction the reviewer is referring to is already in place.

Nevertheless, we also took the reviewers suggestion to statistically check for any differences between all the OFF conditions. We did this by first calculating the mean for all the OFF PLVs pooled across all conditions. We then used a one sample Wilcoxon signed-rank test to compare the OFF PLVs for each condition to the pooled mean and corrected for multiple testing. We found no OFF conditions that were significantly different from the pooled mean. As a check, applying exactly the same statistical methods to all the ON conditions does show that the conditions at certain amplitudes are significantly different from the pooled mean.

Reviewer #1:

As the main conclusion of the Experiment 1 is that all stimulation conditions and all electrode configurations except of Contralateral-Hind and Ipsilateral-Hind resulted in significantly increased PLV, this may suggest the drawback of the proper control experiment or additionally that the PLV does not reflect neural entrainment.

Reply:

We believe that the new analysis of the electric field in the brain during transcutaneous-only stimulation now shows convincingly that volume conduction was not an issue during limb stimulation. We would also argue that the fact that two experimental conditions did not show a significantly increased PLV (contra-lateral hind and ipsi-lateral hind) actually indicates the observed effects are not due to some kind of artifact since the effects were not present in all conditions. In line with this, during all conditions (transcranial-only or transcutaneous-only) only about 75% of neurons showed an increase in PLV from the OFF to the ON conditions. These neurons which did not show entrainment are included in the group analysis. However, since the majority of neurons did show an increase in PLV from OFF to ON, we measured an effect at the group level.

We have now added a paragraph to the Results Section to point this out, in addition to a new table in Supplementary Results (Table 3) listing the percentage of neurons that showed increased neural entrainment for each condition. We have also added a new figure to Supplementary Results (Figure 3) of one neuron showing very different responses to different types of transcutaneous-only stimulation.

Thus, we believe that the necessary experimental checks and controls are in place.

We answer the point on whether or not PLV reflects neural entrainment below.

Reviewer #1:

Related to this point, the term ‘neural entrainment’ is not well defined. On the one hand, Vossen et. al, 2015 distinguished on-line effect which is factually “entrainment” and off-line effect which reflects rather plastic changes than entrainment. On the other hand, Hanslmayr et. al, 2014 reported that off-line effect was evident only after behaviorally relevant stimulation, resulting in memory impairment (in that case). They suggested that the evidence of entrainment ongoing oscillations is in the presence of after effect of stimulation. Therefore, I think it is necessary here to provide the evidence whether there were any differences between during and immediately after transcranial- and transcutaneous-only stimulation.

Reply:

Indeed, we did not provide a clear definition of the term ‘neural entrainment’ in the manuscript. By neural entrainment, we mean that the timing or phase of a neural signal is adjusted to become synchronized with that of an external stimulus. In that sense, our definition of neural entrainment is mathematically well described by the PLV. We now define the term neural entrainment more clearly in the Manuscript in both the Results and Methods sections.

Using this definition, most neural entrainment will occur during stimulation (or on-line). It may be that the neural signal remains entrained to the stimulus even when the stimulus is switched off (off-line). Or, as suggested by Vossen et al, it could be that changes in neural signal after stimulation reflect plasticity effects. This is an interesting issue and debate. However, in this manuscript we did not specifically investigate the difference between these two possibilities. Separating the two possibilities is also not central (nor critical) to the message in our manuscript. Given this, and the fact that the manuscript is already rather long and contains quite a few experiments, we would prefer not to comment on this issue in the current manuscript.

Reviewer #1:

Furthermore, the conclusion of entrainment by one or another type of stimulation needs to be supported by a control condition, including two closely spaced electrodes that sufficiently stimulates skin receptors but inducing shunting in order to prevent brain stimulation. This control condition also applies for the human studies.

Reply:

The analysis of the signals recorded from the 32-channel probe in the brain during limb stimulation shows that none of the transcutaneous-only conditions directly produced a significant electric field in the brain. This is now clearly presented and discussed in Supplementary Results, Figure 1 and 2. Thus, we believe this provides a sufficient control condition for the animal experiments.

For the human experiments, the transcutaneous-only electrodes were all placed on the arm and were separated by 8 cm. This is described in the Methods, Experiment 3, Electrode Placement. Thus, for the human experiments we already used the control condition suggested by the reviewer.

Reviewer #1:

From the second and the third experiments, the authors reported that PLV was increased for most of the subjects in the no anesthesia condition while it decreased under anesthesia condition in the second experiment. Nevertheless, a closer look at the individual data (Supplementary Fig. 2) shows that PLV changes between conditions are almost flat (S1, S4, S6, S8, S9, S10, S11, S12) or are very small (the remaining subjects). Indeed the authors described these as ‘very little effect’ and ‘a large increase’ respectively. On the group level, it resulted in significant PLV changes at maximum 2% (from 0.04 up to 0.06), and it is important to note that the PLV ranges from 0 to 1. Based on these

small changes the authors made a strong conclusion, implying that the electrical stimulation of peripheral nerves in the scalp plays a significant role in entraining physiological tremor. Please provide the effect sizes for all the PLVs in the results and discuss them.

Reply:

We agree that the changes to tremor PLVs caused by tACS appear to be small. However, we would stress that these values are in line with those reported by other authors (Brown and Brittain studies) and by us in a previous study (Khatoun et al, Using high-amplitude and focused transcranial alternating current stimulation to entrain physiological tremor, 2018). In addition, we and others consistently find significant effects on tremor PLV at the group level. Upon the reviewers suggestion we calculated effects sizes using Cohen's d metric (difference in the two means divided by the pooled standard deviation) and now report these when we did detect a significant difference. We actually found effect sizes with a d of around 1, which puts them in the large effects size range as defined by Cohen (Statistical power analysis for the behavioral sciences. (L. Erlbaum Associates, 1988)). Two factors contribute to the large effect sizes: 1) we used a paired design (and used the appropriate Cohen's d metric for paired data), 2) the standard deviations in the PLVs are relatively small. The small differences apparent in the means of the bar graphs in Fig. 4 and 6 is somewhat misleading and does not do a great job of presenting the paired data. Therefore (and based on the suggestion of another reviewer), we have now added individual data points to the bar graphs and connected points from the same subject with a line. This gives a better representation of the paired data and the effects on individual subjects. We now also report effect sizes for PLV comparisons that showed a significant effect in Experiments 2, 3 and 4. We have added a paragraph to Experiment 3 (paragraph 2) Results to comment on the effect sizes and PLVs.

Reviewer #1:

With regard to the intensities in experiment 2, the differences were splitted between 'Off', 'Low' and 'High'. Interestingly, in the 'Low' group, they varied between 0.16 and 1, while in the 'High' group between 0.8 and 2.5 mA. As there is an overlap of the intensity values in these two groups and a little inconsistency, it would be necessary to include the computational models of the intracranially induced E-fields for all intensity values for the 2nd and 3rd experiments.

Reply:

The main factor contributing to the range in amplitude levels was that some subjects were much more sensitive to stimulation than others. One subject was not comfortable going above 0.8 mA. These ranges are in line with other tACS studies in which tACS amplitude is often defined on an individual subject basis as a percentage of their sensation threshold. We have adopted the reviewer's suggestion and calculated the E-fields predicted from the computational head model (Fig. 1) for each stimulation intensity used. We now included this data Table 2 (Supplementary Results) with one column showing the model estimated E-field in the brain and another column for the skin in each subject. We opted for comparing E-fields in a 10 mm³ volume of skin and brain. The procedure for calculating this is now described in the Methods, *Human Head Computational Model*, final paragraph. This new data also address a previous comment of the reviewer related to the E-field in the skin needed to stimulate peripheral nerves. The model data indicate that the E-fields in the skin are certainly strong enough to cause stimulation of peripheral nerves. We do not have a computational model for the arm stimulation used in Experiment 3 and so were unable to include this. However, we believe the data shown for Experiment 2 are the most useful.

Reviewer #1:

Other minor points:

Experiment 2: testing the hypothesis of transcutaneous effects using anaesthesia. What I understand here is that the skin areas are not completely numb, simply the perception threshold is elevated. A sufficient proof would be needed here using some kind of complete nerve anaesthesia.

Reply:

This is correct. The skin becomes slightly numb due to the topical anesthesia blocking sodium channels in peripheral nerves. This leads to a large significant increase in the perceptual threshold for electrical stimulation (see Fig. 5). The only alternative forms of stronger anesthesia that we are aware of must be delivered via an injection into the skin or nerve directly. Based on our experience, our ethical committee would not give us permission to perform this kind of experiment on healthy volunteers, nor on patients (unless it was to deliver a therapy which may alleviate disease symptoms and this is certainly not the case here). Thus, we are unable to perform the experiment suggested by the reviewer. We also do not believe that it is necessary to perform this experiment. The amount of anesthesia delivered by the topical EMLA cream was sufficient to see a statistically large (based on Cohen's d effect size) and significant reduction in tremor entrainment PLV in both healthy volunteers and essential tremor patients (see results from Experiment 2).

Reviewer #1:

Ref. 25 cannot be used as supportive in the present context since it is unclear if the DC component or the AC component were responsible for the results.

Reply:

We now also add an extra reference which just uses tACS to modulate memory. We do agree that for Ref 25 it is not completely clear if the modulation was caused by the tACS or the tDCS. However, we prefer to leave it in place, as it is an important paper using both tACS and tDCS to modulate memory.

Reviewer #1:

"Topical anesthetics do not penetrate deeply into the body and so will not affect the threshold potential of cortical neurons under the tACS electrodes." This sentence should be removed.

Reply:

This does sound like an obvious and perhaps pointless statement to make. However, having presented these results to colleagues and discussed them, this was often a question that came up. Therefore, we prefer to include the sentence.

Reviewer #1:

"The mean stimulation peak-amplitudes (not peak-to-peak)," still needs clarification; is it peak to baseline?

Reply:

Yes. It is baseline or zero to peak. We now specify this at a few points in the text.

Reviewer #1:

"Invasive measurements in epilepsy patients have already shown that the electric fields generated in the cortex with this type of tACS are not strong enough to cause neural entrainment³³." It depends on the experimental conditions. Other studies including nonhuman primates and epilepsy patients found that classically used intensities (current densities) can entrain neuronal oscillations (Opitz et al, 2016).

Reply:

We were aware of the Opitz et al 2016 paper and reread it again. However, we did not find any results showing measurements of neural entrainment in that paper. The paper makes in-vivo measurements of the electric field strength but does not show any results relating to how these fields can or cannot entrain neural oscillations.

References:

- So, P. P. M., Stuchly, M. A., & Nyenhuis, J. A. (2004). Peripheral nerve stimulation by gradient switching fields in magnetic resonance imaging. *IEEE Transactions on Biomedical Engineering*, 51(11), 1907–1914. <https://doi.org/10.1109/TBME.2004.834251>
- Vossen, A., Gross, J., & Thut, G. (2015). Alpha power increase after transcranial alternating current stimulation at alpha frequency (a-tACS) reflects plastic changes rather than entrainment. *Brain Stimulation*, 8(3), 499–508. <https://doi.org/10.1016/j.brs.2014.12.004>
- Hanslmayr, S., Matuschek, J., & Fellner, M. C. (2014). Entrainment of prefrontal beta oscillations induces an endogenous echo and impairs memory formation. *Current Biology*, 24(8), 904–909. <https://doi.org/10.1016/j.cub.2014.03.007>
- Opitz, A., Falchier, A., Yan, C.-G., Yeagle, E. M., Linn, G. S., Megevand, P., Schroeder, C. E. (2016). Spatiotemporal structure of intracranial electric fields induced by transcranial electric stimulation in humans and nonhuman primates. *Scientific Reports*, 6(1), 31236. <https://doi.org/10.1038/srep31236>

Reviewer #2 (Remarks to the Author):

This work aims to distinguish the direct effects of tACS on the brain versus indirect effects due to peripheral nerve stimulation. This is a very timely and worthwhile effort. In rodent the study shows that weak 1Hz sinusoidal currents applied to the skull entrain neuronal firing (in absence of any scalp nerves that could have been stimulated). Similar effects are achieved with larger intensities applied to the limbs ruling out direct brain stimulation. In healthy human subjects it is shown that finger tremor can be entrained by sinusoidal electrical stimulation at the physiological tremor frequency (~10Hz), when the electrodes are placed on the scalp or on the contralateral arm, and that this effect is reduced when a topical anesthetic is applied to the scalp. The authors conclude that motor effects of tACS are dominated by peripheral nerve stimulation and are not the result of direct brain stimulation. The basic hypothesis that tACS effects are mediated by peripheral nerve stimulation is quite compelling and the experiments to test this are well conceived.

However, the authors overreach in the interpretation of the data when they conclude, in some places, that tACS has no direct effect on the brain. Experiment #1 on rodents shows neural entrainment in the absence of peripheral nerve stimulation, which seems to directly support the notion that transcranial stimulation with 1V/m does in fact have a direct effect on the brain. The experiment #2 with anesthesia on the scalp did not abolish the entrainment effects suggesting that peripheral stimulation may not be necessary to entrain these tremors. Experiment #3 shows that tremor can be entrained without stimulating the brain, but this is not surprising given that this kind of tremor apparently can be entrained in a number of ways, including visual stimulation. In total, it is fair to conclude from the data that peripheral nerve stimulation can entrain physiological tremor. However, one can not conclude from this data that the brain is not affected directly, or that this results generalizes beyond this tremor effect to other motor function, or that it generalizes to other frequencies. And certainly nothing can be said about other functions such as auditory perception or memory, as it is indicated at one point in the discussion.

I suggest the authors carefully edit the current paper to better calibrate their claims, namely that the data is consistent with both direct and indirect effects. Additionally, a larger sample may provide more reliable evidence, as currently the sample size of 12 subjects with weak effects seems inadequate for the importance of their central claim to this research area.

Reply:

We thank the reviewer for their careful and insightful review. We agree with the two main criticisms of the reviewer; namely that we may have over reached or generalized a little too much in our conclusion and that more experimental evidence would strengthen our central claim. We have addressed these issues in three specific ways

1) We conducted a new experiment in 12 essential tremor patients, split into two groups – 6 with without scalp anesthesia and 6 with scalp anesthesia. We found that tACS could entrain pathological tremor in the group without anesthesia, but tACS could not entrain pathological tremor in the group with anesthesia. This new experiment adds important evidence that supports our central claim that tACS effects on the motor system in humans are caused by stimulation of peripheral nerves. It also generalizes the effects to include both physiological and pathological tremor, which was requested by another reviewer. This experiment has been added to the manuscript as Experiment 2B.

We opted to extend the experimental evidence by testing tACS on pathological tremor (now Experiment 2B) and not by increasing the number of subjects in the physiological tremor experiment (now Experiment 2A) for a number of reasons: 1) As stated above it generalizes our findings beyond physiological tremor and to other frequencies. 2) The sample size of 12 in (now) experiment 2A was

already based on a power calculation (reported in Methods Section) using effects sizes from a previous experiment (Khatoun et al 2018, Using high-amplitude and focused transcranial alternating current stimulation to entrain physiological tremor. *Sci. Rep.* 8, 4927). 12 subjects is more than enough to observe effects of this size. 3) While the numerical changes in PLV are small, the effects size in statistical terms is by no means weak. Based on the suggestion of another reviewer, we now included effect size calculation (based on Cohen's d metric), which show that the effect sizes are in fact large. As suggested by this reviewer, our figures now do a better job of showing this by including individual subject data points connected by lines.

2) We conducted a second experiment in 12 healthy volunteers to measure the effects of peripheral nerve stimulation on EEG beta activity, also suggested by another reviewer. We measured individual EEG beta activity in subjects and then deliver peripheral nerve stimulation at that frequency. We found that peripheral nerve stimulation, even at very low sensation ratings, entrained EEG beta activity. In this experiment, we had to use pulsed stimulation, since it was not possible to satisfactorily remove the artifact caused by sinewave stimulation from the EEG signal. Thus, we can now generalize our findings to say that rhythmic peripheral nerve stimulation not only entrains tremor but it also entrains human neural activity. We also observe these effects across a range of frequencies: beta activity is around 20 Hz, physiological tremor is around 10 Hz and essential tremor is around 3 Hz. The EEG experiment has been added to the manuscript as Experiment 4.

3) We have thoroughly edited the manuscript (Abstract, Introduction and Discussion) to present a more balanced view. In summary: We now state that both weak direct transcranial stimulation and transcutaneous stimulation can cause neural entrainment. Our experimental evidence shows that in humans tACS effects on the motor system appear to be caused by transcutaneous stimulation. However, we cannot completely rule out some effects from transcranial stimulation. More experiments need to be done to test a possible effect of the transcutaneous mechanism in tACS effects on auditory, memory and cognitive function.

We believe the revised manuscript has been improved by the reviewers input and thank him for that. Below we address each specific comment in detail.

Reviewer #2:

Detailed comments (more or less in the order they appear in the manuscript):

In experiment #1 the transcutaneous case uses 1, 2 and 3 mA. For a small animal these are huge intensities. In some sense the surprising result of this experiment is that in the brain 0.1mA was enough and 10 times more current was needed on the skin to achieve comparable effects on spike entrainment – approximate 0.2mA vs 2mA, right? This experiment clearly supports direct effects on the brain, and directly contradicts the main conclusion that effects are purely peripheral.

Reply:

It is correct that comparatively high amplitudes were needed with transcutaneous stimulation compared to transcranial stimulation to see neural entrainment. However, one must bear in mind the large differences in skull thickness between the rodent and human. Models and measurements indicate that with 1 mA on the human scalp we only reach about 0.5 V/m in the brain. In contrast, 0.1 mA on the rat skull gave us an electric field strength of around 1 V/m in the rat brain. Additionally, there is no reason to expect peripheral nerve thresholds in the rat to be very different from those in humans. The skin may be thinner in the rat, but hair was not removed. Thus, one may expect reasonably similar stimulation amplitudes to be needed for the transcutaneous experiments in rats and humans. Concerning this, the transcutaneous levels chosen here are in line with most studies in humans.

We disagree with the statement that by showing that fields of around 1V/m can entrain neural activity we contradict our main conclusion. Our rat experiment simply shows that both transcranial and transcutaneous stimulation are viable mechanisms for causing neural entrainment. The human experiment (2, 3 and 4) were then designed to test which mechanism is dominating tACS effects on the motor system in humans. Our results indicate that tACS motor system effects appear more likely to be driven by the transcutaneous mechanism. We now state this position more clearly in the revised manuscript.

We agree that an interesting finding from our animal experiments was that fairly weak electric fields in the brain can cause some amount of neural entrainment. While it would appear that this transcranial mechanism does not play a big role in tACS motor system effects, it may do so in tACS effects on other systems. This remains to be tested. The revised version of the manuscript now addresses these points. We hope that it presents a clearer and more balanced view.

Reviewer #2:

“A focused montage was chosen because it limits the spread of current to other cranial nerves such as the optic nerve, and it is known that stimulation of the optic nerve also causes neural entrainment 10,41” Not sure reference 41 is relevant here, but you should cite: Kar K, Krekelberg B. Transcranial electrical stimulation over visual cortex evokes phosphenes with a retinal origin. *Journal of neurophysiology*. 2012 Aug 1;108(8):2173-8.

Reply:

Reference 41 (now 43) does show that flashing a light (i.e. visual stimulation of the optic nerve) entrains physiological tremor. We now make this clear in the sentence and also add the Kar et al reference suggested by the reviewer.

[43] Mehta, A. R., Pogosyan, A., Brown, P. & Brittain, J.-S. Montage Matters: The Influence of Transcranial Alternating Current Stimulation on Human Physiological Tremor. *Brain Stimul.* 8, 260–268 (2015)

Reviewer #2:

“When the scalp was anesthetized [in the sample subject], increasing tACS amplitude did not cause tremor entrainment to increase.” Is this a statistical statement? If so, please add the test statistics. Otherwise say there was no “numerical” increase in PLV.

Reply:

This has been changed.

Reviewer #2:

Figure 4 bottom right has circles (one for each condition), which are not explained in the figure caption. They are eventually explained in the text. By looking at the scatter this subject really seems quite cherry-picked to exaggerate the effect of the topical anesthetic.

Reply:

The grey circles are mentioned in the figure caption when describing this panel. Perhaps the reviewer missed this. We think that we do a fair job of showing all individual data as points in the figures in the main manuscript and as separate figures in the Supplementary Results. We have now added more individual data points to most figures as requested by the reviewer in a comment below. The subject shown on the top of Fig. 4 is a good example to illustrate the effects of tACS and topical skin anesthesia. We agree that this subjects data points do lie towards the higher end of the response scale, but we do not attempt to hide this and actually point-out that fact by showing the grey circle in the group plot.

Reviewer #2:

“We applied tACS, matched to the individual volunteers physiological tremor frequency”. Please report the frequencies in the main text/figures. Effects of stimulation are thought to depend critically on frequency (e.g. physiological mechanisms causing 1Hz oscillations are quite different from 10Hz). In general, the paper should qualify tACS by frequency. It is quite possible that effects on one frequency do not replicated at another, and vice-versa: lack of an effect at one frequency does not mean that tACS is ineffective at all frequencies. For a review on the different types of mechanisms/effects observed at at different frequencies see Reato D, Rahman A, Bikson M, Parra LC. Effects of weak transcranial alternating current stimulation on brain activity—a review of known mechanisms from animal studies. *Frontiers in human neuroscience*. 2013 Oct 23;7:687.

Reply:

We have now added the mean tremor frequencies for Experiment 2 and 3 and the mean beta peak frequency for Experiment 4. The frequency range of the oscillations in Experiment 1 were already reported. With the new experiments we show that transcutaneous stimulation of peripheral nerves has effects on the motor system across a wide range of frequencies: 1-2Hz in the rat brain, ~3 Hz for pathological tremor, ~10 Hz for physiological tremor and ~20 Hz for EEG beta activity. We now mentions this in the Discussion and refer the Reato et al paper discussing mechanisms.

Reviewer #2:

Figure 5 suggests that entrainment did not entirely disappear with local anesthetics. There seems to be no significant difference with anesthetic in the high stimulation condition, and no difference to no anesthesia. There is also no interaction between anesthetic and intensity of stimulation as would be predicted by the hypothesis. So overall this results do not rule out a residual transcranial effect. Given the strong entrainment of neural activity in rodent in Experiment 1 with transcranial-only stimulation, you can not rule out transcranial effects based on these data.

Reply:

In the healthy volunteers, the topical anesthetics did not completely block the effects of transcutaneous stimulation – subjects still felt something, particularly at higher stimulation levels. However, the topical anesthetics did have a large, significant, effect on sensation threshold and intensity rating (see Fig. 5). The fact that the subjects could still feel stimulation at the higher levels in the anesthesia condition probably contributes to the non-significant trend for increased PLV from the LOW to the HIGH anesthesia condition, and to the absence of a significant difference in the two HIGH conditions. Nevertheless, there is a significant difference between the two LOW conditions and a significant main effect of anesthesia (linear mixed model). The difference between the LOW anesthesia and no anesthesia conditions is in statistical terms a large effect (calculated using Cohen’s d metric). We now report this in the manuscript. An alternative would be to use stronger local anesthetics, but the only ones we are aware of must be injected into the skin or nerve. This procedure would not be approved by our ethics committee.

Our results, showing an effect of anesthesia on tremor entrainment is now strengthened by the new data on pathological tremor in essential tremor patients. In this case, the anesthesia appeared to have a very clear effect: 1) In the anesthesia group we observed no increase in PLV from the OFF to the ON condition and in this case the ON condition is quite high (5 mA). 2) As a results of effective anesthesia the patients did not feel the stimulation strongly, thus allow us to go to high stimulation amplitudes of 5 mA. We think the strong effects of the topical anesthesia in this group may be due to the thinner skin in the elderly patients, but cannot be sure.

Overall, based on evidence from the two experiments we can now say with reasonable confidence that the effects of tACS on tremor are mostly mediated by transcutaneous and not transcranial

stimulation. The electric field in the brain will be there during standard tACS, and it is certainly possible that it causes small amounts of neural entrainment in motor cortex in humans (although we currently don't have data on this). However, our experiments show that this electric field in the brain does not make a significant contribution to the observed tremor entrainment effect. We now clearly state this in the Discussion.

Reviewer #2:

"These results indicate that stimulation of peripheral nerves in the scalp plays a significant role in entraining tremor and thus support our hypothesis that tACS effects are mediated by a transcutaneous mechanism." There is an important ambiguity in the language here. The results show that peripheral nerve stimulation plays /some/ role in entrainment. The result does not show that peripheral stimulation is the /only/ cause for entrainment. To show that, one would have to show the complete abolishment of entrainment (which is obviously difficult as a lack significant effect does not mean there is no effect at all). Apparently the local anesthetic is only partially effective (figure 5) and this may have caused the residual entrainment and its apparent dependence on stimulation amplitude (was this significant?). However, one can not rule out that the remaining entrainment is not the result of direct brain stimulation as an additional factor. In total, rather than "mediated" the better word may be "contribute". Now, of course the investigators can hypothesize anything they like. But to say that the result "supports" this hypothesis is a bit strong. At most they could say here that the result "is consistent with" the hypothesis. I would not discuss these small word choices at such length if it was not that this is at the core of the problem I see with the current writeup, which is that the results are over-interpreted as evidence for a pure peripheral effect in experiments 1 and 2. When really both are equally consistent with a concurrent direct brain stimulation effect.

Reply:

The parts of this comment has been answered in our reply to the previous comment. Regarding word choice, as we stated earlier, we have taken the reviewers comments on board and have edited the manuscript to present a more careful interpretation of the results and a more balanced, view.

Reviewer #2:

The investigators are to be commended for showing individual subject data in some of the figures (3 & 4). However, some other figures still only show bar/box plots which hide the data (5 and 6 bottom right). Please show individual data in all cases. Additionally, since these are repeated measures on the same subjects it would be appropriate to connect points that belong to the same subjects with a line. This often makes it easier to see small within-subject effects despite large between-subject variances.

Reply:

We have now added individual data points to Fig. 4 with lines joining points from the same subject. Indeed, we think it now gives a better representation of the complete dataset. We have also added individual data points to the threshold and sensation ratings in Figs. 5 and 6. Where possible (without getting too messy) we have added lines joining individual subjects.

Reviewer #2:

Incidentally, where individual points are shown, I count around 36 points, but there are only 12 subjects. Please clarify where that factor of 3 comes from. In particular, please clarify how many points were used for the statistical analysis. Evidently repeated measures on the same subject/condition need to be averaged before using them in the analysis, as they are likely correlated and would inflate the statistical significance.

Reply:

The 36 points come because we have three repetitions (i.e. three measurements of PLV) for each condition in Experiment 2A. When performing the post-hoc (Wilcoxon signed-rank) and the two-way repeated measures ANOVA, we of course averaged these three PLV values together to give one measurement per subject per condition. When doing the linear mixed model analysis we used all 36 points as these models are not sensitive to correlated values. However, we also tried the linear mixed models using averaged values and just 12 points. While the exact values change a little, the main effects and conclusions remain the same.

Reviewer #2:

Please explain early on in the text what you mean by tremor. In most papers I have read, tremor refers to abnormal tremor such as in Parkinson. Here you test normal tremor which is more or less pronounced across subjects. This was not clear to me until I reached the methods section. An even there, the Experimental protocol don't add much clarity. It only mentions "tremor inducing posture". I finally found it when reading the methods on tremor measurement, which is really quite late given the importance of the procedure. Please explain the procedure early in the results section on experiment 2.

Reply:

We now give more details on physiological tremor earlier in the results section. We have also added a new experiment (Experiment 2B) that extends the results to include pathological tremor.

Reviewer #2:

In caption of Figure 6 you write "Even when the tACS electrodes were not positioned on the head, increasing stimulation amplitude still caused an increase in tremor entrainment in this subject." It appears that there is no difference in entrainment between the two stimulation intensities. Suggest rephrasing.

Reply:

This sentence has been rewritten.

Reviewer #2:

Supplementary figure 6 is quite compelling. It would be great if this data was available for all subjects. The claim that is made in this paper is quite strong, by questioning the conclusion of docents of tACS studies. It would be important to base this claim on equally strong data. Given that some of the effects are weak, with PLV raising only a bit above noise levels and only 9 responders out of 12, this study seems underpowered. The human experiments do not seem overly complicated and could be readily reproduced on a larger cohort. A good way to do this would be to use the current data to establish required sample size with a corresponding power analysis, and then repeat the experiments exactly on a larger sample with identical outcome measures and statistical tests. This would be essentially a replication of the current experiment and, if successful, lend quite a bit of support for the strong claims made here. Collecting data as shown in figure 6 may go a long way to show that differences across subjects are larger than differences across stimulation sites, pointing at a common mechanism. But even just a repetition of Experiment #2, showing a significant interaction between anesthesia and amplitude would be helpful.

Reply:

We have previously investigated the effects of tACS on physiological tremor - Khatoun, A. et al. "Using high-amplitude and focused transcranial alternating current stimulation to entrain physiological tremor". Sci. Rep. 8, 4927 (2018). Before conducting the current study we used the data from Khatoun et al 2018 to perform a sample size calculation. This power analysis is reported in the Methods Section and indicated that 11 subjects would be sufficient to detect significant effects with a power of 0.8. Thus, Experiment 2A is not underpowered. We do agree that the effects on PLV

appear small, and not all subjects show a response. But this is typical for any tACS experiment. Based on a suggestion from another reviewer, we calculated and now report effect sizes (Cohen's d metric). This calculation shows that while the numerical change in PLV may appear small, the actual effect size that we measure is, in statistical terms, large.

Nevertheless, we do take the point that more data would strengthen our conclusions. However, since Experiment 2A is already correctly powered and shows large effect sizes, we believed it better to extend our findings from physiological tremor to include pathological tremor. Importantly, this is also in a different frequency range (~10 Hz vs ~3 Hz). Therefore, we included Experiment 2B which adds both more subjects showing a reduction in tACS effects with anesthesia and extends our findings to pathological tremor.

Unfortunately, only the four subjects in Supplementary Figure 6 completed both Experiments 2A and 3. Lastly, we would also reiterate the point that we have carefully edited the interpretation of our results throughout the manuscript.

Reviewer #2:

"Results from our study provide the first direct evidence that the effects of tACS on motor function are mostly caused by transcutaneous stimulation and not by transcranial stimulation as is widely assumed. The observed patterns of neural entrainment were similar for both transcutaneous-only and transcranial-only stimulation. This supports the hypothesis that a transcutaneous mechanism could be responsible for many of the tACS effects observed in humans."

All this strikes me as over-interpretation of the results. The rodent experiment showed the exact opposite of what is stated here. Namely, that transcranial-only was effective at entraining neuronal activity. No one would ever doubt that rhythmic activation of peripheral nerves causes rhythmic spiking in the brain. To the critics of tACS, the surprising finding is that the same happened when peripheral nerves were not present at all (transcranial-only). If anything, the effects seem to be stronger in that case (requiring only 1/10 of the intensity when stimulating transcranially).

Reply:

Again, this point comes back to a more careful and balanced interpretation of our results and has been answered in our response to previous comments. The paragraph mentioned here from the Discussion has been edited to reflect this. We now specifically state that weak electric fields in the brain do cause small amounts of neural entrainment, however, in the case of human motor system effects, the transcutaneous mechanism appears to cause most of the effects observed in our experiments.

Reviewer #2:

"The results from Experiments 2 and 3 show that the effects of tACS on physiological tremor are mostly caused by the transcutaneous, and not the transcranial mechanism." Again, there is no support in the data for the word "mostly" in this sentence. Rather, the data suggests that peripheral stimulation alone can entrain tremor. But it does in no way rule out direct brain stimulation effects.

Reply:

Again, see our response to previous comments. This sentence no longer appears in the revised manuscript.

Reviewer #2:

"Our results show that the effects of tACS on physiological tremor are not mediated by transcranial stimulation of neurons in the motor cortex but are caused by transcutaneous stimulation of

peripheral nerves in the skin.” And here again “not mediated by transcranial” is over-reaching. There is no evidence for that.

Reply:

This sentence has been changed to “Our results show that the effects of tACS on tremor appear unlikely to be mediated by transcranial stimulation of neurons in the motor cortex but instead are dominated by transcutaneous stimulation of peripheral nerves in the skin”. Based on results from combined experiments, we believe this to be a fair statement.

Reviewer #2:

“tACS effects on non-motor systems”. In this paragraph, again, there is an overreach. What is shown is that peripheral stimulation can entrain tremor, and this should motivate investigators to be careful in their interpretation of other tACS studies. However, the jump to say that all motor effects (beyond tremor entrainment) and all other hearing, cognitive and memory effects may all be the result of peripheral stimulation is exaggerated.

Reply:

We agree. We have now edited this paragraph to present a more balanced view.

Reviewer #2:

“Here, we have shown that the tACS mechanism, at least for the motor system, is mostly caused by transcutaneous stimulation and not transcranial stimulation. ... ” Again, too broad. You have shown tremor entrainment with peripheral stimulation. Nothing more. The “motor system” does a lot more than tremble.

Reply:

This sentence has been changed to “Our results indicate that the transcutaneous mechanism appears to play an important role in causing tACS motor system effects”.

Reviewer #2:

“In our study, we showed that the transcutaneous mechanism is the dominant mechanism driving tremor entrainment via tACS....Our results clearly show that tACS effects on the motor system are mostly caused by transcutaneous stimulation of peripheral nerves in the skin and not by transcranial stimulation of cortical neurons as had been previously assumed.” Again, “dominant”, “mostly” is too strong.

Reply:

We have edited this. The first part now reads “Results from our study indicate that the transcutaneous mechanism is likely to be the dominant mechanism driving tACS motor system effects”. The second part reads “Our results indicate that tACS effects on the motor system appear to be dominated by transcutaneous stimulation of peripheral nerves in the skin and not by transcranial stimulation of cortical neurons as had been previously assumed.”

Reviewer #2:

“Therefore, tACS induced phosphenes (which we assume are caused by electrical stimulation of the optic nerve 52,53 , although there is some debate on this 54)” There really is not much debate about this anymore since the publication of Kar & Krekelberg 2012.

Reply:

We have now changed this to ‘was debated’ and added the Kar and Krekelberg reference.

Reviewer #2:

The methods explain that the HIGH intensity condition was adjusted in steps of 0.1mA to be comfortable. Yet the table in the supplement suggests otherwise as these are mostly multiples of 0.5. Please clarify. Also, why did you not keep these numbers all constant at a less aggressive level that works for most subjects to reduce variability?

Reply:

It is difficult to measure an effect of tACS on physiological tremor. From experience (pilot experiments and Khatoun et al 2018) we found that in general increasing the amplitude of the stimulation increases the effect of tACS. Therefore, we opted to go for higher levels with the drawback that there is more variability in the amplitude between subjects. It is correct that the intensity was adjusted in steps of 0.1 mA. The fact that many amplitudes are in multiples of 0.5 mA arises because we aimed for a maximum value of 2.5 mA.

Reviewer #2:

Its a bit unusual to report so many digits for z-statistics.

Reply:

We now report less digits for the z-statistics.

Reviewer #3 (Remarks to the Author):

Given the rousing debate on whether or not tACS can effectively stimulate the human cortex, this innovative study appears timely and is potentially relevant for a broad neuroscientific readership, provided that the reported findings do actually generalize from the motor cortex to other brain regions and from the phenomenon of tACS-induced tremor entrainment to other cognitive functions. However, for this generalization currently no evidence is provided. While the authors do formally restrict their interpretation to the contribution of transcutaneous contributions to tremor entrainment, the gist and general tone of the paper goes far beyond that limited claim. While I strongly encourage critical scrutiny when it comes to the mechanisms underlying non-invasive brain stimulation techniques, and I very much appreciate the presented study, particular care must be taken not to erroneously put all entrainment effects of tACS under general suspicion based on this particular finding.

Reply:

We thank the reviewer for their constructive input. One of the reviewer's main concerns (in agreement with other reviewers) was that we might have over interpreted our results or generalized too much in our conclusions. We agree with this criticism. We have substantially revised the manuscript to present a more limited interpretation of our results and a more balanced view of the field. Additionally, based on comments for other reviewers we have added two new experiments: Experiment 2A now shows in 12 essential tremor patients (6 without anesthesia and 6 with anesthesia) that tACS effects on pathological tremor are mostly caused by transcutaneous stimulation. Experiment 4 now shows that peripheral nerve stimulation (using rhythmic pulsed stimulation instead of sinewave to reduce the electrical artifact) causes entrainment of EEG beta band activity. We believe the revised manuscript has been improved due to the reviewer's input and with the addition of the new experiments. Below we provide a more detailed response to the specific concerns.

Reviewer #3:

I have the following specific concerns:

1. It has to be made very clear in the title, abstract, and throughout the paper that conclusions can only be drawn for the contribution of transcutaneous tACS effects on tACS-induced tremor entrainment, and not to any other stimulation site or cognitive function.

Reply:

The manuscript (Abstract, Introduction, Results and Discussion) has been thoroughly edited to make clear that our results are restricted to tACS effects on the motor system. The new experiments actually expand our findings to both pathological tremor and somatosensory-motor cortex EEG activity and thus strengthen this specific conclusion. We make it clear that our results do not extend to other systems or non-motor function. We simply state that it would now be interesting to check any potential role of the transcutaneous mechanism for tACS non-motor system effects.

Reviewer #3:

2. While the study shows a contribution of transcutaneous tACS effects on tACS-induced tremor entrainment, it does not disprove that there is generally a transcranial contribution to this phenomenon. In particular, other studies have used different electrode montages (M1 to ipsilateral arm) and may have more effectively stimulated the relevant brain regions than the local centre-surround montage used in the current study. Also, since local anaesthesia did neither completely abolish tremor entrainment, nor did it abolish sensory perception of the stimulation, the two factors remain entangled.

Reply:

The reviewer raises a valid point (similar to comment 3 below): the weak field in the brain may cause some direct neural entrainment that has a contribution to tremor entrainment. Indeed, unfocused tACS montages with non-cephalic return electrodes would draw more current into the head. We now mention this in the Discussion, *tACS electrode montage*. On the other hand, these unfocused montages will also cause stimulation of the optic nerve (even if it is below the perceptual threshold) and a non-cephalic return electrode will cause peripheral nerve stimulation. Thus, the stronger field in the brain for an unfocused montage comes with a number of limitations. Lastly, the new data in essential tremor patients shows that even with a stronger electric field in the brain (we increased stimulation to 5 mA in the anesthesia group), we still did not observe tremor entrainment when the peripheral nerves were blocked.

Nevertheless, we acknowledge the important point raised by the reviewer that we did not completely rule out a contribution from the weak electric field in the brain. And, as the reviewer states in Comment 3, our animal study actually shows that these weak field do cause some neural entrainment.

We now highlight this point in the Discussion in two places: *tACS effects on non-motor systems*, 2nd paragraph and in *Limitations*.

However, our results still clearly show that, while there is a weak electric field in the brain that may or may not cause neural entrainment in the human motor cortex (we don't yet have direct data), the transcutaneous mechanism is the dominant mechanism driving tremor entrainment. Results from Experiment 2A show this in healthy volunteers and Experiment 2B confirms this in essential tremor patients. The new results for Experiment 4 also show that peripheral nerve stimulation causes relatively strong entrainment in somatosensory-motor cortex EEG activity.

Reviewer #3:

3. While it is very difficult to match stimulation intensity for rat and human transcranial brain stimulation, the authors did their best and actually report entrainment of spiking in rat motor cortex by "transcranial-only" stimulation at intensities comparable to what can be achieved in humans. This result is actually supporting rather than disproving the possibility for a truly transcranial contribution.

Reply:

This comments overlaps with Comment 2, and we have answered both points in our reply to Comment 2. Briefly, we now draw the reader's attention to this fact in the Discussion and at the relevant section in the Results. Again, we agree that the weak field in the brain may cause some neural entrainment but our results show that most of the tACS effect on tremor is caused by the transcutaneous mechanism. We believe the revised manuscript does a better job of presenting these ideas and results in a more balanced way.

Reviewer #3:

4. The neuronal mechanism behind the suggested peripheral transcutaneous entrainment effect on M1 remains obscure and is not discussed by the authors. If it is mediated via afferent input to S1 and thus indirectly to M1, it should be mainly contralateral to the stimulated limb (experiment 1), but it works equally well for both left and right limbs. Even more unclear are the mechanisms by which somatosensory stimulation anywhere on the body surface would entrain neuronal activity everywhere in the brain. In experiment 1, spiking data was only acquired from left M1, while a generalization to other brain regions would have required recordings of other (non-motor, e.g. visual and associative) cortical areas.

Reply:

In the revised manuscript we now limit our interpretation of the results to tACS effects on the motor system. We now simply say that it would be interesting to investigate a potential role of the

transcutaneous mechanism on tACS non-motor system effects. Thus we believe, a full discussion of that potential mechanism is not necessary in the revised manuscript. However, a discussion of the potential mechanism through which peripheral nerve stimulation entrains neural activity in the motor cortex is merited and has now been included in the Discussion.

Reviewer #3:

5. In analogy to experiment 2, the ultimate proof of a peripheral origin of the observed entrainment of spiking in rat motor cortex would be a deafferentation, lesioning the nerves projecting from a locally stimulated limb (with both electrodes on the same limb (as in experiment 3) to the motor cortex.

Reply:

We are confident that the limb stimulation in the rat experiments caused neural entrainment via peripheral nerve stimulation. The alternative mechanism (suggested by another reviewer) could be that current from limb stimulation spread out into the brain (through volume conduction) and thus causes direct stimulation of neurons in the cortex (similar to the transcranial-only condition). However, we now provide an analysis of our data that shows this is not the case. We believe the presentation of this new data and analysis in the Supplementary Results (see Supplementary Figs. 1 and 2) is a good alternative to the deafferentation experiment suggested by the reviewer.

Our experimental setup allowed us to monitor the electric field in the brain caused by both the transcranial-only stimulation condition and the transcutaneous-only stimulation condition. Fig. 1 in the supplementary methods outlines our procedure for doing this. Briefly, the stimulation (either transcranial or transcutaneous) is a low frequency sine wave and the voltage in the brain at this frequency was directly measured using the 32-channel recording probe. This data was used to establish that our transcranial-only stimulation caused an electric field strength in the brain of around 1 V/m for every 0.1 mA of transcranially applied current. We now used exactly the same measurement procedures to quantify the electric field strength in the brain during transcutaneous-only stimulation and found that for 3 mA transcutaneous-only we get an electric field of around 0.07 V/m in the brain. This is in the same range as the local field potential strength. Indeed, using the same procedures we quantified the average electric field strength of the local field potential during stimulation off condition for both the transcranial-only and the transcutaneous-only conditions and found it to be 0.041 V/m and 0.037 V/m respectively. Thus, the 0.07 V/m electric field measured during 3 mA of transcutaneous-only stimulation is likely to be due to the local field potential and not volume conduction from the stimulated limb. In any case, even if it was due to volume conduction it is still well below the 1 V/m need to cause direct neural entrainment of cortical neurons. We have now added a new figure to Supplementary Results (Figure 2) fully explain and describing these results and we refer to this in the main text.

Reviewer #3:

6. It is well possible that both transcranial and transcutaneous stimulation effects can cause entrainment of spiking and physiological tremor, and the fact that both stimulation forms cause similar entrainment in experiment 1 supports this notion. It is even conceivable that both mechanisms work in parallel without the effects accumulating, but with each one per se being sufficient to entrain spiking and tremor, respectively. In both rats and humans, concurrent transcutaneous and transcranial stimulation (with an arbitrarily phase-delay between the stimulation signals) would reveal which one provides the predominant entrainment of M1

Reply:

We acknowledge the important point made by the reviewer. Indeed, our data from Experiment 1 do show that weak electric fields in the brain (down to around 1 V/m) can cause neural entrainment. We now make this clear in the revised manuscript at a few locations: In a paragraph at the end of the Results , *Experiment 1* and at a few places in the revised Discussion. However, we believe that our data from Experiments 2A show that tremor entrainment is significantly reduced with anesthesia and in Experiment 2B it appears to disappear completely with anesthesia in essential tremor patients. Thus, while the field in the brain will be present during tACS it appears to contribute very little, if at all, to tremor entrainment. Our data from the new Experiment 4 now also show that even very weak rhythmic stimulation of peripheral nerves, caused significant entrainment of EEG activity originating from the somatosensory-motor cortices. Thus, we believe the evidence presented in the revised manuscript points towards transcutaneous stimulation being the dominant mechanism for tACS motor effects. However, we do now provide a more balanced interpretation of our results in the revised manuscript.

Reviewers' comments:

Reviewer #1 (Remarks to the Author):

In the revised manuscript the authors addressed many concerns. Nevertheless an uncomfortable feeling remains with this manuscript after having read also the comments of the other reviewers and statements of the authors themselves (e.g. "It is difficult to measure an effect of tACS on physiological tremor." What essentially remains is that the authors draw attention to the (well known) fact that stimulation of sensory afferents can entrain brain activity. They fail however to convincingly show a (tentative) inefficacy of tACS for entrainment. The uncomfortable feeling is further facilitated by complex analyses throughout all experiments making judgement on the quality and reproducibility of data difficult. The tremor analysis e.g. is difficult to understand, something like the length of a path "travelled" by a finger or hand comparing different stimulation conditions would be easier to understand. As I wrote in my first review, experimental quality deficits may not be compensated by quantity.

Remaining issues:

1. There is an inconsistency in values of the E-field enough for any significant amount of entrainment: in description of Supplemental Figure 2: 0,9 V/m, in discussion: 1V/m.
2. "The interaction between tACS amplitude and scalp anesthesia was not significant (same linear mixed model, $p=0.294$). A post-hoc analysis would not allow us to draw firm conclusions as to which specific conditions caused significantly more or less tremor entrainment." Using a one-tailed statistical test is appropriate when having a clear hypothesis and justified predictions for the outcomes. Here this probably applies for stimulation intensity (off, low or high) with an expected linear entrainment, however testing the conditions anesthesia versus non-anesthesia may need a two-tailed Wilcoxon sign rank test? The statement is controversial to further conclusions in the reported results: "The no anesthesia LOW and HIGH amplitude conditions caused more entrainment than the OFF condition".
3. Ref. 13 year missing
4. "Results from in-vivo animal experiments and slice work show that electric fields of that strength cause only minimal or no neural entrainment²⁹⁻³¹" of course slices need higher entrainment currents because of deafferentation. The slice reference needs either to be eliminated in the present context or dealt with separately.
5. "Thus, we suggest that for the reported effects on the motor system, the 't' in tACS may actually represent transcutaneous and not transcranial stimulation." I would prefer to add one word in parenthesis here: "Thus, we suggest that for the reported effects on the motor system, the 't' in tACS may actually represent transcutaneous and not (only) transcranial stimulation.
6. "In Experiment 3 we moved the tACS electrodes to the contralateral arm, effectively blocking the transcranial mechanism." Should read "In Experiment 3 we moved the tACS electrodes to the M1 projecting to the arm contralateral to tremor recording, hereby preventing a stimulation of the involved motor cortex, leaving however open an influence via transcallosal inhibition"
7. "...nerves can account for tACS effects on the motor system across a wide range of frequencies." Should read "... nerves can account for EEG influences similar to those induced by tACS of the motor system". A wide range of frequencies was not investigate
8. "In the group that did not receive topical anesthesia (n=6, right panel) motor cortex tACS did not entrain pathological tremor." The "not" needs to be removed.

Reviewer #2 (Remarks to the Author):

All my previous comments have been adequately addressed, and I commend the authors for performing two additional experiments. The new experiment on pathological tremor is interesting and compelling.

However, I have some concerns on the new EEG experiment with 20 Hz stimulation. The result is interesting, although not surprising, and it has an important flaw, which would need an additional control to rule out. First to the obvious: It is well known that periodic sensory stimulation, of any kind really (auditory, visual, tactile, nociceptive), will generate a steady-state evoked potentials at the stimulation frequency. And obviously it will be stronger for stronger stimuli. So this experiments really tells us nothing that we did not already know. Perhaps the experiment is novel in inducing this (tactile/nociceptive) sensation with electric pulses, and perhaps the frequency following response had not been demonstrated up to 20Hz for for somatosensory stimuli, but regardless, we don't really learn much new from it. I guess it is still valuable as a reminder for those readers that do not know about steady-stave evoked potentials. Now to the flaw with the experiment: A pulse presented at 30Hz, no matter its exact waveform or duration will generate a strong 30Hz voltage artefact. The authors admit that stimulation artefacts appear at the scalp surface when stimulating the periphery with sinusoids. So the fact that they find a 30Hz signal on the scalp that scales with intensity is entirely meaningless. The argument has to be that the neural frequency following response is stronger than this artefact. There are two ways I can see demonstrating this, and preferably both methods should be used to rule out this confound: 1. You could argue that the strength of the artifact is smaller, or much smaller, than the neural response. To do so, you would use the exact same equipment and the exact same electrode locations, except(!) that you would swap stimulator and recording, i.e. you would conect the "EEG" to the arm, and the stimulator to the scalp. The signal due to volume conduction should be exactly the same. Then repeat the exact same experiment with the exact same current magnitudes and data analysis. If you do not get the same result (or at least a significantly weaker result) then you have shown that you are measuring something more than volume conduction. 2. An additional and perhaps more compelling control would be to show that the activity you measured at the scalp has a spatial distribution consistent with cortical origin. For that you would show the distribution of the evoked steady state potentials across the entire scalp. If you get a locally constrained distribution, ideally over sematosensory cortex with dipolar appearance, then you have demonstrated your case.

The authors may very well decide that, given the obviousness of the result, this extra work is not warranted. In my view nothing would be lost if this experiment 4 was removed altogether. But if they want to keep it in, I think one of these controls is necessary.

One small item:

In the response you write: "The skin may be thinner in the rat, but hair was not removed. Thus, one may expect reasonably similar stimulation amplitudes to be needed for the transcutaneous experiments in rats and humans. Concerning this, the transcutaneous levels chosen here are in line with most studies in humans." I am afraid that you are missing an important point: smaller electrode and smaller conductive medium (rat vs human) leads to much different field magnitudes for the identical current (fields scale quadratically with length scale). On the small dimensions in the animal, a 2mA current leads to large fields. In fact, you show at least a factor of 10 larger fields for the same currents. When you write "These current amplitudes are in a similar range to those used in human tACS", I suggest to add something like "... although they are expected to cause larger field amplitudes due to the smalled dimensions (electrode size and distance, as well as tissue dimensions)". If you want a citation for that, here is a reference explaining this based on first principles: Jacek P. Dmochowski, Marom Bikson, Lucas C. Parra, "The point spread function of the human head and its implications for transcranial current stimulation," *Physics in Medicine and Biology*, 57:1-19, 2012.

Reviewer #3 (Remarks to the Author):

The authors added additional experiments and analyses to corroborate their claim that the entrainment of physiological and pathological tremor by tACS is mediated in large parts by the associated somatosensory co-stimulation rather than by transcranial currents. I do see my previous concerns with respect to that specific claim resolved, and I am convinced of a considerable transcutaneous contribution of tACS to the entrainment of tremor. This is a very good paper, BUT:

In my original review I stated the following:

"Given the rousing debate on whether or not tACS can effectively stimulate the human cortex, this innovative study appears timely and is potentially relevant for a broad neuroscientific readership, provided that the reported findings do actually generalize from the motor cortex to other brain regions and from the phenomenon of tACS-induced tremor entrainment to other cognitive functions. However, for this generalization absolutely no evidence is provided. [...] In my opinion, the paper should be toned down in its claims and would better fit a more specialized journal." This assessment has not changed with respect to the revised version of the manuscript. The specific claim (see above) has been further corroborated, but the results do still not generalize to other brain regions or functions. While this is now largely acknowledged by the authors, having much reduced consequences for the scientific community, its limited consequences make this manuscript in fact better suited for a more specialized audience.

I also stated the following (the part omitted above):

"While the authors do formally restrict their interpretation to the contribution of transcutaneous contributions to tremor entrainment, the gist and general tone of the paper goes far beyond that limited claim. While I strongly encourage critical scrutiny when it comes to the mechanisms underlying non-invasive brain stimulation techniques, and I very much appreciate the presented study, particular care must be taken not to erroneously put all entrainment effects of tACS under general suspicion based on this particular finding."

The abstract still starts by questioning the validity of tACS in general instead of focussing on tremor entrainment, and I do see absolutely no reason to keep up the humorous suggestion to attribute the "t" in tACS to "transcutaneous", since there is no evidence it affects the entire technique, but merely the interpretation of a relatively small number of studies. The title and the abstract should thus be clearly rephrased to refer to the motor cortex and specific kind of studies affected (tremor entrainment). The current title and abstract are strongly misleading and may lead to a misconception in the broader scientific community and the media.

Reviewer #1 (Remarks to the Author):

In the revised manuscript the authors addressed many concerns. Nevertheless an uncomfortable feeling remains with this manuscript after having read also the comments of the other reviewers and statements of the authors themselves (e.g. “ It is difficult to measure an effect of tACS on physiological tremor.” What essentially remains is that the authors draw attention to the (well known) fact that stimulation of sensory afferents can entrain brain activity. They fail however to convincingly show a (tentative) inefficacy of tACS for entrainment. The uncomfortable feeling is further facilitated by complex analyses throughout all experiments making judgement on the quality and reproducibility of data difficult. The tremor analysis e.g. is difficult to understand, something like the length of a path “travelled” by a finger or hand comparing different stimulation conditions would be easier to understand. As I wrote in my first review, experimental quality deficits may not be compensated by quantity.

Reply:

We thank the reviewer for their comments on the revised manuscript. We have taken onboard the remaining issues pointed out and replied to those individually below.

We find it unfortunate that the reviewer has an uncomfortable feeling about the manuscript. We believe that the experiments and analysis are clearly explained, and that the results are transparent. We do not believe that our analysis methods are overly complex and we strongly believe they are reproducible. In fact, we selected the phase locking analysis methods for both the rat and human experiments because they have already been used by other authors in the tACS field (Ozen, S. et al. *J. Neurosci.*, 2010; Mehta, A. R., Pogosyan, A., Brown, P. & Brittain, J.-S. *Brain Stimul.*, 2015), and we could reproduce them in this manuscript.

The suggestion by the reviewer to use ‘length of path traveled by the finger or hand’ (i.e. tremor amplitude), is unlikely to work since it has already been shown that tACS can entrain physiological tremor but has no (or only a minimal) effect of physiological tremor amplitude (Mehta, A. R., Pogosyan, A., Brown, P. & Brittain, J.-S. *Brain Stimul.*, 2015; Khatoun, A. et al. *Sci. Rep.* 8, 2018).

The reviewer makes the following comment: “What essentially remains is that the authors draw attention to the (well known) fact that stimulation of sensory afferents can entrain brain activity”. We believe it is vitally important that point is made clear, since almost every tACS paper published to date has overlooked this as a potential mechanism. The fact that peripheral nerve stimulation causes brain entrainment maybe well known in some fields, but does not appear to be well know, or its influence understood, in the tACS field.

Lastly, we disagree with the following comment: “They fail however to convincingly show a (tentative) inefficacy of tACS for entrainment”. Our results in Experiment 2 show that standard tACS does cause tremor entrainment. However, through the use of a scalp anesthetic, we clearly show that entrainment is mediated by stimulation of peripheral nerves in the scalp and not transcranial stimulation of the motor cortex.

Reviewer #1:

Remaining issues:

1. There is an inconsistency in values of the E-field enough for any significant amount of entrainment: in description of Supplemental Figure 2: 0.9 V/m, in discussion: 1V/m.

Reply:

In the Discussion, when we refer to the exact value for the rat experiments we have changed this to 0.9 V/m. At other points in the Discussion where we wish to give a guideline figure we state 'around 1 V/m'.

Reviewer #1:

2. "The interaction between tACS amplitude and scalp anesthesia was not significant (same linear mixed model, $p=0.294$). A post-hoc analysis would not allow us to draw firm conclusions as to which specific conditions caused significantly more or less tremor entrainment." Using a one-tailed statistical test is appropriate when having a clear hypothesis and justified predictions for the outcomes. Here this probably applies for stimulation intensity (off, low or high) with an expected linear entrainment, however testing the conditions anesthesia versus non-anesthesia may need a two-tailed Wilcoxon sign rank test? The statement is controversial to further conclusions in the reported results: "The no anesthesia LOW and HIGH amplitude conditions caused more entrainment than the OFF condition".

Reply:

Our hypothesis was that tACS effects were caused by stimulation of peripheral nerves in the scalp. Therefore, our prediction was that scalp anesthesia would cause a reduction in tremor entrainment. This was based on the well known mechanism whereby stimulation of sensory afferents entrain brain activity (as the reviewer points out earlier). Thus, a one-tailed statistical test is the correct test to use and we prefer to continue reporting this in the manuscript. Nevertheless, out of interest we did run a two-sided post-hoc test and still find a significant effect of anesthesia on tremor entrainment between the two LOW amplitude conditions (two-sided Wilcoxon signed-rank test, $p=0.031$).

"The no anesthesia LOW and HIGH amplitude conditions caused more entrainment than the OFF condition". This is simply a statement of the results of the exploratory post-hoc analysis and is never listed as a specific conclusion. The preceding paragraph makes this clear. Nevertheless, to avoid any confusion, we now begin the sentence with the following, "The exploratory post-hoc analysis shows that the no anesthesia LOW ..."

Reviewer #1:

3. Ref. 13 year missing

Reply:

The year has been added.

Reviewer #1:

4. "Results from in-vivo animal experiments and slice work show that electric fields of that strength cause only minimal or no neural entrainment^{29–31}" of course slices need higher entrainment currents because of deafferentation. The slice reference needs either to be eliminated in the present context or dealt with separately.

Reply:

We now point out that deafferentation will likely influence the electrical field strength needed to cause entrainment in the slice. However, it is unclear how this would affect the results – lack of afferents may make it more difficult to polarize a cell; while on the other hand lack of rhythmic input from other brain areas may make it easier to cause entrainment.

Reviewer #1:

5. "Thus, we suggest that for the reported effects on the motor system, the 't' in tACS may actually represent transcutaneous and not transcranial stimulation." I would prefer to add one word in

parenthesis here: “Thus, we suggest that for the reported effects on the motor system, the ‘t’ in tACS may actually represent transcutaneous and not (only) transcranial stimulation.

Reply:

We agree with the sentiment of what the reviewer wishes to convey. However, we prefer to do this by changing ‘actually represent’ to ‘mostly represent’. This sentence has also been slightly rephrased in the Introduction and Discussion: “We suggest that tACS effects may mostly be mediated by transcutaneous and not transcranial stimulation”.

Reviewer #1:

6. “In Experiment 3 we moved the tACS electrodes to the contralateral arm, effectively blocking the transcranial mechanism.” Should read “In Experiment 3 we moved the tACS electrodes to the M1 projecting to the arm contralateral to tremor recording, hereby preventing a stimulation of the involved motor cortex, leaving however open an influence via transcallosal inhibition”

Reply:

The statement as written in the manuscript is correct. We have not made the change suggested by the reviewer.

Reviewer #1:

7. “...nerves can account for tACS effects on the motor system across a wide range of frequencies.” Should read “... nerves can account for EEG influences similar to those induced by tACS of the motor system”. A wide range of frequencies was not investigate

Reply:

The wide range of frequencies refers to the effects measured across all experiments (not just the EEG experiment), which do cover a wide range of frequencies (~1 Hz, ~3 Hz, ~10 Hz and ~20 Hz). We now make this clear in the sentence which now begins “Combined results from all experiments show ...”.

Reviewer #1:

8. “In the group that did not receive topical anesthesia (n=6, right panel) motor cortex tACS did not entrain pathological tremor.” The “not” needs to be removed.

Reply:

This has been corrected.

Reviewer #2 (Remarks to the Author):

All my previous comments have been adequately addressed, and I commend the authors for performing two additional experiments. The new experiment on pathological tremor is interesting and compelling.

However, I have some concerns on the new EEG experiment with 20 Hz stimulation. The result is interesting, although not surprising, and it has an important flaw, which would need an additional control to rule out. First to the obvious: It is well known that periodic sensory stimulation, of any kind really (auditory, visual, tactile, nociceptive), will generate a steady-state evoked potentials at the stimulation frequency. And obviously it will be stronger for stronger stimuli. So this experiments really tells us nothing that we did not already know. Perhaps the experiment is novel in inducing this (tactile/nociceptive) sensation with electric pulses, and perhaps the frequency following response had not been demonstrated up to 20Hz for somatosensory stimuli, but regardless, we don't really learn much new from it. I guess it is still valuable as a reminder for those readers that do not know about steady-state evoked potentials.

Reply:

We agree with the reviewer that this result is not completely novel and is somewhat expected. Nevertheless, we believe it is important to keep it in the manuscript as it will demonstrate to readers, less familiar with steady-state evoked potentials, how tACS could cause brain oscillations through transcutaneous stimulation.

Reviewer #2:

Now to the flaw with the experiment: A pulse presented at 30Hz, no matter its exact waveform or duration will generate a strong 30Hz voltage artefact. The authors admit that stimulation artefacts appear at the scalp surface when stimulating the periphery with sinusoids. So the fact that they find a 30Hz signal on the scalp that scales with intensity is entirely meaningless. The argument has to be that the neural frequency following response is stronger than this artefact. There are two ways I can see demonstrating this, and preferably both methods should be used to rule out this confound: 1. You could argue that the strength of the artifact is smaller, or much smaller, than the neural response. To do so, you would use the exact same equipment and the exact same electrode locations, except(!) that you would swap stimulator and recording, i.e. you would connect the "EEG" to the arm, and the stimulator to the scalp. The signal due to volume conduction should be exactly the same. Then repeat the exact same experiment with the exact same current magnitudes and data analysis. If you do not get the same result (or at least a significantly weaker result) then you have shown that you are measuring something more than volume conduction. 2. An additional and perhaps more compelling control would be to show that the activity you measured at the scalp has a spatial distribution consistent with cortical origin. For that you would show the distribution of the evoked steady state potentials across the entire scalp. If you get a locally constrained distribution, ideally over somatosensory cortex with dipolar appearance, then you have demonstrated your case.

The authors may very well decide that, given the obviousness of the result, this extra work is not warranted. In my view nothing would be lost if this experiment 4 was removed altogether. But if they want to keep it in, I think one of these controls is necessary.

Reply:

We agree with the reviewer that a 20 Hz stimulation pulse presented on the arm could in theory create an electrical stimulation artifact at 20 Hz in the vicinity of the recording electrodes on the scalp. While our low-pass filter, set at 100 Hz, will remove the high frequency components of the square wave pulse, a low frequency artifact could remain. In practice, we find this artifact to be,

depending on the subject, either very small or simply not present. If present, it is always much smaller than any neural response and close to the noise floor.

To demonstrate this we have adopted one of the reviewer's suggestions for a control: we switched the recording and stimulating electrodes so that we record from the arm (where there is no neural response) and stimulate on the head (which, due to electrical reciprocity principals, should still create an equivalent artifact). We have also developed a second control of our own: using the standard setup we collect an amplitude growth function of the neural response. If the response we record is dominated by the artifact it should scale linearly with increasing stimulation amplitude (i.e. the volume conducted artifact linearly increases as the stimulation pulse amplitude is increased). However, if the response we record is a true neural response and has little or no artifact, we should find that at some (subthreshold) amplitudes there is no neural response. Then as we increase stimulation amplitude we should see that the neural response appears and increases, possibly in a nonlinear way.

We measured both these control conditions in four of the subjects who participated in Experiment 4. The results are now included in Supplementary Results, Fig 13. The results show that for all practical purposes there is no stimulation artifact present.

Reviewer 2#:

One small item:

In the response you write: "The skin may be thinner in the rat, but hair was not removed. Thus, one may expect reasonably similar stimulation amplitudes to be needed for the transcutaneous experiments in rats and humans. Concerning this, the transcutaneous levels chosen here are in line with most studies in humans." I am afraid that you are missing an important point: smaller electrode and smaller conductive medium (rat vs human) leads to much different field magnitudes for the identical current (fields scale quadratically with length scale). On the small dimensions in the animal, a 2mA current leads to large fields. In fact, you show at least a factor of 10 larger fields for the same currents. When you write "These current amplitudes are in a similar range to those used in human tACS", I suggest to add something like " ... although they are expected to cause larger field amplitudes due to the smaller dimensions (electrode size and distance, as well as tissue dimensions)". If you want a citation for that, here is a reference explaining this based on first principles: Jacek P. Dmochowski, Marom Bikson, Lucas C. Parra, "The point spread function of the human head and its implications for transcranial current stimulation," *Physics in Medicine and Biology*, 57:1-19, 2012.

Reply:

We thank the reviewer for pointing this out. We have now added a cautionary clause to the sentence and make reference to the suggested paper.

Reviewer #3 (Remarks to the Author):

The authors added additional experiments and analyses to corroborate their claim that the entrainment of physiological and pathological tremor by tACS is mediated in large parts by the associated somatosensory co-stimulation rather than by transcranial currents. I do see my previous concerns with respect to that specific claim resolved, and I am convinced of a considerable transcutaneous contribution of tACS to the entrainment of tremor. This is a very good paper, BUT:

In my original review I stated the following:

"Given the rousing debate on whether or not tACS can effectively stimulate the human cortex, this innovative study appears timely and is potentially relevant for a broad neuroscientific readership, provided that the reported findings do actually generalize from the motor cortex to other brain regions and from the phenomenon of tACS-induced tremor entrainment to other cognitive functions. However, for this generalization absolutely no evidence is provided. [...] In my opinion, the paper should be toned down in its claims and would better fit a more specialized journal."

This assessment has not changed with respect to the revised version of the manuscript. The specific claim (see above) has been further corroborated, but the results do still not generalize to other brain regions or functions. While this is now largely acknowledged by the authors, having much reduced consequences for the scientific community, its limited consequences make this manuscript in fact better suited for a more specialized audience.

Reply:

We thank the reviewer for their time and effort. We appreciate that the reviewer acknowledges the quality of the manuscript. However, we do not agree with the view that it would be better suited to a more specialized audience. We continue to believe that the revised manuscript will be of great interest to a broad scientific readership.

Reviewer #3:

I also stated the following (the part omitted above):

"While the authors do formally restrict their interpretation to the contribution of transcutaneous contributions to tremor entrainment, the gist and general tone of the paper goes far beyond that limited claim. While I strongly encourage critical scrutiny when it comes to the mechanisms underlying non-invasive brain stimulation techniques, and I very much appreciate the presented study, particular care must be taken not to erroneously put all entrainment effects of tACS under general suspicion based on this particular finding."

The abstract still starts by questioning the validity of tACS in general instead of focusing on tremor entrainment, and I do see absolutely no reason to keep up the humorous suggestion to attribute the "t" in tACS to "transcutaneous", since there is no evidence it affects the entire technique, but merely the interpretation of a relatively small number of studies. The title and the abstract should thus be clearly rephrased to refer to the motor cortex and specific kind of studies affected (tremor entrainment). The current title and abstract are strongly misleading and may lead to a misconception in the broader scientific community and the media.

Reply:

We agree with the reviewer that the title of the manuscript may lead to misconceptions and that we should be careful not to "put entrainment effects of tACS under general suspicion". Therefore, we have now changed the title to: "tACS motor system effects can be caused by transcutaneous stimulation of peripheral nerves". We believe this accurately reflects our findings.

In line, with the reviewers suggestions, we have made several changes to the abstract: We have removed the 't in tACS' part from the abstract (and from the rest of the manuscript). Additionally, we now clearly state in the final sentence that it is debatable whether or not our finding would generalize to all non-motor system tACS effects. We would also stress that we do not aim to question the validity of tACS results. We only question the assumed mechanism through which they are mediated. We have now added a few words to the abstract to make this point clear. We believe that the abstract (already heavily revised after the first review) now presents a balanced view of the findings. We do begin by describing the general principal of tACS and the main results in the field. However, we think this is important to put our results in context.

Reviewers' comments:

Reviewer #1 (Remarks to the Author):

My uncomfortable feeling about the validity of the data has not been resolved. I still (in agreement with reviewer 3) believe that these data should at best be published in a specialized journal in line with the new (now much better) title "tACS motor system effects can be caused by transcutaneous stimulation of peripheral nerves".

I thank the authors for drawing my attention to the Metha paper: "In fact, we selected the phase locking analysis methods for both the rat and human experiments because they have already been used by other authors in the tACS field (Ozen, S. et al. *J. Neurosci.*, 2010; Mehta, A. R., Pogosyan, A., Brown, P. & Brittain, J.-S. *Brain Stimul.*, 2015), and we could reproduce them in this manuscript." This is only partially true. The quoted paper by Metha is entitled "Montage Matters: The Influence of Transcranial Alternating Current Stimulation on Human Physiological Tremor". The result of this paper is just the opposite of that claimed here by the authors: The behavioral effects of transcranial alternating current stimulation appear to be critically dependent on the position of the reference electrode. If the present authors claim would be true, then montage should not matter. Interestingly this (in the present context most important) paper is not quoted (although the authors refer to it in the rebuttal letter!) and as a consequence could not even be discussed then. No further comments here.

"The suggestion by the reviewer to use 'length of path traveled by the finger or hand' (i.e. tremor amplitude), is unlikely to work since it has already been shown that tACS can entrain physiological tremor but has no (or only a minimal) effect of physiological tremor amplitude". This is indeed true in a way for the Metha paper, but not for at least one other paper of this group: "Brittain et al, *Current Biology* 23, 436-440." More generally I am still not impressed by using a key method of analysis claiming entrainment of tremor in which the most simple outread (tremor amplitude) is expected by the authors to be not altered.

"Lastly, we disagree with the following comment: "They fail however to convincingly show a (tentative) inefficacy of tACS for entrainment". Our results in Experiment 2 show that standard tACS does cause tremor entrainment. However, through the use of a scalp anesthetic, we clearly show that entrainment is mediated by stimulation of peripheral nerves in the scalp and not transcranial stimulation of the motor cortex." I had another look the Fig. 4. In the physiological tremor results the significance seems to be generated by two outliers, the pathological tremor group is severely underpowered with N=6. I still think that these are no convincing data, also in the context of the statement of the authors themselves on the difficulty of proving alterations in tremor recordings during the previous review round.

"Thus, a one-tailed statistical test is the correct test to use and we prefer to continue reporting this in the manuscript. Nevertheless, out of interest we did run a two-sided post-hoc test and still find a significant effect of anesthesia on tremor entrainment between the two LOW amplitude conditions (two-sided Wilcoxon signed-rank test, $p=0.031$)." I disagree: The authors now describe for which conditions one- or two-tailed tests were used. I still think that using a one-tailed test is not appropriate here, of course they reduce p-values twofold. As stated by the authors 'out of interest we did run a two-sided post-hoc test and still find a significant effect' with $p=0.031$ which as I understood (not sure if correct) the condition is exactly twice as much than reported in the manuscript. If all p-values would be recalculated with a two-tailed test some of the reported significances would not survive.

So, taking into account the sample size used for statistical testing (for example, 6 subjects in one group having received anesthesia and 6 subjects in the group without anesthesia in Exp.2A) and using the suggested post-hoc test I would assume that these results would not survive significance. Anyhow, a professional statistician should see this and I would follow his / her advice.

Reviewer #2 (Remarks to the Author):

My comments have been adequately addressed.

I only suggest that you add in the discussion a sentence or two discussing the interpretation of the new experiment 4 as a simple instance of steady-state evoked potentials. Not mentioning this would seem disingenuous. Perhaps you did already, but I can not tell as you have developed a poor habit of not reflecting in the response letter what exact edits you did, nor to mark up the manuscript with color to see the edit. For the future, you should try those simple aids to help reviewers get on your side quicker.

Reviewer #3 (Remarks to the Author):

In this revision my remaining concerns have been satisfactorily addressed. It is up to the Editor to decide whether the work is overall of sufficient interest to the journal's broad readership.

Response to Reviewer #1 Comments

Reviewer #1 raised a number of technical concerns about the manuscript in their final review. We thank the reviewer for pointing out these potential technical issues. However, as we will describe in detail below none of these technical concerns stand up to closer scrutiny.

"My uncomfortable feeling about the validity of the data has not been resolved. I still (in agreement with reviewer 3) believe that these data should at best be published in a specialized journal in line with the new (now much better) title "tACS motor system effects can be caused by transcutaneous stimulation of peripheral nerves".

I thank the authors for drawing my attention to the Metha paper: "In fact, we selected the phase locking analysis methods for both the rat and human experiments because they have already been used by other authors in the tACS field (Ozen, S. et al. J. Neurosci., 2010; Mehta, A. R., Pogosyan, A., Brown, P. & Brittain, J.-S. Brain Stimul., 2015), and we could reproduce them in this manuscript." This is only partially true. The quoted paper by Metha is entitled "Montage Matters: The Influence of Transcranial Alternating Current Stimulation on Human Physiological Tremor". The result of this paper is just the opposite of that claimed here by the authors: The behavioral effects of transcranial alternating current stimulation appear to be critically dependent on the position of the reference electrode. If the present authors claim would be true, then montage should not matter. Interestingly this (in the present context most important) paper is not quoted (although the authors refer to it in the rebuttal letter!) and as a consequence could not even be discussed then. No further comments here."

REPLY 1. The suggestion by the reviewer that we did not quote or refer to the Metha et al paper in our manuscript is not correct. By my count we refer to the Metha et al paper 3 times in the Discussion (4 in the now updated Discussion), 4 times in the Results and 1 time in the Methods. In fact, we have a section in the Discussion entitled 'tACS electrode montage' where we explicitly discuss the non-cephalic electrode montages used in the Metha et al study. Thus, we strongly reject the suggestion by the reviewer that we are trying to avoid discussing this paper.

The findings from Metha et al are in fact not in opposition to our results. Metha et al showed that different unfocused montages (note we only used focused montages) have different effects on tremor. In view of our findings (that tremor entrainment is caused by peripheral or cranial nerve stimulation) there are number of simple explanations for the Metha et al results: 1) The different montages and electrode positions cause differing amounts of peripheral nerve stimulation. 2) The different montages cause differing amounts of optic nerve stimulation. Data from Metha et al (we refer to Fig. 3B therein) show that all montages (except cM1) cause some amount of phosphenes – which they also show influence tremor entrainment. In the revised manuscript we have now added a specific paragraph to directly address the reviewer's concern.

"The suggestion by the reviewer to use 'length of path traveled by the finger or hand' (i.e. tremor amplitude), is unlikely to work since it has already been shown that tACS can entrain physiological tremor but has no (or only a minimal) effect of physiological tremor amplitude". This is indeed true in a way for the Metha paper, but not for at least one other paper of this group: "Brittain et al, Current Biology 23, 436-440." More generally I am still not impressed by using a key method of analysis claiming entrainment of tremor in which the most simple outread (tremor amplitude) is expected by the authors to be not altered."

REPLY 2. We chose tremor entrainment as a metric to assess tACS effects as it has been relatively widely used and reproduced by others. This is not the case with tACS effects on tremor amplitude (one of the two alternative metrics suggested the reviewer). Only one group published ground-

breaking findings on tACS effect on tremor amplitude in Parkinson's patients over 5 years ago (Brittain et al, Current Biology, 2013). However, that group has not followed-up on those ground-breaking findings and they have not been reproduced by other groups. This is puzzling given the huge potential for a noninvasive treatment of Parkinson's disease tremor. The same group also showed that tACS does not affect physiological tremor amplitude nor does it affect tremor amplitude in essential tremor patients. Thus, there is no experimental evidence to show that tremor amplitude is a better metric of tACS effects than tremor entrainment. The published experimental evidence actually shows that tremor entrainment is a more reproducible metric. The methods that we use to calculate tremor entrainment and phase locking values are standard. They are not overly complex and similar metrics have been used by many key papers in this field [e.g. Ozen, S. et al. J. Neurosci., 2010; Mehta, A. R., Pogosyan, A., Brown, P. & Brittain, J.-S. Brain Stimul., 2015; Brittain, J.S. et al Curr Biol. 2013].

In the first round of reviews, Reviewer #1 specifically suggested that, as an alternative to tremor entrainment, we should use peripheral nerve stimulation combined with EEG measurements: *"It would be much more straight forward to use methods to directly evaluate neuronal function, such as EEG or MEG, combined with the peripheral-only stimulation in experiment 3, to prove neural entrainment at the scalp level"*. We spent a significant amount of time and effort performing exactly the experiment requested by the reviewer and added this as a new experiment (Experiment 4). We find it unfortunate and difficult to understand that, having done exactly the experiment suggested by the reviewer, they are still not convinced by the data.

"I had another look the Fig. 4. In the physiological tremor results the significance seems to be generated by two outliers,..."

REPLY 3. A careful visual inspection of the physiological tremor data shown in Fig. 4 did not reveal to us any obvious outliers which may be driving the significance. However, to be statistically sure, we subjected the physiological tremor data to two different statistical testes to detect outliers: Grubbs test and MAD (greater than 3 scaled median absolute deviation away from the median). Neither test detected any outliers. We now report this in our Methods. Therefore, we are sure that the significance is not generated by outliers.

"...the pathological tremor group is severely underpowered with N=6."

REPLY 4. We do agree that the sample size in each pathological tremor group is small. Nevertheless, as we have shown it is enough to detect an effect of anesthesia, and enough to state that scalp anesthesia reduces the effect of tACS on pathological tremor. We also stress that the outcome from this one pathological tremor experiment must be viewed in the context of the entire manuscript. We have multiple experiments from both animals and healthy volunteers across a range of metrics (single unit data, EEG data, physiological and pathological tremor) that all point to a strong role for peripheral nerve stimulation in mediating tACS effects.

"I still think that these are no convincing data, also in the context of the statement of the authors themselves on the difficulty of proving alterations in tremor recordings during the previous review round."

see also Review Round 2: *"Nevertheless an uncomfortable feeling remains with this manuscript after having read also the comments of the other reviewers and statements of the authors themselves (e.g. 'It is difficult to measure an effect of tACS on physiological tremor.')*"

REPLY 5. The reviewer comes back a number of times to a statement we made in response to a comment from Reviewer #2 in the first review round. The reviewer cites this statement as evidence to support the 'uncomfortable' or unconvinced feeling they have about the validity of our data. We believe the reviewer may have misunderstood the original statement made by us and taken it somewhat out of context. In the first round of reviews, Reviewer #2 asked about why we used higher amplitudes and about reducing variability, to which we replied: *"It is difficult to measure an effect of tACS on physiological tremor. From experience (pilot experiments and Khatoun et al 2018) we found that in general increasing the amplitude of the stimulation increases the effect of tACS. Therefore, we opted to go for higher levels with the drawback that there is more variability in the amplitude between subjects"*. Thus, to state this in other words, it is well known and discussed within the field that there is high variability in the response to any kind of transcranial stimulation (tDCS or tACS). Subject variability is a hot topic of discussion at any tDCS /tACS meeting. We found that stimulating at higher amplitudes in general reduced the variability of the response to tACS. Thus, there is nothing inherently different or more difficult about tACS experiments performed in our lab than those carried out in any other tDCS/tACS lab. Thus, we believe that Reviewer #1 picked up on this statement and misunderstood it or interpreted it out of context. We hope this explanation now clarifies the matter for the reviewer.

"I disagree: The authors now describe for which conditions one- or two-tailed tests were used. I still think that using a one-tailed test is not appropriate here, of course they reduce p-values twofold. As stated by the authors 'out of interest we did run a two-sided post-hoc test and still find a significant effect ' with $p=0.031$ which as I understood (not sure if correct) the condition is exactly twice as much than reported in the manuscript. If all p-values would be recalculated with a two-tailed test some of the reported significances would not survive. So, taking into account the sample size used for statistical testing (for example, 6 subjects in one group having received anesthesia and 6 subjects in the group without anesthesia in Exp.2A) and using the suggested post-hoc test I would assume that these results would not survive significance. Anyhow, a professional statistician should see this and I would follow his / her advice."

REPLY 6. We disagree with the reviewer on this point. Since our hypothesis was consistently that anesthesia will reduce tremor entrainment, a one-tailed test is the correct statistical test. We predict that our scalp anesthesia will reduce tACS effects. There is no evidence to suggest scalp anesthesia could also increase tACS effects. Therefore, using a two-tailed test would be incorrect. The same can be said for stimulation amplitudes effect on tACS. We predict that higher amplitudes will increase entrainment. Therefore, a one-tailed test is appropriate.

However, the debate over using a one or two-tailed test is irrelevant for two reasons: A) Most of our main effects are tested and detected using a linear mixed model where one does not choose between a one or two-tailed test. B) Referring specifically to post-hoc testing, the suggestion by the reviewer that *"If all p-values would be recalculated with a two-tailed test some of the reported significances would not survive"*, is simply not correct. We explain these points with specific examples for each experiment taken directly from the manuscript:

In Experiment 1 we used a one-sided Wilcoxon signed rank test to compare between the OFF and ON conditions in the transcutaneous-only condition. The results were as follows: Wilcoxon signed rank test, one-sided, Bonferroni corrected ($n=12$) p-values and Z values: **contra-fore:** 1mA Z = -3.2 p = 0.007, 2mA Z = -3.5 p = 0.003, 3mA Z = -4.6 p < 0.001; **ipsi-fore:** 1mA Z = -2.8 p = 0.03, 2mA Z = -5.1 p < 0.001, 3mA Z = -4.3 p < 0.001; **contra-hind:** 1mA Z = -2.5 p = 0.07, 2mA Z = -3.6 p = 0.002, 3mA Z = -4.8 p < 0.001; **ipsi-hind:** 1mA Z = -3.0 p = 0.014, 2mA Z = -3.0 p = 0.015, 3mA Z = -2.6 p = 0.058). Thus, using a one-tailed test we detected a significant effect of transcutaneous-only stimulation on

PLV in 10 out of 12 conditions (only ipsi-hind 3 mA and contra-hind 2 mA didn't show significance). If we used a two-sided test we would still detect a significant effect in 9 out of 12 conditions (only the the ipsi-fore 1 mA condition would lose significance). Therefore, using a one or two-tailed test would not alter our main conclusion from Experiment 1.

In Experiment 2A we used a linear mixed model (LMM) to test for the effect of anesthesia and stimulation amplitude on physiological tremor entrainment. We found a significant effect of anesthesia ($p=0.020$) and stimulation amplitude ($p<0.001$). There is no question of using a one or two-tailed test here (point A above). In the exploratory post-hoc testing we used a one-tailed Wilcoxon signed-rank test to show that in the LOW amplitude conditions applying anesthesia caused a significant decrease in tremor entrainment ($p=0.015$). This is one of the main findings in our manuscript. This result would still be significant if we used a two-tailed test.

In Experiment 2A, we also used a one-sided Wilcoxon signed-rank test to show that increasing stimulation amplitude from OFF to LOW to HIGH in the no anesthesia condition increased tremor entrainment ($p=0.031$ and $p=0.031$). Using a two-tailed test the effect of stimulation amplitude on tremor entrainment would not reach significance. However, the effect of stimulation amplitude in Experiment 2A is not the focus of our manuscript and has no effect on our conclusions.

In Experiment 2B we don't use a LMM because of the simpler experimental design. Again, here the effect of anesthesia on pathological tremor was significant using a one-tailed test ($p=0.016$) and would still be significant using a two-tailed test.

In Experiment 3 we used an LMM to show that when tACS electrodes are placed on the arm increasing stimulation amplitude increases tremor entrainment ($p=0.013$). Again, no question of using a one or two-tailed test here (point A above). We used post-hoc testing, one-sided Wilcoxon signed rank test, to show that the LOW and HIGH conditions had significantly more entrainment than the OFF condition ($p = 0.024$ and $p=0.0103$). Again, these effects would still be present if we used a two-tailed test.

Exactly the same can be said for the results from EEG experiment suggested by the reviewer (Experiment 4) for both the PLV metric and the amplitude component metric.

As suggested by the reviewer we did already perform our statistics with the help of a professional statistician. We did this through a service offered to all KU Leuven researchers (<https://gbiomed.kuleuven.be/english/research/50000687/50000696>). We would happily have our data reviewed by any other professional statistician or by the reviewer.

Thus to summarize, we appreciate the time the reviewer has taken to point out a number of potential technical concerns about: the interpretation of our results in context of published literature (REPLY 1); experimental metrics (REPLY 2); statistical analysis (REPLY 3, 4, and 6); and experimental difficulty (REPLY 5). We now hope that our detailed replies to each of these points will convince the reviewer that their potential technical concerns are unfounded.

Response to Reviewer #2 Comments

"My comments have been adequately addressed. I only suggest that you add in the discussion a sentence or two discussing the interpretation of the new experiment 4 as a simple instance of steady-state evoked potentials."

A sentence has been added to the second paragraph of the Discussion.